# MIXUP FOR SURVIVAL ANALYSIS

## ABSTRACT

Survival analysis with censored outcomes underpins risk prediction in domains such as medicine and engineering. While deep survival models capture complex input-output relationships, they may benefit from vicinal risk minimization techniques such as mixup, which have proven effective as simple, model-agnostic data augmentation methods. We introduce a mixup for survival analysis, called H-Mixup, a principled framework that adapts mixup to censored time-to-event data by defining augmentation strategies that yield interpolated outcomes consistent with valid survival trajectories, encouraging local linearity in hazard functions. Theoretically, we show that H-Mixup contracts empirical Rademacher complexity and can tighten generalization bounds, while noting that the overall bound still depends on vicinal bias, which varies with alignment between mixing assumptions and the underlying risk structure. Empirical results on semi-synthetic and real-world datasets suggest that H-Mixup improves predictive performance for deep survival models. Overall, this study addresses the gap between mixup and survival analysis, providing a general recipe for vicinal regularization via simple data augmentation in time-to-event modeling.

## 1 INTRODUCTION

Survival analysis, or time-to-event modeling, addresses a key question in predictive modeling: not just if an event will occur, but the probability of it occurring over a period of time. This paradigm is central in medicine, engineering, and the social sciences, underpinning clinical prognosis, reliability testing, and risk prediction. A defining and persistent challenge is right-censoring, where the true outcome is only partially observed because a study ends or a subject is lost to follow-up. Accordingly, principled treatment of right-censoring is essential for credible survival modeling (Kaplan & Meier, 1958; Cox, 1972). Classical statistical methods such as the Cox proportional hazards model (Cox, 1972) provided a foundation for decades, combining interpretability with practical utility. Yet these approaches rely on strong structural assumptions, such as proportional hazards and linear covariate effects, that often fail in modern, high-dimensional applications. Recent advances in deep survival analysis have introduced flexible methodologies that capture complex, nonlinear dependencies and time structure (Katzman et al., 2018; Fotso, 2018; Kvamme et al., 2019; Kvamme & Borgan, 2019; Nagpal et al., 2021; Han et al., 2021; Zhong et al., 2021; Vauvelle et al., 2023). Progress has also encompassed dynamic prediction from longitudinal data (Jarrett et al., 2018; Lee et al., 2020; Putzel et al., 2021; Zeng et al., 2025), multimodal fusion of imaging/omics/clinical signals (Vale-Silva & Rohr, 2021; Fu et al., 2023; Mao & Liu, 2024), and representation learning via self-supervision and contrastive objectives (Hong et al., 2022b;a; Lee et al., 2024; Thrasher et al., 2024; Li et al., 2025). Despite their flexibility, deep survival models share the usual challenges of deep learning: they can overfit and generalize poorly, loss landscapes are irregular, predictions may rely on spurious patterns in low-data regions, and performance can degrade under shift (Zhang et al., 2017; Neyshabur et al., 2017; Belkin et al., 2019; Jiang et al., 2020).

In modern machine learning, one of the most successful strategies to mitigate overfitting has been mixup (Zhang et al., 2018). Mixup regularizes learning by augmenting samples that are convex combinations of inputs and labels. This simple strategy has been shown to smooth decision boundaries, reduce memorization, and improve calibration across diverse domains (Verma et al., 2019a; Thulasidasan et al., 2019). The theoretical appeal of the vicinal risk minimization (VRM) lies in its ability to approximate the true data distribution by enriching the training distribution with local vicinities, thereby improving generalization (Chapelle et al., 2001). Yet despite its widespread success, mixup

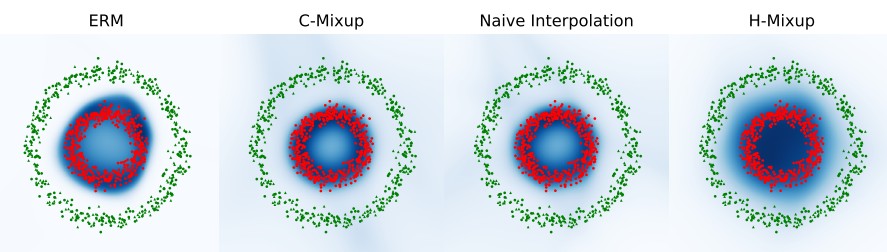

Figure 1: Motivating example comparing ERM, C-Mixup (Yao et al., 2022), naive interpolation (equation 4), and H-Mixup (equation 5; proposed). Red dots are high-risk instances; green dots are low-risk. Blue shading indicates predicted survival probability (lighter is larger) at median follow-up. ERM, C-Mixup, and naive interpolation spuriously assign low risk to the center of the high-risk cluster, pushing risk to the boundaries. In contrast, H-Mixup preserves high central risk while smoothing the vicinity. While C-Mixup and naive interpolation average follow-up times without a coherent vicinal distribution, H-Mixup achieves the desired smoothing in covariate-time space. Further details are in Appendix A.

has remained almost entirely absent from survival analysis. Survival outcomes are not simple labels, but partially observed manifestations of a latent generative process governed by censoring. Naively interpolating survival outcomes may violate the probabilistic structure of the problem and produce invalid synthetic data. This reveals a fundamental methodological gap: while VRM has been extensively studied and applied in classification and regression, there is no established framework that makes it compatible with survival analysis. This gap is more than a technical inconvenience. It highlights a broader misalignment between the general-purpose theories of mixup and the specialized demands of deep survival analysis. On one side, mixup offers strong empirical and theoretical evidence for its role in improving generalization and robustness in deep networks. On the other, survival analysis presents one of the most high-stakes domains for machine learning. Bridging these two has the potential to unlock the regularization benefits of mixup for survival models, but doing so requires rethinking how mixup interacts with censored outcomes.

In this paper, we introduce the first theoretically grounded mixup framework designed for survival analysis. Mixup was originally developed for classification and regression tasks. Survival analysis involves censored time-to-event outcomes, making naive interpolation of raw data invalid, as illustrated in Figure 1. We extend mixup to the survival analysis by shifting the focus from observed censored outcomes to the latent event-time distribution, deriving a data augmentation scheme that provides the regularization benefits of mixup while yielding a vicinal distribution valid under right-censoring. Our key contributions are:

- Novel algorithmic framework: We propose the first mixup strategy, called H-Mixup, for survival analysis that interpolates covariates and hazards through data augmentation, providing a principled extension of mixup to time-to-event data.

- Theoretical foundations: We show that H-Mixup encourages local linearity in the hazard function and can lead to tighter generalization bounds than empirical risk minimization (ERM).

- Comprehensive validation: We demonstrate on semi-synthetic and real-world datasets that H-Mixup consistently improves performance across diverse deep survival models.

In summary, our work bridges mixup and survival analysis, showing that its theory and practice can be extended to handle time-to-event data with right censoring.

## 2 MIXUP FOR SURVIVAL ANALYSIS

### 2.1 PRELIMINARIES

We consider a standard survival analysis framework for right-censored data. Let $T \geq 0$ be a continuous random variable representing the time to an event of interest, and let $C \geq 0$ be the censoring time. For each subject $i = 1, \ldots, n$ in a given dataset, we observe a covariate vector $x_i \in \mathbb{R}^p$.

The true event and censoring times, denoted $t_i$ and $c_i$, are not observed. Instead, the observed data for subject $i$ consists of the triplet $(x_i, o_i, \delta_i)$, where $o_i = \min(t_i, c_i)$ is the observed follow-up time and $\delta_i = \mathbb{I}(t_i \leq c_i)$ is the event indicator, which takes the value 1 if the event is observed and 0 otherwise. The dataset has $n$ independent and identically distributed (i.i.d.) observations $\mathcal{D} = \{(x_i, o_i, \delta_i) : i = 1, \cdots, n\}$. In this study we assume non-informative censoring, which posits that the event time $T$ and censoring time $C$ are conditionally independent given the covariate vector $X$. Formally, $T \perp C|X$ This condition ensures that, conditional on the covariates, the censoring mechanism provides no additional information regarding the time to event. The objective of survival analysis is to formulate a model for the event-time distribution conditional on the covariates. Let this model be parameterized by a vector $\theta \in \Theta$, where $\Theta$ is the parameter space. The model is characterized by the conditional survival function $S_\theta(t|x) = P_\theta(T > t|X = x)$ and the corresponding probability density function $f_\theta(t|x) = -\frac{d}{dt}S_\theta(t|x)$. The instantaneous risk of an event is given by the hazard function, $h_\theta(t|x) = f_\theta(t|x)/S_\theta(t|x)$. The parameters $\theta$ are estimated by maximizing the likelihood of the observed data $\mathcal{D}$. The likelihood contribution for the subject $i$ is the probability density $f_\theta(o_i|x_i)$ if the event is observed ($\delta_i = 1$), and the survival probability $S_\theta(o_i|x_i)$ if the observation is censored ($\delta_i = 0$). Given the assumption that the observations are i.i.d., the full negative log-likelihood function for the training data is:

$$\ell(\theta, \mathcal{D}) = -\sum_{i=1}^{n} [\delta_i \log h_\theta(o_i|x_i) + \log S_\theta(o_i|x_i)]. \tag{1}$$

In the context of machine learning, it is standard practice to minimize the negative log-likelihood, which serves as the loss function.

Mixup is a VRM scheme (Chapelle et al., 2001; Zhang et al., 2018) that forms convex combinations of input and target variables. Given two training examples $(x, y)$ and $(x', y')$ and a mixing weight $\lambda \sim \text{Beta}(\alpha, \alpha)$, mixup constructs

$$\tilde{x} = \lambda x + (1 - \lambda)x', \qquad \tilde{y} = \lambda y + (1 - \lambda)y', \tag{2}$$

and trains a model $\phi$ on $(\tilde{x}, \tilde{y})$ with a proper loss such as cross-entropy or mean squared error. Indeed, mixup encourages output linearity with respect to linear interpolations between pairs of inputs,

$$\phi(\lambda x + (1 - \lambda)x') \approx \lambda \phi(x) + (1 - \lambda)\phi(x'), \tag{3}$$

which smooths the hypothesis class in the vicinity of the data and thus equivalent to performing VRM with simple data augmentation.

In survival analysis, the target is observed through $(O, \Delta)$ and the goal is to regularize the mapping $x \mapsto \phi(\cdot|x)$, where $\phi$ denotes a hazard, survival, or event probability function, depending on the chosen survival model. Naively interpolating raw data points or directly applying techniques designed for regression, such as C-Mixup (Yao et al., 2022) which does not account for the censoring mechanism, does not appropriately induce vicinal distributions that yield smooth predictions across the joint input-time space.

Consider, a naive interpolation of the raw data points $(x, o, \delta)$ and $(x', o', \delta')$, defined as

$$\tilde{x} = \lambda x + (1 - \lambda)x', \quad \tilde{o} = \lambda o + (1 - \lambda)o', \quad \tilde{\delta} = \begin{cases} \delta & \text{prob} = \lambda \\ \delta' & \text{prob} = 1 - \lambda \end{cases} \tag{4}$$

where we randomly select $\delta$ with probability $\lambda$ and $\delta'$ with probability $(1 - \lambda)$. This can be viewed as a probabilistic interpolation of the event indicator, since it must take the value 0 or 1. Coordinate-wise interpolation of $(o, \delta)$ does not, in general, preserve the structural constraints that define valid survival objects (e.g., monotonicity of $S$) once projected through $T$. Consequently, the induced targets can drift off the manifold of admissible survival outcomes and fail to provide any principled smoothness regularization. Figure 1 shows that both C-Mixup and the naive interpolation fail to regularize the estimated survival probability smoothly. This necessitates the proposal of mixup strategy tailored to the survival analysis.

## 2.2 HAZARD-MIXUP (H-MIXUP)

Instead of naively interpolating the raw data points, we propose a model-agnostic mixup for survival analysis called *H-Mixup* as follows:

$$\tilde{x} = \lambda x + (1 - \lambda)x', \quad (\tilde{o}, \tilde{\delta}) = \begin{cases} (o/\lambda, \delta) & \text{if } o/\lambda \leq o'/(1 - \lambda) \\ (o'/(1 - \lambda), \delta') & \text{otherwise.} \end{cases} \tag{5}$$

It first samples $\lambda \sim \text{Beta}(\alpha, \alpha)$ and takes the usual linear interpolation in the input space, $\tilde{x} = \lambda x + (1 - \lambda)x'$, and then aligns time so that the two parent trajectories are comparable; the mixed outcome $\tilde{o}$ is the first event under these aligned times. Under this construction, the mixed example behaves like its instantaneous risk was the time-aligned convex combination of the parent hazards.

**Theorem 1** (Temporally-Scaled Hazard Linearity of H-Mixup). *Let $(x, o, \delta)$ and $(x', o', \delta')$ be two observed instances. We define the mixed instance $(\tilde{x}, \tilde{o}, \tilde{\delta})$ via H-Mixup construction with $\lambda \in (0, 1)$. Under the assumption of covariate-dependent censoring ($T \perp C|x$), the constructed observable pair $(\tilde{o}, \tilde{\delta})$ is distributionally equivalent to a valid right-censored realization of a latent event time $\tilde{T}$, characterized by the hazard function:*

$$h(t|\tilde{x}) = \lambda h(\lambda t|x) + (1 - \lambda)h((1 - \lambda)t|x'). \tag{6}$$

*This implies that H-Mixup generates instances consistent with the hazard in equation 6. Thus, training a model to minimize NLL on the augmented instances encourages the learned hazard to satisfy equation 6.*

We defer the proof of this theorem to Appendix B.1. The mixup for classification and regression encourages local linearity in model outputs. Analogously, encouraging local linearity $\phi(t|\tilde{x}) \approx \lambda\phi(t|x) + (1-\lambda)\phi(t|x')$ for each $t \geq 0$ is the most straightforward application of mixup to survival analysis. However, this construction treats the observed time $O$ as a static label rather than a duration. By merely averaging the risk intensities $h(t|\tilde{x}) = \lambda h(t|x) + (1 - \lambda)h(t|x')$, this creates a synthetic instance implicitly encouraged to experience the event at both observed times $o$ and $o'$ simultaneously . This results in a multi-modal distribution (e.g., risk spikes around $o$ and $o'$) that creates less clear event probability densities. On the other hand, H-Mixup stretches or shrinks time to align the events. This produces a single, smooth risk curve that looks like a natural intermediate outcome, rather than a confusing mix of two separate events.

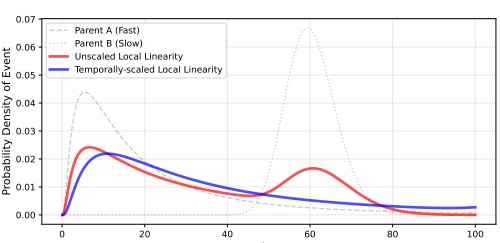

Figure 2: Comparison of interpolation methods showing that unscaled local linearity (red) results in a multi-modal distribution by merely averaging hazards. This approach generates the synthetic instance to experience risk spikes at both parent event times simultaneously, unlike the temporally-scaled approach (blue)

H-Mixup follows the standard mixup recipe (Zhang et al., 2018) of encouraging linear behavior between training examples, but applies it to the hazard function. Theorem **??** shows that instead of naively averaging static labels, H-Mixup encourages the model to learn a temporally-aligned combination of the parent risks. This approach aligns with the regularization goals of previous mixup methods (Zhang et al., 2018; Verma et al., 2019a;b; Carratino et al., 2022), while correctly handling the scalable nature of time in survival data. We further explore alternative survival-specific mixup rules in Appendix C, including variants that interpolate the survival and cumulative hazard functions.

The mixed observed times tend to skew a bit later, but not deterministically. Although $\tilde{o}$ can be numerically larger than the original times, this does not inflate the true follow-up duration. H-Mixup defines a new random variable $\tilde{O}$ under a vicinal distribution $P_v$. Therefore, $\tilde{o}$ is a reparametrization of the time for the mixed samples; $\tilde{o}$ and $o$ are defined in different scales. The apparent inflation is an artifact of this rescaled random variable rather than a change in the underlying event process. Provided the censored likelihood is evaluated under this new vicinal data distribution, the estimator remains consistent. Moreover, since survival models predict normalized quantities rather than raw outcomes, they avoid scale sensitivity.

## 2.3 Event Ratio Adjustment

Like other mixup strategies, H-Mixup changes the event-censoring ratio in the augmented distribution. Specifically, it increases the proportion of $\delta = 1$ when event times are stochastically earlier than censoring times (the typical right-censoring setup with administrative cutoff or late dropout), and decreases it otherwise (e.g., when censoring often occurs very early). Changes in marginal label proportions are known to affect calibration under prior label shift in classification (Guo et al., 2017; Saerens et al., 2002), and absolute-risk calibration in survival is likewise sensitive to event/censor sampling and weighting (Crowson et al., 2016; van Houwelingen & Putter, 2012). Thus, if augmentation perturbs the event fraction away from the population rate $\pi = \mathbb{P}(\delta = 1)$, maximum-likelihood fits may drift in baseline risk unless we correct this shift.

Let $\hat{\pi}$ be the empirical event rate under the mixup-augmented training distribution $\mathbb{P}_{\mathrm{v}}$, and let $\pi_*$ be the desired rate (pre-augmentation minibatch). Define the weights: $w(\tilde{\delta}) = \tilde{\delta}\frac{\pi_*}{\hat{\pi}} + (1 - \tilde{\delta})\frac{1-\pi_*}{1-\hat{\pi}}$, and the weighted negative log-likelihood: $\mathcal{R}_{\mathrm{w}}(\phi) = \mathbb{E}_{\mathbb{P}_{\mathrm{v}}}[w(\tilde{\delta})\ell(\phi; \tilde{o}, \tilde{\delta}, \tilde{x})]$. Assume a label-shift analogue holds:

$$\mathbb{P}_{\mathrm{v}}(\tilde{x}, \tilde{o}|\tilde{\delta}) \equiv \mathbb{P}_*(\tilde{x}, \tilde{o}|\tilde{\delta}), \qquad \mathbb{P}_{\mathrm{v}}(\tilde{\delta}) \neq \mathbb{P}_*(\tilde{\delta}), \tag{7}$$

where $\mathbb{P}_*$ is the target (population) distribution. Then the weights exactly correct for the marginal shift:

$$\mathcal{R}_{\mathrm{w}}(\phi) = \sum_{\delta \in \{0,1\}} \int w(\delta)\ell(\phi; o, \delta, x) \underbrace{\mathbb{P}_{\mathrm{v}}(x, o|\delta)}_{=\mathbb{P}_*(x, o|\delta)} \mathbb{P}_{\mathrm{v}}(\delta)dxdo \tag{8}$$

$$= \sum_{\delta \in \{0,1\}} \int \ell(\phi; o, \delta, x)\mathbb{P}_*(x, o|\delta) \underbrace{w(\delta)\mathbb{P}_{\mathrm{v}}(\delta)}_{=\mathbb{P}_*(\delta)} dxdo = \mathbb{E}_{\mathbb{P}_*}\big[\ell(\phi; o, \delta, x)\big] = \mathcal{R}_*(\phi).$$

Thus, $\mathcal{R}_{\mathrm{w}}(\phi)$ is an unbiased surrogate for the target risk $\mathcal{R}_*(\phi)$ and preserves absolute-risk calibration in expectation. For Cox partial likelihood or discrete-time logistic hazards, the same $w(\tilde{\delta})$ enter as case weights, maintaining consistent baseline risk estimation under biased event/censor sampling (Prentice, 1986; Borgan et al., 2000).

## 3 Theoretical Analysis

We now present a theoretical analysis of H-Mixup. Due to the space limit, we defer preliminaries and details to Appendix B.

**Theorem 2** (Generalization Bound for Survival Mixup). *Let a mixup strategy define a vicinal distribution $\mathbb{P}_{\mathrm{v}}$. Under standard regularity conditions, for any $\eta \in (0, 1)$, with probability at least $1 - \eta$, the following inequality holds for all $\phi \in \mathcal{H}$:*

$$R_{\mathbb{P}}(\phi) \leq \widehat{R}_{n,\mathbb{P}_{\mathrm{v}}}(\phi) + B(\phi) + 2\widehat{\mathfrak{R}}_n(\ell \circ \mathcal{H} \mid S_{\mathrm{v}}) + \sigma_{\mathrm{v}}\sqrt{\frac{2\log(1/\eta)}{n}} \tag{9}$$

*where $B(\phi) = R_{\mathbb{P}}(\phi) - R_{\mathbb{P}_{\mathrm{v}}}(\phi)$ is the vicinal bias specific to the chosen mixup strategy, and $\sigma_{\mathrm{v}}$ is the variance proxy of the loss under $\mathbb{P}_{\mathrm{v}}$.*

This result follows the standard VRM generalization template (Chapelle et al., 2001; Carratino et al., 2022) that works for classification and regression tasks. However, the extension to survival is not an immediate corollary: the sample space involves $(x, o, \delta)$ with censoring, and the losses require survival-specific arguments. We prove in Appendix B.2 that the same decomposition still holds once the vicinal distribution and censoring-aware loss are appropriately defined.

The regularization benefits of vicinal interpolation are established in prior VRM/mixup work (Zhang et al., 2018; Carratino et al., 2022; Chapelle et al., 2001). In brief: (i) for Lipschitz losses, replacing $S$ by $S_{\mathrm{v}}$ contracts local oscillations and shrinks the effective input diameter, yielding a smaller empirical Rademacher term $\widehat{\mathfrak{R}}_n(\ell \circ \mathcal{H}|S_{\mathrm{v}}) \leq q, \widehat{\mathfrak{R}}_n(\ell \circ \mathcal{H}|S)$; (ii) evaluating $\ell$ on mixed samples concentrates more tightly, so the deviation proxy contracts, $\sigma_{\mathrm{v}}^2 \leq \sigma^2$. We treat these as standard consequences of VRM and do not re-derive them here.

The remaining ingredient is the vicinal bias, which does depend on how outputs are mixed. For H-Mixup, we use the mixed covariate $\tilde{x} = \lambda x + (1-\lambda)x'$ and a conservative, deterministic time rule $\tilde{o} = \min(o/\lambda, o'/(1-\lambda))$, with $\tilde{\delta}$ and the loss $\ell(f; \tilde{o}, \tilde{\delta}, \tilde{x})$ specified in Appendix B. Intuitively, after rescaling $o$ and $o'$ by their weight, $\tilde{o}$ selects the earlier effective time, akin to a system that fails when either component fails, so it does not understate risk. The induced bias is small when hazards vary smoothly along the segment between $x$ and $x'$ and when the local time-rescaling $(\lambda, 1-\lambda)$ is a reasonable approximation to the data-generating dynamics. Otherwise, $B_H(\phi)$ can dominate and erode the gains from complexity/variance contraction. We formalize this as Proposition 1, which bounds $B_H(\phi)$ by a feature-shift term, controlled by the Lipschitz smoothness of $f$ in $x$, plus a time-alignment term, controlled by temporal regularity of the hazard under mild rescaling.

**Proposition 1** (Vicinal Bias of H-Mixup). *Let $\mathbb{P}$ be the true distribution and $\mathbb{P}_v$ the vicinal distribution from H-Mixup. Assume: (i) the loss is $L_h$-Lipschitz in the hazard; (ii) $x \mapsto h_\phi(\cdot|x)$ is $L_x$-Lipschitz (as a map into $L^1(\mu)$); (iii) the true hazard $h^*(\cdot|x)$ is $\beta$-Holder in time with constant $C_t$. Then, if $\lambda \sim \text{Beta}(\alpha, \alpha)$,*

$$B_H(\phi) \lesssim L_h L_x \frac{\alpha}{(2\alpha+1)}\mathbb{E}[\|X_i - X_j\|] + L_h C_t \mu([0,\tau])\Big(\frac{\alpha}{2(2\alpha+1)}\Big)^\beta. \tag{10}$$

The proof first rewrites the H-Mixup bias so that it separates into two parts: the difference caused by replacing each parent's features with the mixed features, and the difference caused by rescaling the parent timelines. The feature difference is controlled by the smoothness of the model in the covariates, while the time-rescaling difference is controlled by how smoothly the true hazards vary in time. Taking expectations over the mixing weights then gives the final bound. Again, we defer the rigorous proof to Appendix.

In summary, the bound shows that H-Mixup improves generalization whenever the smoothness induced by the mixup and the resulting concentration of the loss outweigh the approximation error. Its effectiveness is thus governed by a measurable trade-off: smaller $\widehat{\mathfrak{R}}_n(\ell \circ \mathcal{H}|S_v)$ and $\sigma_v$ versus the task-specific $B_H(f)$, whose magnitude reflects how well the mixing aligns with the underlying survival mechanism. While Proposition 1 bounds the hazard error $B_H$, survival metrics (e.g., IBS) depend on the survival function. In Appendix B.4, we prove that the deviation between the predicted survival function $S_\phi$ and the true survival function $S^*$ is bounded by $|S_\phi(t) - S^*(t)| \lesssim S^*(t) \cdot B_H \cdot t$. This implies that minimizing hazard bias suppresses the survival bias, weighted linearly by the time horizon.

This kind of encouraging local linearity, where interpolation in the input (or hidden) space translates into approximately linear outputs, can be understood as an inductive bias that regularizes model behavior between samples. Such an inductive bias has been shown to improve generalization in supervised learning by enforcing smoother decision boundaries and reducing overfitting. For instance, Mixup (Zhang et al., 2018) and its extensions such as Manifold Mixup (Verma et al., 2019a), Between-Class Learning (Tokozume et al., 2018), and CutMix (Yun et al., 2019) directly leverage this principle. Subsequent analyses formalize the bias as locally linear out-of-manifold regularization (Guo et al., 2019a) and investigate its theoretical underpinnings (Carratino et al., 2022; Chaudhry et al., 2022). Structured variants such as PuzzleMix (Kim et al., 2020) further refine this inductive bias by preserving local semantics while maintaining interpolation consistency.

## 4 EXPERIMENTS

We evaluate our mixup strategies across combinations of deep survival models and datasets. We use the Deep Cox proportional hazards model (DeepCox; (Katzman et al., 2018; Kvamme et al., 2019)), which is semi-parametric under the proportional hazards assumption; DeepHit (Lee et al., 2018); a discrete-time model trained with an IPCW-weighted integrated Brier score objective (DeepIBS; (Graf et al., 1999; Gerds & Schumacher, 2006; Han et al., 2021)); and DeepMTLR (Fotso, 2018), which (like DeepHit and DeepIBS) is a non-parametric discrete-time approach. Further architectural and loss details are provided in Appendix D.

For evaluation, we report the time-dependent concordance index ($C^{td}$) (Antolini et al., 2005), the integrated Brier score (IBS) (Gerds & Schumacher, 2006), and the expected calibration error (ECE) (Naeini et al., 2015). $C^{td}$ measures the fraction of comparable pairs whose risks are correctly

Table 1: Comparison of ERM, C-Mixup, and H-Mixup ($\alpha = 0.4$) on the semi-synthetic and real-world survival datasets. The best result among ERM, C-Mixup, and H-Mixup is highlighted in **bold**, and the value in parentheses denotes the standard deviation over the five repetitions.

| | Dataset | DeepCox ERM | DeepCox C-Mixup | DeepCox H-Mixup | DeepHit ERM | DeepHit C-Mixup | DeepHit H-Mixup | DeepIBS ERM | DeepIBS C-Mixup | DeepIBS H-Mixup | DeepMTLR ERM | DeepMTLR C-Mixup | DeepMTLR H-Mixup |
|---|---|---|---|---|---|---|---|---|---|---|---|---|---|
| $C^{td}$ (↑) | MNIST | 0.9140 (0.0028) | **0.9315** (0.0032) | 0.9266 (0.0014) | 0.8479 (0.0077) | 0.8403 (0.0056) | **0.9031** (0.0067) | 0.8874 (0.0049) | 0.9016 (0.0030) | **0.9193** (0.0077) | 0.9087 (0.0065) | 0.9160 (0.0043) | **0.9402** (0.0026) |
| | RetinaMNIST | 0.6990 (0.0190) | 0.7056 (0.0158) | **0.7132** (0.0108) | 0.6804 (0.0188) | 0.6776 (0.0212) | **0.7087** (0.0107) | 0.6928 (0.0101) | 0.6842 (0.0145) | **0.7041** (0.0223) | 0.6999 (0.0110) | 0.6711 (0.0098) | **0.7013** (0.0139) |
| | PathMNIST | 0.9063 (0.0099) | 0.9024 (0.0099) | **0.9068** (0.0040) | 0.8832 (0.0107) | 0.8688 (0.0047) | **0.9112** (0.0112) | 0.8952 (0.0082) | 0.8919 (0.0091) | **0.9116** (0.0070) | 0.9036 (0.0122) | 0.8994 (0.0087) | **0.9339** (0.0032) |
| | OrganMNIST3D | 0.8741 (0.0055) | 0.8707 (0.0099) | **0.8819** (0.0022) | 0.8425 (0.0080) | 0.8501 (0.0065) | **0.8810** (0.0124) | 0.8799 (0.0088) | 0.8768 (0.0093) | **0.8999** (0.0116) | 0.8773 (0.0040) | 0.8826 (0.0100) | **0.9010** (0.0102) |
| | COVID-19-NY-SBU | 0.6177 (0.0558) | 0.6121 (0.0293) | **0.6433** (0.0132) | 0.5798 (0.0433) | 0.5596 (0.0970) | **0.6145** (0.0498) | 0.5708 (0.0429) | 0.5845 (0.0469) | **0.5871** (0.0280) | 0.5532 (0.0612) | 0.5717 (0.0608) | **0.5994** (0.0286) |
| IBS (↓) | MNIST | **0.0176** (0.0015) | 0.0201 (0.0014) | 0.0210 (0.0009) | 0.0197 (0.0011) | 0.0185 (0.0017) | **0.0151** (0.0011) | 0.0159 (0.0011) | 0.0163 (0.0067) | **0.0135** (0.0015) | 0.0163 (0.0015) | 0.0168 (0.0068) | **0.0130** (0.0012) |
| | RetinaMNIST | 0.1777 (0.0124) | 0.1711 (0.0089) | **0.1656** (0.0106) | 0.1961 (0.0165) | 0.2113 (0.0050) | **0.1870** (0.0137) | **0.1865** (0.0070) | 0.2045 (0.0075) | 0.1988 (0.0242) | **0.1879** (0.0102) | 0.2350 (0.0122) | 0.1892 (0.0056) |
| | PathMNIST | 0.0355 (0.0054) | 0.0429 (0.0051) | **0.0317** (0.0026) | 0.0773 (0.0151) | 0.0679 (0.0051) | **0.0585** (0.0059) | 0.0495 (0.0019) | 0.0652 (0.0135) | **0.0382** (0.0049) | 0.0533 (0.0157) | 0.0535 (0.0048) | **0.0332** (0.0037) |
| | OrganMNIST3D | 0.1022 (0.0041) | 0.0999 (0.0089) | **0.0949** (0.0071) | 0.0876 (0.0075) | 0.0839 (0.0055) | **0.0701** (0.0081) | 0.0907 (0.0076) | 0.0907 (0.0089) | **0.0772** (0.0061) | 0.0936 (0.0088) | 0.0944 (0.0075) | **0.0824** (0.0101) |
| | COVID-19-NY-SBU | 0.2065 (0.0228) | 0.2183 (0.0302) | **0.1984** (0.0146) | 0.2500 (0.0310) | 0.2384 (0.0468) | **0.2180** (0.0372) | **0.2995** (0.0611) | 0.3375 (0.0861) | 0.3118 (0.0453) | 0.3546 (0.0908) | 0.3454 (0.0509) | **0.3011** (0.0576) |
| ECE (↓) | MNIST | **0.0206** (0.0011) | 0.0320 (0.0020) | 0.0274 (0.0007) | 0.0288 (0.0017) | 0.0260 (0.0042) | **0.0209** (0.0015) | 0.0213 (0.0019) | 0.0214 (0.0023) | **0.0129** (0.0024) | 0.0203 (0.0014) | 0.0213 (0.0028) | **0.0110** (0.0009) |
| | RetinaMNIST | 0.0785 (0.0200) | **0.0696** (0.0117) | 0.0766 (0.0169) | **0.1171** (0.0126) | 0.1269 (0.0079) | 0.1253 (0.0327) | 0.0848 (0.0221) | 0.1092 (0.0111) | **0.0474** (0.0250) | **0.0685** (0.0096) | 0.0905 (0.0060) | 0.0938 (0.0061) |
| | PathMNIST | 0.0302 (0.0046) | 0.0352 (0.0024) | **0.0288** (0.0017) | 0.0886 (0.0192) | **0.0715** (0.0053) | 0.0750 (0.0106) | 0.0546 (0.0062) | 0.0536 (0.0067) | **0.0458** (0.0057) | 0.0558 (0.0164) | 0.0461 (0.0028) | **0.0357** (0.0042) |
| | OrganMNIST3D | 0.0619 (0.0035) | 0.0590 (0.0080) | **0.0544** (0.0052) | 0.0643 (0.0111) | 0.0642 (0.0069) | **0.0536** (0.0090) | 0.0631 (0.0072) | 0.0625 (0.0084) | **0.0474** (0.0059) | 0.0660 (0.0117) | 0.0672 (0.0061) | **0.0516** (0.0056) |
| | COVID-19-NY-SBU | 0.2459 (0.0636) | 0.2562 (0.0483) | **0.2355** (0.0113) | 0.3939 (0.0507) | 0.3783 (0.0673) | **0.3212** (0.0607) | 0.2776 (0.0391) | 0.2705 (0.0703) | **0.2518** (0.0700) | 0.2882 (0.0557) | 0.2675 (0.0403) | **0.2260** (0.0299) |

ordered within a time horizon. IBS integrates, over the evaluation window, the Brier score at time $t$—the IPCW-weighted squared error between the observed event status and the predicted survival probability. ECE summarizes the average absolute gap between observed and expected survival, conditional on model-estimated risk. Higher $C^{td}$ indicates better discrimination, while lower IBS and ECE indicate better accuracy and calibration, respectively.

## 4.1 SEMI-SYNTHETIC IMAGE DATASETS

Following the survival data generation protocol of (Goldstein et al., 2020), we simulated survival datasets on MNIST (LeCun et al., 2002), OrganMNIST3D, PathMNIST, and RetinaMNIST (Yang et al., 2021; 2023). We defer dataset details and the generation process to Appendix E. For MNIST and OrganMNIST3D, we train 6-layer CNNs on $28\times28$ images and $28\times28\times28$ 3D volumes, respectively. For PathMNIST and RetinaMNIST, we train ResNet-18 on $224\times224$ images.

In Table 1, we compare ERM and the proposed H-Mixup strategy on both simulated and real-world datasets. Here, we set $\alpha$ to 0.4, a value commonly used in previous mixup studies (Zhang et al., 2018). As shown in Table 1, in most cases on simulated datasets, H-Mixup improves discrimination, accuracy, and calibration of the model, achieving higher $C^{td}$ and lower IBS and ECE than ERM. Performance gains are particularly clear on MNIST, OrganMNIST3D, PathMNIST, and RetinaMNIST. In these datasets, the models trained with H-Mixup achieved superior results compared to ERM in terms of $C^{td}$. Focusing on medical datasets, such as OrganMNIST3D, PathMNIST, and RetinaMNIST, H-mixup delivers consistent benefits, improving the average performance by 2.24% in $C^{td}$, 12.81% in IBS, and 4.21% in ECE, respectively. These results indicate that hazard-space interpolation is an effective and model-agnostic regularizer that enhances both discrimination and calibration ability of the model under censored datasets. Consequently, we can expect that the H-Mixup can be plugged into various models to benefit time-to-event modeling tasks in the medical domain and other areas, where strong generalization and robust handling of censoring are essential.

## 4.2 REAL-WORLD DATA: COVID-19-NY-SBU

To evaluate the effectiveness of H-Mixup in the real-world setting, we conducted experiments on the COVID-19-NY-SBU dataset Saltz et al. (2021). This dataset consists of 1,384 COVID-19–positive patients, each paired with both a chest X-ray image and survival outcomes, with a censoring ratio of 87.7%. Following Matsumoto et al. (2023), we resized each image to $256\times256$ resolution, and utilized the image closest to the admission time for each patient. We employed DenseNet-121 as the backbone and trained both ERM and H-Mixup for 100 epochs.

Table 2: Comparison of H-Mixup ($\alpha = 0.4$) applied on the raw inputs and hidden representations using the MNIST dataset. To interpolate representations, we utilized outputs from each of the Conv layers or the FC layers. The best performance in each column is highlighted in **bold** and the value in parentheses denotes the standard deviation over the five repetitions.

| | DeepCox | | | DeepHit | | | DeepIBS | | | DeepMTLR | | |
|---|---|---|---|---|---|---|---|---|---|---|---|---|
| | $C^{td}$ (↑) | IBS (↓) | ECE (↓) | $C^{td}$ (↑) | IBS (↓) | ECE (↓) | $C^{td}$ (↑) | IBS (↓) | ECE (↓) | $C^{td}$ (↑) | IBS (↓) | ECE (↓) |
| Input | 0.9266 (0.0014) | **0.0210** (0.0009) | 0.0274 (0.0007) | 0.9031 (0.0067) | 0.0151 (0.0011) | 0.0209 (0.0015) | 0.9193 (0.0077) | 0.0135 (0.0015) | 0.0129 (0.0024) | 0.9402 (0.0026) | **0.0130** (0.0012) | **0.0110** (0.0009) |
| Conv 1 | 0.9265 (0.0019) | 0.0214 (0.0012) | 0.0277 (0.0009) | 0.9029 (0.0080) | 0.0153 (0.0009) | 0.0211 (0.0016) | 0.9163 (0.0095) | **0.0129** (0.0014) | **0.0121** (0.0015) | 0.9360 (0.0044) | 0.0132 (0.0012) | **0.0110** (0.0011) |
| Conv 2 | 0.9258 (0.0016) | 0.0221 (0.0016) | 0.0268 (0.0012) | 0.9056 (0.0077) | **0.0146** (0.0007) | 0.0202 (0.0012) | 0.9186 (0.0094) | 0.0136 (0.0008) | 0.0134 (0.0011) | 0.9374 (0.0028) | 0.0141 (0.0012) | 0.0122 (0.0018) |
| Conv 3 | 0.9263 (0.0014) | 0.0221 (0.0015) | **0.0264** (0.0006) | 0.9001 (0.0133) | **0.0146** (0.0010) | **0.0193** (0.0017) | 0.9196 (0.0084) | 0.0138 (0.0013) | 0.0140 (0.0019) | 0.9407 (0.0027) | 0.0150 (0.0014) | 0.0131 (0.0013) |
| FC 1 | 0.9272 (0.0020) | 0.0217 (0.0010) | 0.0291 (0.0019) | 0.9223 (0.0042) | 0.0147 (0.0011) | 0.0224 (0.0014) | 0.9238 (0.0087) | 0.0161 (0.0043) | 0.0226 (0.0020) | **0.9464** (0.0034) | 0.0172 (0.0012) | 0.0223 (0.0018) |
| FC 2 | **0.9281** (0.0036) | 0.0238 (0.0009) | 0.0390 (0.0026) | **0.9442** (0.0047) | 0.0150 (0.0010) | 0.0232 (0.0020) | **0.9543** (0.0041) | 0.0162 (0.0011) | 0.0241 (0.0033) | 0.9447 (0.0028) | 0.0183 (0.0016) | 0.0272 (0.0031) |

As shown in Table 1, H-Mixup consistently outperforms the ERM baselines on the COVID-19 dataset in most cases and achieves improvements in discrimination, accuracy, and calibration. On average, H-Mixup achieves a 5.3% C-index improvement, along with a 7.9% improvement in IBS and a 16.5% improvement in ECE over ERM. These results show that H-Mixup is effective not only in semi-synthetic settings but also on real-world clinical images where noise, heterogeneity, and extensive censoring make survival prediction particularly challenging. The results also indicate that hazard-space interpolation serves as a practical, model-agnostic regularizer under realistic conditions, confirming its applicability to medical survival prediction.

## 4.3 MANIFOLD MIXUP

H-Mixup is not restricted to the input space and can also be applied to hidden representations as in previous works (Zhang et al., 2018; Verma et al., 2019a). Here, we compare interpolating raw inputs with interpolating hidden representations using the MNIST dataset. We use a neural network that consists of three convolutional (Conv) layers followed by three fully connected (FC) layers. To interpolate representations, we utilized the output obtained from each of the Conv or FC layers, excluding the output of the last FC layer, which produces the final predictions. For clarity, we denote the three Conv layers as Conv 1 to 3 and the next two FC layers as FC 1 and 2. The larger indices indicate deeper layers.

As shown in Table 2, the effective $C^{td}$ can be obtained when interpolating representations at deeper layers, which implies that interpolating hidden representations improves discrimination. However, calibration degrades at deeper layers, as reflected by higher IBS and ECE. Meanwhile, interpolation at the input space yields IBS and ECE similar to those obtained by interpolating at the shallow layer, while its $C^{td}$ is comparable to mixing at mid-depth (e.g., Conv 3). Choosing where to interpolate is therefore nontrivial. Nevertheless, we observed that interpolation in representation space outperforms ERM in most cases and can be used to enhance the performance of deep survival models.

## 4.4 SENSITIVITY ANALYSIS

In Figure 3, we analyze the sensitivity of the hyperparameter $\alpha \in \{0.1, 0.2, 0.3, 0.4\}$ using the DeepMTLR. In H-Mixup, the mixing weight $\lambda$ is sampled from $\text{Beta}(\alpha, \alpha)$. Smaller $\alpha$ pushes $\lambda$ toward 0 or 1, whereas larger $\alpha$ pulls $\lambda$ toward 0.5. As shown in Figure 3, the performance of DeepMTLR is relatively robust across different $\alpha$ on MNIST and OrganMNIST3D for $C^{td}$, IBS, and ECE. On PathMNIST, however, evaluation metrics exhibit some variations with different $\alpha$. In this case, DeepMTLR achieves superior performance as $\alpha$ increases to 0.4.

## 4.5 TABULAR DATA

To assess mixup on non-image data, we conducted experiments on three standard survival datasets: FLChain (Dispenzieri et al., 2012; Therneau, 2025), Framingham–CVD (D'Agostino et al., 2008), and METABRIC (Curtis et al., 2012). Models in this section are fully connected networks with two hidden layers (64 and 32 units) and GELU activations (Hendrycks & Gimpel, 2020). We trained all models with AdamW (Loshchilov & Hutter, 2019) with learning rate $5 \times 10^{-4}$ and weight decay $10^{-6}$. Across all three datasets, H-Mixup delivered performance that was consistently better or

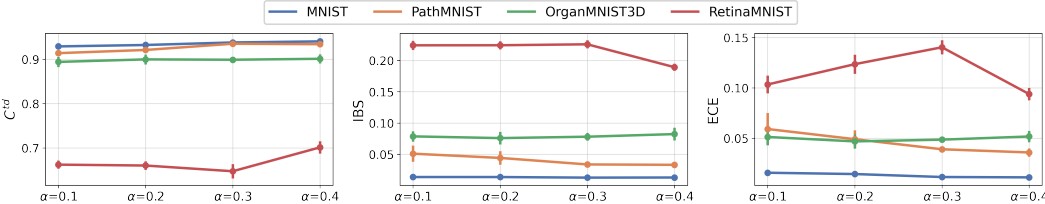

Figure 3: Sensitivity analysis on $\alpha \in \{0.1, 0.2, 0.3, 0.4\}$ for the DeepMTLR model on the MNIST, PathMNIST, OrganMNIST3D and RetinaMNIST datasets. Vertical bars are for standard deviations.

comparable to the corresponding ERM baselines, with gains varying by dataset and model family. Detailed results are presented in Table 9 in the appendix.

## 5 RELATED WORK

Mixup stems from VRM, which improves ERM by training on augmented samples drawn from local vicinities of the data (Chapelle et al., 2001). The original Mixup constructs convex combinations of two inputs and their labels, boosting accuracy and calibration in large-scale classification (Zhang et al., 2018). Follow-ups move mixing into hidden layers to smooth representations (Verma et al., 2019a) or splice image regions with mixed labels (CutMix) (Yun et al., 2019). Theory views mixup as an inductive bias that smooths predictions along lines connecting training points—akin to label smoothing plus Jacobian/derivative regularization, which curbs overconfidence and improves calibration and robustness (Carratino et al., 2022; Zou et al., 2023). In practice, this simple bias has helped across adversarial robustness, noisy-label learning, semi-supervised learning, medical imaging, NLP, and time-series forecasting (Sun et al., 2020; Guo et al., 2019b; Thulasidasan et al., 2019; Kim et al., 2022; Smucny et al., 2022; Nguyen et al., 2023; Demirel & Holz, 2023). Yet existing mixup work largely targets conventional supervised tasks (classification/regression). Survival analysis is different: supervision is distributed over time-to-event outcomes with censoring, and models are optimized in hazard, cumulative-hazard, or survival-probability space. A mixing rule that is harmless in one representation may be incoherent in another, so a direct transplant of mixup is not principled. To date, no studies have adapted VRM/mixup to censored survival in a model-agnostic way or developed a theory for such adaptations. This is the methodological gap we addressed.

## 6 CONCLUSION

We presented H-Mixup, a hazard-space formulation of mixup for survival analysis. By interpolating hazards rather than observed outcomes, H-Mixup directly aligns augmentation with the survival likelihood and introduces a Bregman-type regularizer that enforces local linearity along interpolations. This provides the first principled reconciliation of mixup with censored data. Experiments across diverse tabular and image-based benchmarks show that H-Mixup consistently improves concordance, integrated Brier scores, and calibration, where standard survival models often struggle. Beyond empirical gains, our framework demonstrates that vicinal risk smoothing can be naturally extended to time-to-event data, highlighting new design principles for augmentation under censoring. The model-agnostic nature of H-Mixup ensures compatibility with Cox models, full-likelihood formulations, and discrete-time hazards, making it broadly useful across survival modeling paradigms.

Looking forward, several exciting directions emerge. Our current work assumes non-informative censoring, and extending hazard-space mixup to informative censoring models could open the door to more realistic clinical applications. Richer interpolation schemes such as CutMix or adaptive vicinal distributions may further enhance robustness by capturing more complex local structure in survival data. Perhaps most compellingly, applying survival mixup to sequential and multi-modal longitudinal settings, such as electronic health records, wearable streams, or video-based clinical monitoring, offers the potential to bridge structured augmentation with the complexity of real-world data. Together, these directions position mixup not only as a regularization tool, but as a general framework for learning stable and calibrated survival models in increasingly dynamic settings.

REPRODUCIBILITY STATEMENT

We have taken extensive measures to ensure the reproducibility of our results. All datasets used in this work are publicly available, and can be automatically downloaded by executing the provided code, or are directly included when permitted by licensing terms. Our experimental design, model architectures, training procedures, and evaluation protocols are described in detail in the Appendix. In addition, we attach the full implementation, including data preprocessing scripts, training pipelines, and evaluation routines, to allow for exact replication of our results.

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

# A    DETAILS OF THE TOY EXAMPLE

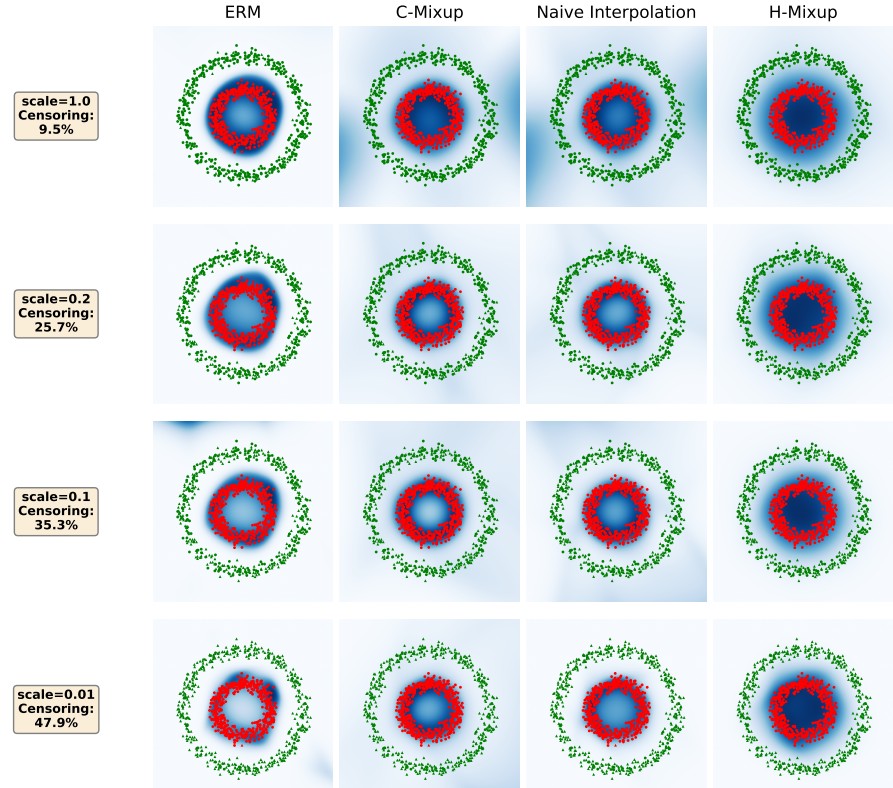

Figure 4: This illustrates performance across varying levels of data availability and censoring, ranging from a scale of 1.0 (9.5% censoring) down to a scale of 0.01 (47.9% censoring). Red and green dots denote high- and low-risk instances, respectively, while blue shading indicates predicted survival probability (lighter corresponds to larger probability) at median follow-up. The results confirm that ERM, C-Mixup (Yao et al., 2022) and naive interpolation (equation 4) spuriously assign low risk to the center of the high-risk cluster, pushing risk to the boundaries. Furthermore, while others are sensitive to censoring, H-Mixup (equation 5) is consistent: it yields the most regularized decision surface and preserves high central risk. This robustness arises because C-Mixup and naive interpolation average follow-up times without a coherent vicinal distribution, whereas H-Mixup achieves the desired smoothing in covariate-time space.

We constructed a two-dimensional synthetic survival dataset to visualize the effect of different mixup strategies. The data consist of two concentric rings of points in $\mathbb{R}^2$: an outer ring ("low risk") and an inner ring ("high risk"). The covariates $X \in \mathbb{R}^2$ were generated by sampling angles uniformly from $[0, 2\pi]$ and radii from normal distributions centered at $1.0$ (outer) and $0.45$ (inner) with standard deviation $0.06$. Each observation was labeled by group membership.

The event times $T$ were sampled from an exponential distribution with hazard

$$h(x) = 0.25 \exp(f_{\text{true}}),$$

where $f_{\text{true}} = 3$ for the low-risk outer ring and $f_{\text{true}} = 10$ for the high-risk inner ring. Independent censoring times $C$ were drawn from an exponential distribution with scale 0.2, and the observed outcomes were $O = \min(T, C)$ and $\Delta = \mathbb{I}(T \leq C)$.

A simple feed-forward neural network with two hidden layers of width 64 and GELU activations was trained under three regimes: (i) **ERM**: empirical risk minimization without augmentation; (ii) **Naive interpolation**: feature-space interpolation; (iii) **H-Mixup**: our proposed hazard-space interpolation. All models were trained for 1000 epochs with batch size 64 using the DeepCox objective and Adam

optimizer (lr $= 10^{-3}$). For the naive interpolation and H-Mixup, we set $\alpha = 0.4$ for the Beta distribution.

To visualize learned decision surfaces, we computed predicted survival probabilities $S(t|x)$ over a $300 \times 300$ grid covering $[-1.5, 1.5]^2$, evaluated at the median observed event time. For display, we plotted $1 - S(t|x)$ as a shaded background (higher values $\Rightarrow$ higher predicted risk), with the training points overlaid (green = low risk, red = high risk). Figure 1 shows the three resulting panels, demonstrating that ERM severely overfits, naive interpolation partially smooths but fails to enforce consistent survival estimates, and H-Mixup yields the most regularized and interpretable decision surface.

Executing `toyexample.py` will reproduce the figure.

# B  PROOFS AND FURTHER DISCUSSIONS

In this section, we provide more explicit and rigorous statements of the theorems presented in the main text and their proofs.

## B.1  TEMPORALLY-SCALED HAZARD LINEARITY OF H-MIXUP

The proof establishes that the mixed observation $(\tilde{o}, \tilde{\delta})$ generated by H-Mixup is distributionally equivalent to a valid right-censored observation of a latent event time $\tilde{T}$, and that the hazard function of $\tilde{T}$ satisfies the required linearity condition.

We first define the mixed latent event time as $\tilde{T} = \min(T/\lambda, T'/(1-\lambda))$. Since the parent samples are drawn independently, $T$ and $T'$ are independent conditional on their covariates $(x, x')$.

The survival function $\tilde{S}(t|x, x') = \mathbb{P}(\tilde{T} > t \mid x, x')$ is obtained by multiplying the individual scaled survival functions:

$$\tilde{S}(t|x, x') = \mathbb{P}\left(\frac{T}{\lambda} > t \mid x\right) \cdot \mathbb{P}\left(\frac{T'}{1-\lambda} > t \mid x'\right) = S(\lambda t \mid x) \cdot S((1-\lambda)t \mid x') \quad (11)$$

The hazard function $h_{\tilde{T}}(t|x, x')$ is derived directly from the survival function using the definition $h(t) = -\frac{d}{dt} \log S(t)$:

$$h_{\tilde{T}}(t \mid x, x') = -\frac{d}{dt} \log \left[S(\lambda t \mid x) \cdot S((1-\lambda)t \mid x')\right] \quad (12)$$

Applying the logarithm property for products and the chain rule for derivatives, the hazard for the first term is:

$$-\frac{d}{dt} \log S(\lambda t \mid x) = \lambda h(\lambda t \mid x) \quad (13)$$

Similarly, the hazard for the second term is $(1-\lambda)h((1-\lambda)t \mid x')$. Summing these yields:

$$h_{\tilde{T}}(t \mid x, x') = \lambda h(\lambda t \mid x) + (1-\lambda)h((1-\lambda)t \mid x') \quad (14)$$

When writing $h(t \mid \tilde{x})$ for the hazard of $\tilde{T}$ at the mixed covariate $\tilde{x} = \lambda x + (1-\lambda)x'$, the expression is precisely equation 6, confirming that $\tilde{T}$ satisfies the temporally-scaled linearity property.

Now we formally show that the H-Mixup output $(\tilde{o}, \tilde{\delta})$ is a standard right-censored observation of $\tilde{T}$ under non-informative censoring.

We define the mixed latent event time $\tilde{T} = \min(T/\lambda, T'/(1-\lambda))$ and the mixed censoring time $\tilde{C} = \min(C/\lambda, C'/(1-\lambda))$. By expanding the definitions $o = \min(T, C)$ and $o' = \min(T', C')$ and using the stability of the minimum operator under positive scaling and regrouping, we find:

$$\tilde{o} = \min\left(\frac{\min(T, C)}{\lambda}, \frac{\min(T', C')}{1-\lambda}\right) = \min\left(\min\left(\frac{T}{\lambda}, \frac{T'}{1-\lambda}\right), \min\left(\frac{C}{\lambda}, \frac{C'}{1-\lambda}\right)\right) \quad (15)$$

$$\tilde{o} = \min(\tilde{T}, \tilde{C}) \tag{16}$$

The mixed censoring indicator $\tilde{\delta} = 1$ if and only if $\tilde{T} \leq \tilde{C}$. This confirms that $\tilde{o}$ is a right-censored observation of the mixed event time $\tilde{T}$.

The initial assumption of covariate-dependent censoring on the parents ($T \perp C \mid x$ and $T' \perp C' \mid x'$), combined with the independence between the two sampled individuals, implies the joint independence:

$$(T, T') \perp (C, C') \mid (x, x') \tag{17}$$

Since $\tilde{T}$ is a function of $(T, T')$ and $\tilde{C}$ is a function of $(C, C')$, the mixed event and censoring times are independent conditional on the parents:

$$\tilde{T} \perp \tilde{C} \mid (x, x') \tag{18}$$

This guarantees that the censoring mechanism is non-informative for the mixed sample. Under non-informative censoring, the event-time hazard of $\tilde{T}$ is identifiable from $(\tilde{o}, \tilde{\delta})$ and equals the latent marginal hazard function (Kalbfleisch & Prentice, 2002). Combining this identifiability with the linearity of $h_{\tilde{T}}(t \mid x, x')$ derived in Step 1 shows that, for each H-Mixup sample generated from parents $(x, x')$, the underlying hazard satisfies the temporally scaled linearity in equation 6. Training a model hazard $h_\theta(t)|\tilde{x}$ on such mixed samples therefore encourages this property.

## B.2 GENERALIZATION BOUND

We start by establishing a generalization bound for survival analysis models.

**Lemma 1** (Generalization Bound for sub-Gaussian Losses). *Let $\mathcal{H}$ be a class of functions, let $\ell : \mathcal{H} \times \mathcal{Z} \to \mathbb{R}_+$ be a loss function, and let $S = (Z_1, \ldots, Z_n)$ be a set of $n$ samples drawn i.i.d. from a distribution $\mathbb{P}$ on $\mathcal{Z}$. We assume that for every $\phi \in \mathcal{H}$, the centered loss variable $X_\phi = \ell(\phi; Z) - R(\phi)$ is sub-Gaussian with a uniform variance proxy $\sigma^2 > 0$. That is, for all $\phi \in \mathcal{H}$:*

$$\mathbb{E}_{Z \sim \mathbb{P}} \left[ \exp(\lambda X_\phi) \right] \leq \exp \left( \frac{\lambda^2 \sigma^2}{2} \right) \quad \text{for all } \lambda \in \mathbb{R}. \tag{19}$$

*Then, for any $\eta \in (0, 1)$, with probability at least $1 - \eta$ over the draw of the sample $S \sim \mathbb{P}^n$, the following inequality holds uniformly for all $\phi \in \mathcal{H}$:*

$$R(\phi) \leq \widehat{R}_n(\phi) + 2\widehat{\mathfrak{R}}_n(\ell \circ \mathcal{H}) + \sigma \sqrt{\frac{2 \log(1/\eta)}{n}}, \tag{20}$$

*where $R(\phi) = \mathbb{E}_{Z \sim \mathbb{P}}[\ell(\phi; Z)]$ is the true risk, $\widehat{R}_n(\phi) = \frac{1}{n} \sum_{i=1}^{n} \ell(\phi; Z_i)$ is the empirical risk, and $\widehat{\mathfrak{R}}_n(\ell \circ \mathcal{H})$ is the empirical Rademacher complexity of the loss class, defined as:*

$$\widehat{\mathfrak{R}}_n(\ell \circ \mathcal{H}) = \mathbb{E}_{\boldsymbol{\epsilon}} \left[ \sup_{\phi \in \mathcal{H}} \frac{1}{n} \sum_{i=1}^{n} \epsilon_i \ell(\phi; Z_i) \Big| S \right] \tag{21}$$

*with $\boldsymbol{\epsilon} = (\epsilon_1, \ldots, \epsilon_n)$ being a vector of independent Rademacher random variables.*

Lemma 1 is a direct corollary of the Rademacher complexity bounds of Bartlett & Mendelson (2002), as streamlined in Mohri et al. (2018). Our formulation replaces boundedness by the more general sub-Gaussian condition, yielding the same $O(\sqrt{\log(1/\eta)/n})$ concentration while retaining a data-dependent empirical complexity term. This uniform control is the cornerstone of our subsequent analysis: when we introduce mixup, we will show that the empirical Rademacher complexity of the induced loss class shrinks by a multiplicative factor, leading to provably tighter generalization guarantees. In this sense, Lemma 1 serves as the baseline bound upon which our mixup-based improvements are quantified.

In much of statistical learning theory, boundedness of the loss is the standard assumption, enabling sharp concentration via Hoeffding-type inequalities (Bartlett & Mendelson, 2002; Mohri et al., 2018). Yet this requirement can be restrictive, as it excludes many natural loss functions that are unbounded but still well-behaved (including negative log-likelihoods in survival models).

The sub-Gaussian condition is a natural relaxation: it guarantees exponential tail decay sufficient for Rademacher-based concentration while subsuming both bounded and many unbounded light-tailed losses. By adopting this assumption, Lemma 1 recovers the classical $O(\sqrt{\log(1/\eta)/n})$ rate, but in a form that is applicable to the unbounded loss functions that arise in survival analysis.

By the contraction principle, any $L$-Lipschitz loss satisfies $\widehat{\mathfrak{R}}_n(\ell \circ \mathcal{F}) \lesssim L\widehat{\mathfrak{R}}_n(\mathcal{F})$ (Bartlett & Mendelson, 2002). This guarantees that standard generalization bounds apply directly to survival analysis, provided the loss is Lipschitz. While many widely used objectives—including discrete-time log-likelihoods (e.g., MTLR, DeepHit), IPCW Brier-type losses, and parametric hazard negative log-likelihoods—involve exponential terms that are not globally Lipschitz, they satisfy the property on compact domains. In practice, IPCW stabilization and truncation further reduce the effective variance proxy $\sigma$ (Graf et al., 1999; van der Laan & Robins, 2003), while pairwise or ranking-based risks (e.g., Cox surrogates) admit analogous control through U-statistic symmetrization (Clémençon et al., 2008). Specifically, under practical constraints where model outputs are bounded (e.g., via weight decay or clipping), we can formally bound the Lipschitz constant.

**Lemma 2** (Lipschitz Continuity of Survival NLL). *Let $\mathcal{H}$ be a hypothesis class of functions $\phi :$ $\mathcal{X} \times [0, \tau] \to \mathbb{R}$ representing the log-hazard, such that for all $\phi \in \mathcal{H}$, $\|\phi\|_\infty \leq M$. The negative log-likelihood loss is defined as:*

$$\ell(\phi; x, o, \delta) = -\delta\phi(x, o) + \int_0^o \exp(\phi(x, t))dt \tag{22}$$

*is $L_\ell$-Lipschitz with respect to $\phi$ in the $L_\infty$ norm, with the Lipschitz constant $L_\ell = 1 + \tau e^M$.*

*Proof.* We aim to show that for any $\phi, \psi \in \mathcal{H}$ and data point $z = (x, o, \delta)$, the loss satisfies $|\ell(\phi; z) - \ell(\psi; z)| \leq L_\ell \|\phi - \psi\|_\infty$. We consider the three standard cases for survival objectives:

*Continuous Hazard:* The loss is $\ell(\phi; z) = -\delta\phi(x, o) + \int_0^o \exp(\phi(x, t))\, dt$. The functional derivative with respect to $\phi$ at time $s$ is:

$$\nabla_\phi \ell(\phi) = -\delta \cdot \mathbb{I}(s = o) + \exp(\phi(x, s)) \cdot \mathbb{I}(s \leq o). \tag{23}$$

Bounding the $L_1$ norm of the gradient:

$$\|\nabla_\phi \ell\|_1 \leq |-\delta| + \int_0^o |\exp(\phi(x, t))|dt \leq 1 + \int_0^\tau e^M dt = 1 + \tau e^M. \tag{24}$$

*Discrete Hazard:* For models partitioning time into $K$ bins, the integral is replaced by a sum (or Softmax). The loss typically takes the form of Cross-Entropy, $\ell(\mathbf{z}) = -\log(\frac{\exp(z_c)}{\sum \exp(z_j)})$. The gradient of the Softmax-Cross-Entropy with respect to logits $\phi$ is bounded by $\|\mathbf{p} - \mathbf{y}\|_1 \leq 2$. Thus, the discrete case is Lipschitz with a constant related to the number of bins or simply bounded by 2 (depending on norm choice), which is subsumed by the general bound.

*Ranking Loss and Partial Likelihood:* These objectives involve interactions between samples (pairs or risk sets). For pairwise ranking (e.g., DeepHit's ranking loss), the loss is a sum of convex surrogates $\ell(z_i, z_j) = \sigma(\phi(x_i) - \phi(x_j))$ (e.g., logistic or hinge). Since the derivative of the surrogate $\sigma'$ is bounded (e.g., $|\sigma'(u)| \leq 0.25$ for logistic), the loss is Lipschitz with respect to the model outputs. Similarly, for Cox Partial Likelihood, the loss for subject $i$ involves a log-sum-exp over the risk set: $\ell_i = \log \sum_{j \in \mathcal{R}_i} \exp(\phi(x_j)) - \phi(x_i)$. The gradient with respect to the vector of outputs behaves identically to the Softmax case (Case 2), with the $L_1$ norm of the gradient bounded by 2. In all three cases, assuming bounded model outputs, the loss function admits a finite Lipschitz constant. $\square$

These observations imply that the baseline generalization bound is not only theoretically valid but also practically meaningful for survival analysis, where loss functions can otherwise exhibit high variance or pairwise dependence. Lemma 2 anchors our analysis by demonstrating that standard survival objectives fit into the established generalization framework (Vapnik, 1998; Shalev-Shwartz & Ben-David, 2014; Bartlett & Mendelson, 2002; Graf et al., 1999; van der Laan & Robins, 2003; Clémençon et al., 2008). Moreover, it highlights that improvements in variance control or Lipschitz

continuity, such as those induced by mixup, translate directly into tighter complexity terms and sharper guarantees.

Now we present the proof for Theorem 2.

*Proof of Theorem 2.* Fix a mixup rule that induces a vicinal distribution $\mathbb{P}_v$ on the input-output space $\mathcal{Z}$. For any hypothesis $\phi \in \mathcal{H}$, we can decompose the true risk $R_{\mathbb{P}}(\phi)$ as:

$$R_{\mathbb{P}}(\phi) = R_{\mathbb{P}_v}(\phi) + \underbrace{\big(R_{\mathbb{P}}(\phi) - R_{\mathbb{P}_v}(\phi)\big)}_{=B(\phi)}. \tag{25}$$

The term $B(\phi)$ represents the systematic bias introduced by evaluating the model on the vicinal distribution instead of the true distribution. It thus suffices to upper bound the vicinal risk $R_{\mathbb{P}_v}(\phi)$ uniformly over $\mathcal{H}$.

We proceed by applying the standard Rademacher generalization bound (Lemma 1) to the vicinal distribution. We assume the construction of the empirical vicinal dataset $S_v = (\tilde{z}_1, \ldots, \tilde{z}_n)$ involves $n$ i.i.d. draws from $\mathbb{P}_v$ (practically achieved by sampling indices with replacement and applying the mixup map). Furthermore, based on Lemma 2, the survival loss is bounded and Lipschitz on the compact domain defined by the mixup. This implies that the centered loss random variable $X_\phi^{(v)} = \ell(\phi; Z) - R_{\mathbb{P}_v}(\phi)$ is sub-Gaussian with some variance proxy $\sigma_v^2$.

Applying standard uniform convergence results (Bartlett & Mendelson, 2002) to the sample $S_v \sim \mathbb{P}_v^n$, with probability at least $1 - \eta$:

$$R_{\mathbb{P}_v}(\phi) \leq \widehat{R}_{n,\mathbb{P}_v}(\phi) + 2\widehat{\mathfrak{R}}_n\big(\ell \circ \mathcal{H} \mid S_v\big) + \sigma_v \sqrt{\frac{2\log(1/\eta)}{n}}. \tag{26}$$

Substituting this bound back into the decomposition of $R_{\mathbb{P}}(\phi)$ yields:

$$R_{\mathbb{P}}(\phi) \leq \widehat{R}_{n,\mathbb{P}_v}(\phi) + B(\phi) + 2\widehat{\mathfrak{R}}_n\big(\ell \circ \mathcal{H} \mid S_v\big) + \sigma_v \sqrt{\frac{2\log(1/\eta)}{n}}. \tag{27}$$

This concludes the proof. $\qquad\square$

### B.3 Vicinal Bias of H-Mixup

Write the H-Mixup input as $\tilde{x} = \lambda x + (1 - \lambda)x'$ and denote $m = \min\{\lambda, 1 - \lambda\} \in [0, \frac{1}{2}]$. Let $\tilde{y}$ be the H-Mixup target in hazard space (the specific construction only enters via the bounds below). Define the vicinal bias as the discrepancy between the mixed target and the true conditional hazard at the mixed input, measured inside the loss:

$$B_H(\phi) = \mathbb{E}\big[\ell\big(h_\phi(\cdot|\tilde{x}), \tilde{y}\big) - \ell\big(h_\phi(\cdot|\tilde{x}), h^*(\cdot|\tilde{x})\big)\big].$$

By $L_h$-Lipschitzness of $\ell$ in its hazard argument and Jensen's inequality,

$$B_H(\phi) \leq L_h \mathbb{E}\Big[\big\|\tilde{y} - h^*(\cdot|\tilde{x})\big\|_{L^1(\mu)}\Big]. \tag{28}$$

Add and subtract $\lambda h^*(\cdot|x) + (1 - \lambda)h^*(\cdot|x')$ to split equation 28 into a *spatial* and a *temporal* term:

$$\big\|\tilde{y} - h^*(\cdot|\tilde{x})\big\|_{L^1(\mu)} \leq \underbrace{\big\|\lambda h^*(\cdot|x) + (1 - \lambda)h^*(\cdot|x') - h^*(\cdot|\tilde{x})\big\|_{L^1(\mu)}}_{\text{spatial term}} \tag{29}$$

$$+ \underbrace{\big\|\tilde{y} - \big(\lambda h^*(\cdot|x) + (1 - \lambda)h^*(\cdot|x')\big)\big\|_{L^1(\mu)}}_{\text{temporal term}}.$$

By convexity and triangle inequality,

$$\big\|\lambda h^*(\cdot|x) + (1 - \lambda)h^*(\cdot|x') - h^*(\cdot|\tilde{x})\big\|_{L^1(\mu)} \tag{30}$$

$$\leq \lambda\big\|h^*(\cdot|x) - h^*(\cdot|\tilde{x})\big\|_{L^1(\mu)} + (1 - \lambda)\big\|h^*(\cdot|x') - h^*(\cdot|\tilde{x})\big\|_{L^1(\mu)}$$

$$\leq \lambda L_x\|x - \tilde{x}\| + (1 - \lambda)L_x\|x' - \tilde{x}\| = L_x\big(\lambda(1 - \lambda) + (1 - \lambda)\lambda\big)\|x - x'\|$$

$$= 2L_x\lambda(1 - \lambda)\|x - x'\|.$$

Taking expectations over $\lambda \sim \text{Beta}(\alpha, \alpha)$ and using the variance identity $\mathbb{E}[\lambda(1-\lambda)] = \frac{\alpha}{2(2\alpha+1)}$ yields

$$\mathbb{E}[\text{spatial term}] \leq L_x \mathbb{E}[2\lambda(1-\lambda)\|X_i - X_j\|] = 2L_x \frac{\alpha}{2(2\alpha+1)} \mathbb{E}[\|X_i - X_j\|] = L_x \frac{\alpha}{2\alpha+1} \mathbb{E}[\|X_i - X_j\|]. \tag{31}$$

H-Mixup perturbs time locally (e.g., via a small rescaling/warping controlled by $m$). By the assumed $\beta$-Holder regularity in time,

$$\left\|\tilde{y} - (\lambda h^*(\cdot|x) + (1-\lambda)h^*(\cdot|x'))\right\|_{L^1(\mu)} \leq C_t m^\beta \mu([0, \tau]). \tag{32}$$

Taking expectations and using that $u \mapsto u^\beta$ is concave on $[0, 1]$ for $\beta \in (0, 1]$ (so by Jensen's $\mathbb{E}[m^\beta] \leq \mathbb{E}[m]^\beta$), and bounding $\mathbb{E}[m]$ via the variance identity:

$$\mathbb{E}[\text{temporal term}] \leq C_t \mu([0, \tau]) \left(\frac{\alpha}{2\alpha+1}\right)^\beta. \tag{33}$$

Plugging equation 31 and equation 33 into equation 28 gives

$$B_H(\phi) \leq L_h \left(L_x \frac{\alpha}{2\alpha+1} \mathbb{E}\|X_i - X_j\| + C_t \mu([0, \tau]) \left(\frac{\alpha}{2\alpha+1}\right)^\beta\right), \tag{34}$$

which implies the stated bound (up to absolute constants hidden in $\lesssim$).

### B.4 EFFECT OF H-MIXUP ON THE SURVIVAL FUNCTION BIAS

**Proposition 2** (Propagation of Vicinal Bias to Survival Function). *Let $S_\phi(t|\tilde{x})$ and $S^*(t|\tilde{x})$ be the survival functions corresponding to the H-Mixup hazard $h_\phi(\cdot|\tilde{x})$ and the true hazard $h^*(\cdot|\tilde{x})$, respectively. If the hazard vicinal bias is uniformly bounded such that $|h_\phi(s|\tilde{x}) - h^*(s|\tilde{x})| \leq B_H$ for all $s \in [0, t]$, then the survival function bias is bounded by:*

$$|S_\phi(t|\tilde{x}) - S^*(t|\tilde{x})| \leq S^*(t|\tilde{x}) \left(\exp(B_H \cdot t) - 1\right). \tag{35}$$

*For small $B_H \cdot t$, this implies a linear error growth: $|S_\phi(t|\tilde{x}) - S^*(t|\tilde{x})| \lesssim S^*(t|\tilde{x}) \cdot B_H \cdot t$.*

*Proof.* Recall the relationship between the survival function and the hazard function: $S(t|x) = \exp\left(-\int_0^t h(s|x)ds\right)$. Let $\Delta(t) = \int_0^t (h_\phi(s|\tilde{x}) - h^*(s|\tilde{x}))ds$ be the cumulative difference in hazards. The difference in survival probabilities is:

$$|S_\phi(t|\tilde{x}) - S^*(t|\tilde{x})| = \left|\exp\left(-\int_0^t h_\phi(s|\tilde{x})ds\right) - \exp\left(-\int_0^t h^*(s|\tilde{x})ds\right)\right|$$

$$= \exp\left(-\int_0^t h^*(s|\tilde{x})ds\right) \cdot |\exp(-\Delta(t)) - 1|$$

$$= S^*(t|\tilde{x}) \cdot |\exp(-\Delta(t)) - 1|.$$

Using the uniform bound on the hazard bias, we have $|\Delta(t)| \leq \int_0^t |h_\phi(s|\tilde{x}) - h^*(s|\tilde{x})|ds \leq B_H \cdot t$. Since the function $f(z) = |e^z - 1|$ is increasing in $|z|$, we can bound the term by substituting the maximum magnitude of $\Delta(t)$:

$$|S_\phi(t|\tilde{x}) - S^*(t|\tilde{x})| \leq S^*(t|\tilde{x}) \left(\exp(B_H \cdot t) - 1\right).$$

Using the first-order Taylor expansion $e^x \approx 1 + x$ for small $x$, the error scales as $S^*(t|\tilde{x}) \cdot B_H \cdot t$, effectively weighting the instantaneous hazard bias by the horizon $t$ and the survival probability. $\square$

The theoretical implications of the vicinal bias extend naturally beyond continuous hazard estimation to models that parameterize the probability mass function (PMF) directly, such as DeepHit (Lee et al., 2018) or MTLR (Fotso, 2018). While H-Mixup is motivated by the linearity of the conditional hazard, these discrete-time methods implicitly model a discrete hazard rate $h_d(t) = P(T = t|T \geq t)$. Consequently, H-Mixup acts as a regularizer that encourages this implicit hazard to interpolate linearly between samples, effectively smoothing the underlying risk manifold. The vicinal bias $B_H$

therefore captures the discrepancy between this imposed linearity and the true data distribution, regardless of whether the model architecture outputs a continuous rate or a discrete vector.

Proposition 2 serves as the bridge between these formulations. Since the survival function is uniquely determined by the cumulative hazard, the derived bound on $|S_\phi(t|\tilde{x}) - S^*(t|\tilde{x})|$ provides a universal guarantee for distributional fidelity. For a model like DeepHit, which optimizes a likelihood or ranking objective based on the PMF, the survival function is simply the complement of the cumulative mass. Thus, the error bound on $S(t)$ strictly constrains the cumulative deviation of the predicted PMF from the true distribution. Furthermore, because standard discrete loss functions (such as cross-entropy or pairwise ranking) satisfy the necessary Lipschitz conditions on compact domains, the fundamental trade-off, i.e., reducing Rademacher complexity via augmentation at the cost of introducing vicinal bias, remains valid. In this view, $B_H$ should be interpreted not merely as an estimation error specific to hazard models, but as a measure of the manifold error introduced by the mixing strategy, limiting the achievable accuracy of any consistent estimator trained on the augmented data.

## C  OTHER MIXUP STRATEGIES FOR SURVIVAL ANALYSIS

### C.1  S-MIXUP

**S-Mixup** interpolates the input vector only, and randomly chooses one out of the pair of $(o_i, \delta_i)$.

$$\tilde{x} = \lambda x + (1 - \lambda)x', \qquad (\tilde{o}, \tilde{\delta}) = \begin{cases} (o, \delta) & \text{with prob. } \lambda \\ (o', \delta') & \text{with prob. } 1 - \lambda \end{cases} \tag{36}$$

This approach encourages local linearity in the survival function $S(t|x)$, that is, $S(t|\lambda x + (1 - \lambda)x') \approx \lambda S(t|x) + (1 - \lambda)S(t|x')$ for all $t \geq 0$. Training at $\tilde{x}$ minimizes a convex combination of risks from $x_i$ and $x_j$. For proper survival-probability losses, convexity implies the Bayes solution is the convex combination of the two optima. For hazard negative log-likelihoods, solving the first-order condition yields the formula above, which is always between the two endpoints. Hence S-Mixup enforces local linearity or smoothing of $S(\cdot|x)$.

**Proposition 3** (S-Mixup Implies Survival Linearity). *Let $S(t|x)$ and $S(t|x')$ be the true conditional survival functions for* **x** *and* **x**'*, respectively. Then the true conditional survival function for the mixed covariate* $\tilde{\mathbf{x}}$*, denoted $S(t|\tilde{x})$, satisfies*

$$S(t|\tilde{x}) = \lambda S(t|x) + (1 - \lambda)S(t|x').$$

*Proof.* By the law of total probability,

$$\begin{aligned} S(t|\tilde{x}) &= \mathbb{P}(\tilde{T} > t|\text{outcome from } \mathbf{x})\mathbb{P}(\text{outcome from } \mathbf{x}) \\ &\quad + \mathbb{P}(\tilde{T} > t|\text{outcome from } \mathbf{x}')\mathbb{P}(\text{outcome from } \mathbf{x}') \\ &= \mathbb{P}(T > t|\mathbf{x}) \cdot \lambda + \mathbb{P}(T' > t|\mathbf{x}') \cdot (1 - \lambda) \\ &= \lambda S(t|x) + (1 - \lambda)S(t|x'). \end{aligned}$$

$\square$

**Proposition 4** (Vicinal Bias of S-Mixup). *Let $\ell(f(x); o, \delta)$ be the training loss. Assume:*

*(i) $\ell(\cdot; o, \delta)$ is $L_{\text{pred}}$-Lipschitz in the prediction (e.g., in $L^1(\mu)$ over time);*

*(ii) $f$ is $L_x$-Lipschitz from inputs to predictions (same norm as in (i));*

*(iii) $\lambda \sim \text{Beta}(\alpha, \alpha)$, independently of $(Z_i, Z_j)$.*

*Define the S-Mixup vicinal risk and bias*

$$R_{\text{v}}^S(f|\tilde{X}, Z_i, Z_j) = \lambda \, \ell\big(f(\tilde{X}); O_i, \Delta_i\big) + (1 - \lambda) \, \ell\big(f(\tilde{X}); O_j, \Delta_j\big),$$

$$B_S(f) = \mathbb{E}_{Z_i, Z_j, \lambda}\Big[\lambda\big(\ell_i(X_i) - \ell_i(\tilde{X})\big) + (1 - \lambda)\big(\ell_j(X_j) - \ell_j(\tilde{X})\big)\Big], \quad \ell_k(X) = \ell\big(f(X); O_k, \Delta_k\big).$$

*Then*

$$B_S(f) \ \leq \ L_{\text{pred}} L_x \, \frac{\alpha}{2\alpha + 1} \, \mathbb{E}\big[\|X_i - X_j\|\big].$$

*Proof.* By (i)–(ii),

$$\left|\ell_i(X_i) - \ell_i(\tilde{X})\right| \leq L_{\text{pred}} \left\|f(X_i) - f(\tilde{X})\right\| \leq L_{\text{pred}} L_x \|X_i - \tilde{X}\| = L_{\text{pred}} L_x (1-\lambda) \|X_i - X_j\|,$$

and symmetrically

$$\left|\ell_j(X_j) - \ell_j(\tilde{X})\right| \leq L_{\text{pred}} L_x \lambda \|X_i - X_j\|.$$

Therefore, using the nonnegativity of the weights,

$$B_S(f) \leq \mathbb{E}[\lambda \cdot L_{\text{pred}} L_x (1-\lambda) \|X_i - X_j\| + (1-\lambda) \cdot L_{\text{pred}} L_x \lambda \|X_i - X_j\|]$$
$$= 2 L_{\text{pred}} L_x \, \mathbb{E}[\lambda(1-\lambda)] \, \mathbb{E}[\|X_i - X_j\|] \quad \text{(independence of } \lambda \text{ and } (X_i, X_j)\text{)}.$$

For $\lambda \sim \text{Beta}(\alpha, \alpha)$, $\mathbb{E}[\lambda(1-\lambda)] = \frac{\alpha}{2(2\alpha+1)}$, hence

$$B_S(f) \leq 2 L_{\text{pred}} L_x \cdot \frac{\alpha}{2(2\alpha+1)} \, \mathbb{E}\|X_i - X_j\| = L_{\text{pred}} L_x \frac{\alpha}{2\alpha+1} \, \mathbb{E}\|X_i - X_j\|.$$

$\square$

### C.2 CH-MIXUP

**CH-Mixup** linearly interpolates the input vector, takes a weighted harmonic mean of the observed time, and randomly selects the event indicator.

$$\tilde{x} = \lambda x_i + (1-\lambda) x_j, \quad \tilde{o} = (\lambda o_i^{-1} + (1-\lambda) o_j^{-1})^{-1} \quad \tilde{\delta} = \begin{cases} \delta_i & \text{with prob. } \lambda \\ \delta_j & \text{with prob. } 1-\lambda \end{cases} \tag{37}$$

This strategy encourages the local linearity in the cumulative hazard function $H(t|x)$. This is different from the local linearity in $S$. Since the cumulative hazard function is $H(t|o, \delta) \approx \delta \cdot \mathbb{I}\{t \leq o\}/o$ under the exponential assumption, taking a harmonic mean of times corresponds to linear interpolation of the cumulative hazard. Hence CH-Mixup encourages *local linearity in the cumulative hazard function* $H(t|x)$, in contrast to S-Mixup which enforces local linearity in the survival function $S(t|x)$.

**Theorem 3** (CH-Mixup Approximates Cumulative Hazard Linearity). *Under exponential surrogates, i.e., $T \sim \text{Exp}(h)$ and $T' \sim \text{Exp}(h')$ with $h \approx 1/o$ and $h' \approx 1/o'$, the conditional cumulative hazard for the mixed covariate, $h(t|\tilde{x})$, is approximately a convex combination of the parent cumulative hazards:*

$$h(t|\tilde{x}) \approx \lambda H(t|x) + (1-\lambda) H(t|x').$$

*Proof sketch under exponential surrogates.* Assume $T \sim \text{Exp}(h)$ and $T' \sim \text{Exp}(h')$, so $H(t|x) = ht$ and $H_{x'}(t) = h't$.

*(1) Hazard-rate linearity.* By definition of $\tilde{o}$,

$$\frac{1}{\tilde{o}} = \frac{\lambda}{o} + \frac{1-\lambda}{o'}.$$

Using the exponential approximation $h \approx 1/o$ and $h' \approx 1/o'$, we obtain

$$h_{\tilde{x}} \approx \frac{1}{\tilde{o}} = \lambda h + (1-\lambda) h'.$$

*(2) Cumulative hazard linearity.* Since $h(t|\tilde{x}) = h_{\tilde{x}} t$, it follows that

$$h(t|\tilde{x}) \approx \big(\lambda h + (1-\lambda) h'\big) t = \lambda(ht) + (1-\lambda)(h't) = \lambda H(t|x) + (1-\lambda) H(t|x').$$

$\square$

## C.3  NAIVE INTERPOLATION

Given two samples $(x, o, \delta)$ and $(x', o', \delta')$ and a mixing weight $\lambda \in (0, 1)$, the naive interpolation rule produces

$$\tilde{x} = \lambda x + (1 - \lambda)x', \qquad \tilde{o} = \lambda o + (1 - \lambda)o', \qquad \tilde{\delta} = \begin{cases} \delta & \text{prob} = \lambda, \\ \delta' & \text{prob} = 1 - \lambda. \end{cases}$$

At the population level, the naive interpolation rule does not imply linearity of any of the standard survival representations: survival $S$, hazard $h$, or cumulative hazard $H$. Its primary effect is feature-output smoothing rather than enforcing a specific survival relationship.

For clarity, first ignore censoring so that the observed time equals the latent event time: $o = T$ and $o' = T'$. Under Naive Interpolation, the mixed time is the convex combination

$$\tilde{T} = \lambda T + (1 - \lambda)T'.$$

Let $f_T$ and $f_{T'}$ denote the conditional densities of $T$ and $T'$ given $x$ and $x'$, respectively. Assuming $T$ and $T'$ are independent, the density of the mixed time $\tilde{T}$ is given by a scaled convolution:

$$f_{\tilde{T}}(t) = \int_{-\infty}^{\infty} \frac{1}{1 - \lambda} f_T(u) f_{T'}\left(\frac{t - \lambda u}{1 - \lambda}\right) du,$$

from which the mixed survival follows as

$$S(t|\tilde{x}) = \mathbb{P}(\tilde{T} > t|\tilde{x}) = \int_t^{\infty} f_{\tilde{T}}(u)du.$$

In general, this convolution does *not* reduce to the convex combination $\lambda S_x(t) + (1 - \lambda)S_{x'}(t)$, nor does it match a multiplicative (log-linear/additive hazard) relation that would yield linearity in $H$ or $h$. Hence,

$$S(t|\tilde{x}) \neq \lambda S(t|x) + (1 - \lambda)S(t|x') \quad \text{in general,}$$

and analogous non-linearity holds for $H$ and $h$. With censoring reintroduced, $\tilde{o}$ is still a convex combination of observed times and therefore does not restore linearity in $(S, H, h)$; it further detaches the synthetic target from a coherent event-time model. Consequently, Naive Interpolation lacks the theoretical interpretability enjoyed by strategies that impose structure in a specific survival representation.

Naive Interpolation chiefly injects noise by smoothing both the inputs $\tilde{x}$ and the scalar time output $\tilde{o}$. It leverages generic regularization benefits of input interpolation but does not enforce a consistent survival relationship in $S$, $H$, or $h$.

## C.4  DISCUSSION

The consistent outperformance of H-Mixup can be attributed to a fundamental principle: it functions as a more direct and potent regularizer by operating on the hazard function, $h(t)$, which represents the most granular, instantaneous measure of risk. The core survival quantities are derived in a hierarchical manner, where the cumulative hazard, $H(t)$, is the integral of the hazard, and the survival function, $S(t)$, is an exponential transform of the cumulative hazard.

$$H(t) = \int_0^t h(s)ds \quad \text{and} \quad S(t) = \exp(-H(t))$$

Regularization applied directly to $h(t)$, as in H-Mixup, propagates through this hierarchy, ensuring that smoothness in the hazard function leads to smoothness in the derived cumulative hazard and survival functions. This relationship, however, is not symmetric. Regularizing the integrated quantities, as CH-Mixup and S-Mixup do, provides a substantially weaker constraint on the underlying hazard function.

The limitation of indirect regularization stems from the inherent smoothing property of the integral operator. Two markedly different hazard functions can produce nearly identical cumulative hazard functions. To illustrate, consider a smooth, well-behaved hazard function $h_1(t)$ and a volatile counterpart $h_2(t)$ containing a high-frequency component:

$$h_2(t) = h_1(t) + \epsilon \sin(\omega t)$$

Here, the term $\epsilon \sin(\omega t)$ can be interpreted as overfitting to high-frequency noise in the training data. An effective regularizer should penalize such behavior. H-Mixup, operating directly on the hazard, would correctly identify the significant difference between $h_1(t)$ and $h_2(t)$ and penalize the unstable solution.

In contrast, a regularizer targeting the cumulative hazard is far less sensitive. The cumulative hazard corresponding to $h_2(t)$ is:

$$H_2(t) = \int_0^t h_2(s)ds = \int_0^t h_1(s)ds + \int_0^t \epsilon \sin(\omega s)ds = H_1(t) - \frac{\epsilon}{\omega}[\cos(\omega s)]_0^t$$

For a large frequency $\omega$, the magnitude of the difference between $H_1(t)$ and $H_2(t)$ diminishes and is bounded by $2\epsilon/\omega$. Consequently, CH-Mixup would perceive the two cumulative functions as nearly identical and would fail to sufficiently penalize the undesirable volatility in $h_2(t)$. This is because it is sensitive to the smoothness of the integrated risk profile but largely insensitive to the smoothness of the integrand itself. S-Mixup, being one transform further removed, is susceptible to the same weakness.

In summary, H-Mixup imposes a stronger and more localized constraint on the model's output. By directly penalizing erratic or high-frequency fluctuations in the hazard rate, it discourages overfitting to spurious patterns in the training data. CH-Mixup and S-Mixup impose a weaker, non-local constraint that can inadvertently permit such overfitting. Although this represents an empirically observed advantage that may not hold universally for all tasks, it provides a compelling rationale for H-Mixup's superior generalization, which manifests itself as improved calibration and predictive accuracy in practice.

# D   DETAILS ON DEEP SURVIVAL ANALYSIS MODELS

## D.1   DEEPCOX

Let $\{(x_i, o_i, \delta_i)\}_{i=1}^n$ denote covariates $x_i \in \mathbb{R}^p$, observed time $o_i$, and event indicator $\delta_i \in \{0, 1\}$. DeepCox is a continuous-time proportional hazards model in which a neural network $f_\theta : \mathbb{R}^p \to \mathbb{R}$ produces a log-risk score and the individual hazard is

$$\lambda_\theta(t|x) = \lambda_0(t) \exp\big(f_\theta(x)\big),$$

with unspecified baseline hazard $\lambda_0(t)$. Write $r_\theta(x) = \exp(f_\theta(x))$.

Let $\{t_{(1)} < \cdots < t_{(J)}\}$ be the distinct event times, $d_j$ the number of events at $t_{(j)}$, and $R_j = \{i : o_i \geq t_{(j)}\}$ the risk set just before $t_{(j)}$. The Cox partial likelihood is

$$L_{\mathrm{PL}}(\theta) = \mathbb{P}od_{j=1}^J \frac{\mathbb{P}od_{i \in D_j} r_\theta(x_i)}{\left(\sum_{\ell \in R_j} r_\theta(x_\ell)\right)^{d_j}},$$

where $D_j = \{i : o_i = t_{(j)}, \delta_i = 1\}$ is the set of failures at $t_{(j)}$. (With ties, one may use Efron's approximation by replacing the denominator with the Efron correction.) The training objective is the negative log partial likelihood (optionally with weight decay or other regularizers):

$$\ell_{\mathrm{PL}}(\theta) = -\sum_{j=1}^J \left[ \sum_{i \in D_j} f_\theta(x_i) - d_j \log\Big( \sum_{\ell \in R_j} r_\theta(x_\ell) \Big) \right].$$

After fitting, the cumulative baseline hazard can be estimated by Breslow's estimator

$$\widehat{H}_0(t) = \sum_{j:t_{(j)} \leq t} \frac{d_j}{\sum_{\ell \in R_j} r_\theta(x_\ell)},$$

yielding individual survival predictions

$$\widehat{S}_\theta(t|x) = \exp\Big( -\widehat{H}_0(t) r_\theta(x) \Big).$$

### D.2 DEEPHIT

Let the observed data consist of $n$ i.i.d. samples $\{(x_i, o_i, \delta_i)\}_{i=1}^n$, where $x_i \in \mathbb{R}^p$ are covariates, $o_i \in \{t_1, \ldots, t_T\}$ is the observed time discretized on a fixed grid, and $\delta_i \in \{0, 1\}$ is the event indicator ($\delta_i = 1$ if the event occurs, $\delta_i = 0$ if censored). DeepHit parameterizes the conditional distribution of event times through a neural network $f_\theta$ that outputs a probability vector

$$p_\theta(x) = \big(p_\theta(x; t_1), \ldots, p_\theta(x; t_T)\big), \tag{38}$$

with $p_\theta(x; t_j) \geq 0$ and $\sum_{j=1}^T p_\theta(x; t_j) = 1$. Here, $p_\theta(x; t_j)$ represents the probability that an individual with covariates $x$ experiences the event exactly at discrete time $t_j$.

The likelihood contribution for subject $i$ is

$$L_i(\theta) = \begin{cases} p_\theta(x_i; o_i), & \delta_i = 1, \\ \sum_{j:t_j > o_i} p_\theta(x_i; t_j), & \delta_i = 0, \end{cases} \tag{39}$$

so that the overall negative log-likelihood is

$$\ell_{\text{NLL}}(\theta) = -\sum_{i=1}^n \left\{ \delta_i \log p_\theta(x_i; o_i) + (1 - \delta_i) \log \sum_{j:t_j > o_i} p_\theta(x_i; t_j) \right\}. \tag{40}$$

DeepHit augments this likelihood with a pairwise ranking loss $\ell_{\text{rank}}(\theta)$, which encourages the predicted distributions to order subjects consistently with their observed survival times. The final training objective is a convex combination,

$$\ell(\theta) = \ell_{\text{NLL}}(\theta) + \alpha \ell_{\text{rank}}(\theta),$$

where $\alpha > 0$ controls the trade-off.

Because DeepHit models the full event-time distribution rather than a hazard or survival function, it can flexibly capture non-proportional hazards and time-varying effects. In competing-risks settings, the network outputs event-specific probability vectors $p_\theta^k(x)$ for each cause $k$, ensuring identifiability by enforcing normalization across times and causes.

### D.3 DEEPIBS

Let the observed data be $\{(x_i, o_i, \delta_i)\}_{i=1}^n$ with covariates $x_i \in \mathbb{R}^p$, observed time $o_i$, and event indicator $\delta_i \in \{0, 1\}$. Fix a grid $0 = t_0 < t_1 < \cdots < t_T$ and write $I_k = (t_{k-1}, t_k]$. DeepIBS is a discrete-time survival model trained to *directly minimize* the inverse-probability-of-censoring weighted (IPCW) integrated Brier score (IBS).

A neural network $f_\theta$ outputs $T$ real-valued logits $a_\theta^{(1)}(x), \ldots, a_\theta^{(T)}(x)$, which define discrete-time hazards via a sigmoid:

$$h_\theta(t_k | x) = \sigma\big(a_\theta^{(k)}(x)\big) \in (0, 1), \qquad k = 1, \ldots, T.$$

The survival function at grid times is then

$$S_\theta(t_k | x) = \mathbb{P}od_{r=1}^k \big(1 - h_\theta(t_r | x)\big), \qquad S_\theta(t_0 | x) = 1,$$

which is automatically nonincreasing in $k$.

For time $t_j$, define the binary outcome $Y_i(t_j) = \mathbb{I}\{o_i > t_j\}$ and the IPCW weight $w_i(t_j)$ based on an estimate $\widehat{G}$ of the censoring survival function (e.g., Kaplan–Meier fit to $\{(o_i, 1 - \delta_i)\}$):

$$w_i(t_j) = \frac{\mathbb{I}\{o_i \leq t_j, \delta_i = 1\}}{\widehat{G}(o_i^-)} + \frac{\mathbb{I}\{o_i > t_j\}}{\widehat{G}(t_j)}.$$

The (empirical) IPCW Brier score at $t_j$ is

$$\widehat{\text{BS}}_\theta(t_j) = \frac{1}{n} \sum_{i=1}^n w_i(t_j) \big\{Y_i(t_j) - S_\theta(t_j | x_i)\big\}^2.$$

With nonnegative integration weights $\{\nu_j\}_{j=1}^T$ (e.g., $\nu_j \mathbb{P} opt ot_j - t_{j-1}$ and $\sum_j \nu_j = 1$) over a window $[t_{j_1}, t_{j_2}]$, DeepIBS minimizes

$$\mathcal{L}_{\text{IBS}}(\theta) = \sum_{j=j_1}^{j_2} \nu_j \widehat{\text{BS}}_\theta(t_j) = \sum_{j=j_1}^{j_2} \frac{\nu_j}{n} \sum_{i=1}^{n} w_i(t_j) \{Y_i(t_j) - S_\theta(t_j|x_i)\}^2.$$

### D.4 DEEPMTLR

Let the observed data consist of $n$ i.i.d.samples $\{(x_i, o_i, \delta_i)\}_{i=1}^n$, where $x_i \in \mathbb{R}^p$ are covariates, $o_i$ is the observed time, and $\delta_i \in \{0,1\}$ is the event indicator ($\delta_i = 1$ if the event occurs, $\delta_i = 0$ if censored). Fix a nondecreasing grid $0 := t_0 < t_1 < \cdots < t_T$ and define half-open intervals $I_k = (t_{k-1}, t_k]$ for $k = 1, \ldots, T$, with a tail interval $I_{T+1} = (t_T, \infty)$. For a time $o_i$, let $k(i)$ be the unique index such that $o_i \in I_{k(i)}$.

DeepMTLR parameterizes a categorical distribution over the $(T+1)$ intervals via a neural network $f_\theta$ that produces $T$ real-valued logits

$$z_\theta(x) = \left(z_\theta^{(1)}(x), \ldots, z_\theta^{(T)}(x)\right).$$

These are converted to *cumulative* scores

$$s_\theta^{(k)}(x) = \sum_{j=k}^{T} z_\theta^{(j)}(x) \quad (k = 1, \ldots, T), \qquad s_\theta^{(T+1)}(x) = 0,$$

and normalized with a softmax to obtain interval probabilities

$$p_\theta(x; I_k) = \frac{\exp\left(s_\theta^{(k)}(x)\right)}{\sum_{r=1}^{T+1} \exp\left(s_\theta^{(r)}(x)\right)}, \qquad k = 1, \ldots, T+1. \tag{41}$$

This "reverse cumulative-softmax" construction guarantees $p_\theta(x; I_k) \geq 0$ and $\sum_{k=1}^{T+1} p_\theta(x; I_k) = 1$, while implicitly enforcing a nonincreasing survival curve.

From these probabilities, define the discrete survival function and hazard:

$$S_\theta(t_k|x) = \sum_{r>k} p_\theta(x; I_r), \qquad h_\theta(t_k|x) = \frac{p_\theta(x; I_k)}{\sum_{r \geq k} p_\theta(x; I_r)}.$$

The individual contribution for subject $i$ is

$$L_i(\theta) = \begin{cases} p_\theta\left(x_i; I_{k(i)}\right), & \delta_i = 1, \\ S_\theta\left(t_{k(i)}|x_i\right) = \displaystyle\sum_{r>k(i)} p_\theta(x_i; I_r), & \delta_i = 0, \end{cases} \tag{42}$$

leading to the negative log-likelihood

$$\ell_{\text{NLL}}(\theta) = -\sum_{i=1}^{n} \left\{\delta_i \log p_\theta\left(x_i; I_{k(i)}\right) + (1-\delta_i) \log S_\theta\left(t_{k(i)}|x_i\right)\right\}. \tag{43}$$

In practice, DeepMTLR is often trained with $\ell_{\text{NLL}}(\theta)$ alone; time-smoothness penalties on adjacent logits (e.g., $\sum_{j=1}^{T-1} \|z_\theta^{(j+1)}(x) - z_\theta^{(j)}(x)\|_2^2$ averaged over the batch) can stabilize estimates across neighboring bins.

Because DeepMTLR models a full interval distribution via the cumulative-softmax parameterization, it yields a valid monotone survival function and accommodates non-proportional hazards and time-varying effects. It can also be interpreted as a discrete-time likelihood with piecewise-constant hazards induced by $\{p_\theta(x; I_k)\}_{k=1}^{T+1}$.

For competing risks with causes $c = 1, \ldots, C$, output cause-specific interval distributions $\{p_\theta^c(x; I_r)\}_{r=1}^{T+1}$ and define

$$F_\theta^c(t_j|x) = \sum_{r \leq j} p_\theta^c(x; I_r),$$

while ensuring identifiability by sharing the normalization across times and causes, i.e., $\sum_{c=1}^{C} \sum_{r=1}^{T+1} p_\theta^c(x; I_r) = 1$.

# E  DATASETS

## E.1  MNIST-SURVIVAL

**Class-to-Risk Mapping.**  We followed Goldstein et al. (2020) to simulate survival analysis data using MNIST. The foundation of the survival task is a deterministic mapping from a high-level semantic feature of the covariate, the digit's class label $y_i \in \{0, \ldots, 9\}$, to a baseline risk parameter. This parameter, denoted $r(c)$, represents the target mean of the true failure time distribution for all instances of class $c$. This procedure simulates a scenario where distinct patient subgroups (identified here by digit class) have inherently different prognoses. The specific mean-time parameters $r(c)$ are adopted from the X-CAL framework and are defined as follows:

$$
\begin{aligned}
r(0) &= 11.25 & r(5) &= 8.00 \\
r(1) &= 2.25 & r(6) &= 2.00 \\
r(2) &= 5.25 & r(7) &= 11.00 \\
r(3) &= 5.00 & r(8) &= 1.75 \\
r(4) &= 4.75 & r(9) &= 10.75
\end{aligned}
$$

For any given image $x_i$ with its corresponding label $y_i$, the mean of its failure time distribution, $\mu_i$, is therefore set to $\mu_i = r(y_i)$. The task for a predictive model is to learn this mapping implicitly by extracting the relevant visual features from the raw pixel data of $x_i$.

**Event Time Model.**  The true, uncensored event time $T_i$ for each sample is drawn from a Gamma distribution. The choice of the Gamma distribution is motivated by its flexibility in modeling positive-valued random variables and its ability to induce non-proportional hazards through careful parameterization. Conditional on the class label $y_i$, the distribution for $T_i$ is parameterized to have the class-dependent mean $\mu_i = r(y_i)$ and a fixed, global variance $\sigma^2 > 0$. For this benchmark, we fix $\sigma^2 = 10^{-3}$ to ensure that the generated failure times are tightly concentrated around their respective class means, making the underlying risk structure strong and learnable. The distribution is specified as:

$$
T_i | y_i \sim \mathrm{Gamma}(k_i, \theta_i)
$$

Using the shape-scale $(k, \theta)$ parameterization of the Gamma distribution, where the probability density function is $f(t; k, \theta) = t^{k-1} e^{-t/\theta} / (\Gamma(k) \theta^k)$, the mean is $\mathbb{E}[T_i] = k_i \theta_i$ and the variance is $\mathrm{Var}(T_i) = k_i \theta_i^2$. By matching these moments to our target values, we solve for the shape and scale parameters for each instance $i$:

$$
k_i = \frac{\mu_i^2}{\sigma^2} \qquad \text{and} \qquad \theta_i = \frac{\sigma^2}{\mu_i}
$$

This parameterization is the crucial mechanism for generating non-proportional hazards. The hazard function for a Gamma-distributed variable, $h_i(t) = f(t; k_i, \theta_i) / S_i(t)$, where $S_i(t)$ is the survival function, is critically dependent on the shape parameter $k_i$. Since $\sigma^2$ is fixed globally, $k_i$ is directly proportional to the square of the mean, $\mu_i^2$. As $\mu_i$ varies across digit classes, so does $k_i$. Consequently, the shape of the hazard rate curve changes for each class, meaning the ratio of hazards between any two classes, $h_i(t)/h_j(t)$ for $y_i \neq y_j$, is not constant over time. This explicitly violates the proportional hazards assumption and provides a robust testbed for models designed to handle more complex time-to-event dynamics.

**Censoring Model.**  To simulate incomplete observations, which are characteristic of survival datasets, we introduce a right-censoring mechanism. The censoring times $C_i$ are generated from a process that is **non-informative**, meaning the censoring time for an individual is independent of their underlying prognostic factors and true failure time. This is a standard assumption that simplifies modeling. Furthermore, the censoring distribution is applied independently within the training and test splits to ensure comparable censoring rates across the sets. For a given data split $\mathcal{I} \in \{\text{train}, \text{test}\}$, the censoring times are drawn from a uniform distribution whose bounds are determined empirically from the true failure times $T_j$ within that same split:

$$
C_i | \mathcal{I} \sim \mathrm{Unif}\left(a_{\mathcal{I}}, b_{\mathcal{I}}\right)
$$

The lower bound is the minimum observed failure time in the split, $a_{\mathcal{I}} = \min_{j \in \mathcal{I}} T_j$, and the upper bound is the 90th percentile of the failure times in the split, $b_{\mathcal{I}} = Q_{0.9}(\{T_j\}_{j \in \mathcal{I}})$. This data-driven approach pragmatically ensures a moderate and stable rate of censoring (empirically yielding about 45-50% censored instances in each split) without being completely random or trivial.

**Observed Data and Task Formulation.** The final dataset provided for model training and evaluation consists of tuples $(x_i, o_i, \delta_i)$, where $x_i$ is the $1 \times 28 \times 28$ MNIST image with pixel values scaled to the $[0, 1]$ range. The observed outcomes are defined as:

- **Observed Time ($o_i$):** The time of the event or censoring, whichever occurs first. Formally, $o_i = \min(t_i, c_i)$.

- **Event Indicator ($\delta_i$):** A binary flag indicating whether the observed time corresponds to a true failure event ($\delta_i = 1$) or censoring ($\delta_i = 0$). Formally, $\delta_i = \mathbb{I}\{t_i \leq c_i\}$.

The ground-truth information, including the true digit labels $y_i$ and the uncensored failure times $t_i$, are not provided to models.

### E.2 MedMNIST-Survival

Survival analysis is widely used in biomedicine, where medical imaging is a common data modality. To assess the applicability of our approach in this setting, we evaluate on three datasets from MedMNIST (Yang et al., 2021; 2023): PathMNIST (histopathology whole-slide patches), RetinaMNIST (fundus photographs), and OrganMNIST3D (abdominal CT volumes). Because MedMNIST does not provide time-to-event, we generate pseudo time-to-event targets using the same protocol as in our MNIST experiments: class-specific risk scores are specified first, event times are sampled from class-conditional distributions that preserve the desired risk ordering, and censoring times are drawn independently to achieve a prespecified censoring rate.

For the MedMNIST datasets, we avoid arbitrary class risks. Instead, we assign risks based on clinically motivated severity and prognosis associated with each class (e.g., expected short-term progression risk or typical disease burden), rather than purely aesthetic or frequency-based criteria. The resulting class–risk mappings, along with brief clinical rationales, are summarized in the accompanying tables. We follow the standard preprocessing recommended by MedMNIST (dataset-specific resizing and normalization) and use the official train/validation/test splits for fair comparison.

## F  Detailed Results on the Semi-Synthetic Datasets

In this section, we present detailed experimental results that compare H-Mixup with ERM and other mixup strategies, including naive interpolation (Naive Int.), S-Mixup, and CH-Mixup. Tables 4, 5, 6, and 7 summarize the evaluation results on MNIST, PathMNIST, RetinaMNIST, and OrganMNIST3D. Across different datasets, training model with mixup often improves ERM with respect to $C^{td}$, IBS, and ECE. Among the various mixup methods, the proposed method, H-Mixup, shows significantly effective performance than others. In particular, H-Mixup achieved the best performance on average across all evaluation metrics for MNIST, PathMNIST, and OrganMNIST3D. While H-Mixup shows limited results on RetinaMNIST compared to the other datasets, the proposed method still achieves superior performance in $C^{td}$. In contrast, for the other datasets, H-Mixup demonstrates superior generalization and calibration ability to ERM and other mixup strategies. We again highlight that the proposed mixup strategy, which interpolates the hazard of samples, can benefit survival analysis by enforcing local linearity along interpolations.

## G  Detailed Results for the Sensitivity Analysis

In Figure 5, we analyze the sensitivity of the hyperparameter $\alpha$ in the range [0.1, 0.4], which was validated to be effective in previous works (Zhang et al., 2018). In H-Mixup, the mixing weight $\lambda$ is sampled from $\text{Beta}(\alpha, \alpha)$. Smaller $\alpha$ pushes $\lambda$ toward 0 or 1, whereas larger $\alpha$ pulls $\lambda$ toward 0.5. As demonstrated in Figure 5, although the various models often perform best when $\alpha$ is 0.4 in terms of $C^{td}$, the performance gap is not considerable. Similarly, in most cases, IBS and ECE show relatively robust performance as $\alpha$ varies, except for the DeepHit model trained on PathMNIST.

| Label | Class | Clinical implication | Prognostic Score |
|---|---|---|---|
| 2 | Debris / necrosis | Tumour necrosis is an *independent* adverse prognostic factor in CRC; higher necrosis% associates with shorter DFS/OS across cohorts. (Pollheimer et al., 2010; Väyrynen et al., 2016; Kastinen et al., 2023) | 0.5 |
| 5 | Smooth muscle | Muscularis propria on tiles implies proximity to/through muscle; in pT3 disease, extramural depth >5mm is linked to worse OS/DFS and higher recurrence. (Macchi et al., 2022; Cho et al., 2014; Tong et al., 2014) | 1.0 |
| 7 | Stroma (fibroinflammatory) | A high tumour-stroma ratio (stroma-rich tumours) consistently predicts worse survival in CRC across multiple studies/consortia. (Mesker et al., 2007; Huijbers et al., 2013; Park et al., 2014) | 1.5 |
| 4 | Mucus / mucin | Mucinous adenocarcinoma shows distinct biology and, in many studies/meta-analyses, poorer outcomes and different therapy response patterns (though heterogeneity exists by site/stage). (Wang et al., 2024; Park et al., 2015; Zhang et al., 2022) | 2.0 |
| 0 | Adipose (fat) | Pericolic/perirectal fat on WSI often indicates extramural spread ($\geq$pT3) context; greater extramural depth of invasion links to worse OS/DFS and recurrence, guiding adjuvant therapy decisions. (Macchi et al., 2022; Tong et al., 2014; Cho et al., 2014) | 4.0 |
| 8 | Tumour epithelium | Tumour fraction alone is less predictive than stromal context; stroma-rich microenvironments drive adverse risk. (Mesker et al., 2007; Huijbers et al., 2013) | 7.0 |
| 3 | Lymphocytes (immune) | High T-cell infiltration (Immunoscore) robustly predicts lower recurrence risk and improves prognostic stratification beyond TNM in stage I-III colon cancer. (Pagès et al., 2018) | 8.0 |
| 6 | Normal colon mucosa | No direct prognostic meaning. | 10.0 |
| 1 | Background (non-tissue) | No prognostic meaning. | 12.0 |

Table 3: PathMNIST labels mapped to revised, non-linear Prognostic Scores, rearranged by risk. These scores are corrected based on the provided clinical evidence. The principle is that a **smaller score indicates a more hazardous feature** and a worse prognosis. The non-linear scale is intended to better reflect the relative clinical weight of each pathological finding.

# H  RESULTS FOR C-MIXUP

We evaluate C-Mixup (Yao et al., 2022), a representative mixup method tailored for regression tasks. Unlike standard Mixup which samples mixing pairs uniformly, C-Mixup adjusts the sampling probability based on the similarity of target values. Specifically, given an anchor sample $(x_i, y_i)$, the probability $p_{ij}$ of selecting a sample $(x_j, y_j)$ to mix is defined as:

$$p_{ij} \propto \exp\left(-\frac{|y_i - y_j|_2^2}{2\sigma^2}\right) \tag{44}$$

where $\sigma$ is a bandwidth hyperparameter controlling the sensitivity to label differences. This mechanism encourages mixing between samples with similar target values, thereby aiming to learn more robust representations in regression manifolds.To adapt C-Mixup for survival analysis, we treated

Table 4: Comparison of ERM and mixup strategies ($\alpha = 0.4$) for survival analysis on MNIST dataset. The best performance in each row is highlighted in **bold**.

| Dataset | | MNIST | | | | |
|---|---|---|---|---|---|---|
| Mixup | | ERM | Naive Int. | S-Mixup | CH-Mixup | H-Mixup |
| $C^{td}$ ($\uparrow$) | DeepCox | 0.9140 (0.0028) | 0.9307 (0.0028) | **0.9509** (0.0022) | 0.9298 (0.0031) | 0.9266 (0.0014) |
| | DeepHit | 0.8479 (0.0077) | 0.8372 (0.0039) | 0.8283 (0.0014) | 0.8340 (0.0077) | **0.9031** (0.0067) |
| | DeepIBS | 0.8874 (0.0049) | 0.8886 (0.0053) | 0.8515 (0.0109) | 0.9069 (0.0034) | **0.9193** (0.0077) |
| | DeepMTLR | 0.9087 (0.0065) | 0.9218 (0.0027) | 0.9060 (0.0030) | 0.9204 (0.0029) | **0.9402** (0.0026) |
| | Average | 0.8895 (0.0300) | 0.8946 (0.0423) | 0.8842 (0.0551) | 0.8978 (0.0435) | **0.9223** (0.0155) |
| IBS ($\downarrow$) | DeepCox | **0.0176** (0.0015) | 0.0222 (0.0014) | 0.0373 (0.0007) | 0.0216 (0.0009) | 0.0210 (0.0009) |
| | DeepHit | 0.0197 (0.0011) | 0.0220 (0.0010) | 0.0199 (0.0014) | 0.0234 (0.0017) | **0.0151** (0.0011) |
| | DeepIBS | 0.0159 (0.0011) | 0.0178 (0.0010) | 0.0153 (0.0010) | 0.0192 (0.0009) | **0.0135** (0.0015) |
| | DeepMTLR | 0.0163 (0.0015) | 0.0182 (0.0011) | 0.0164 (0.0014) | 0.0194 (0.0013) | **0.0130** (0.0012) |
| | Average | 0.0174 (0.0017) | 0.0201 (0.0024) | 0.0222 (0.0102) | 0.0209 (0.0020) | **0.0157** (0.0037) |
| ECE ($\downarrow$) | DeepCox | **0.0206** (0.0011) | 0.0368 (0.0015) | 0.0728 (0.0022) | 0.0310 (0.0008) | 0.0274 (0.0007) |
| | DeepHit | 0.0288 (0.0017) | 0.0308 (0.0009) | 0.0367 (0.0012) | 0.0325 (0.0011) | **0.0209** (0.0015) |
| | DeepIBS | 0.0213 (0.0019) | 0.0207 (0.0010) | 0.0255 (0.0020) | 0.0250 (0.0012) | **0.0129** (0.0024) |
| | DeepMTLR | 0.0203 (0.0014) | 0.0217 (0.0014) | 0.0208 (0.0018) | 0.0260 (0.0012) | **0.0110** (0.0009) |
| | Average | 0.0228 (0.0041) | 0.0275 (0.0077) | 0.0390 (0.0235) | 0.0286 (0.0037) | **0.0181** (0.0076) |

Table 5: Comparison of ERM and mixup strategies ($\alpha = 0.4$) for survival analysis on PathMNIST dataset. The best performance in each row is highlighted in **bold**.

| Dataset | | PathMNIST | | | | |
|---|---|---|---|---|---|---|
| Mixup | | ERM | Naive Int. | S-Mixup | CH-Mixup | H-Mixup |
| $C^{td}$ ($\uparrow$) | DeepCox | 0.9063 (0.0099) | 0.9055 (0.0070) | **0.9088** (0.0024) | 0.9081 (0.0050) | 0.9068 (0.0040) |
| | DeepHit | 0.8832 (0.0107) | 0.8505 (0.0054) | 0.8645 (0.0145) | 0.8649 (0.0105) | **0.9112** (0.0112) |
| | DeepIBS | 0.8952 (0.0082) | 0.9051 (0.0081) | 0.8921 (0.0052) | 0.9093 (0.0035) | **0.9116** (0.0070) |
| | DeepMTLR | 0.9036 (0.0122) | 0.9104 (0.0098) | 0.8929 (0.0053) | 0.9158 (0.0051) | **0.9339** (0.0032) |
| | Average | 0.8971 (0.0104) | 0.8929 (0.0284) | 0.8896 (0.0184) | 0.8895 (0.0233) | **0.9159** (0.0122) |
| IBS ($\downarrow$) | DeepCox | 0.0355 (0.0054) | 0.0335 (0.0054) | 0.0712 (0.0021) | 0.0330 (0.0030) | **0.0317** (0.0026) |
| | DeepHit | 0.0773 (0.0151) | 0.0647 (0.0059) | 0.0633 (0.0127) | 0.0849 (0.0118) | **0.0585** (0.0059) |
| | DeepIBS | 0.0495 (0.0019) | 0.0505 (0.0093) | **0.0381** (0.0064) | 0.0530 (0.0053) | 0.0382 (0.0049) |
| | DeepMTLR | 0.0533 (0.0157) | 0.0517 (0.0079) | 0.0420 (0.0072) | 0.0505 (0.0043) | **0.0332** (0.0037) |
| | Average | 0.0539 (0.0174) | 0.0501 (0.0128) | 0.0537 (0.0161) | 0.0554 (0.0216) | **0.0404** (0.0124) |
| ECE ($\downarrow$) | DeepCox | 0.0302 (0.0046) | 0.0314 (0.0035) | 0.0780 (0.0040) | 0.0312 (0.0020) | **0.0288** (0.0017) |
| | DeepHit | 0.0886 (0.0192) | **0.0623** (0.0074) | 0.0768 (0.0110) | 0.0857 (0.0163) | 0.0750 (0.0106) |
| | DeepIBS | 0.0546 (0.0062) | 0.0517 (0.0096) | 0.0484 (0.0027) | 0.0547 (0.0059) | **0.0458** (0.0057) |
| | DeepMTLR | 0.0558 (0.0164) | 0.0521 (0.0058) | 0.0440 (0.0043) | 0.0495 (0.0031) | **0.0357** (0.0042) |
| | Average | 0.0573 (0.0240) | 0.0494 (0.0129) | 0.0618 (0.0181) | 0.0553 (0.0226) | **0.0463** (0.0204) |

the observed time $o$ as the regression target $y$ in Eq. (1). While the sampling strategy relies on the similarity of observed times, the mixing of covariates $x$ and event indicators $\delta$ follows the naive interpolation scheme. Table 8 and Table 9 present the detailed results on semi-synthetic and tabular datasets, respectively. As shown in the tables, C-Mixup does not demonstrate superior performance in the context of survival analysis. Notably, it fails to outperform the standard ERM baseline in

Table 6: Comparison of ERM and mixup strategies ($\alpha = 0.4$) for survival analysis on RetinaMNIST dataset. The best performance in each row is highlighted in **bold**.

| Dataset | | | RetinaMNIST | | | |
|---|---|---|---|---|---|---|
| Mixup | | ERM | Naive Int. | S-Mixup | CH-Mixup | H-Mixup |
| $C^{td}$ (↑) | DeepCox | 0.6990 (0.0190) | 0.7046 (0.0112) | 0.7136 (0.0208) | **0.7137** (0.0243) | 0.7132 (0.0108) |
| | DeepHit | 0.6804 (0.0188) | 0.7143 (0.0062) | 0.6836 (0.0139) | **0.7202** (0.0158) | 0.7087 (0.0107) |
| | DeepIBS | 0.6928 (0.0101) | **0.7049** (0.0130) | 0.7012 (0.0065) | 0.7028 (0.0100) | 0.7041 (0.0223) |
| | DeepMTLR | 0.6999 (0.0110) | **0.7116** (0.0156) | 0.7040 (0.0180) | 0.7065 (0.0139) | 0.7013 (0.0139) |
| | Average | 0.6930 (0.0090) | 0.7089 (0.0049) | 0.7006 (0.0125) | **0.7108** (0.0077) | 0.7068 (0.0052) |
| IBS (↓) | DeepCox | 0.1777 (0.0124) | 0.1711 (0.0128) | **0.1656** (0.0131) | 0.1695 (0.0133) | **0.1656** (0.0106) |
| | DeepHit | 0.1961 (0.0165) | 0.1732 (0.0101) | 0.1962 (0.0123) | **0.1727** (0.0096) | 0.1870 (0.0137) |
| | DeepIBS | 0.1865 (0.0070) | **0.1839** (0.0082) | 0.1874 (0.0142) | 0.1897 (0.0184) | 0.1988 (0.0242) |
| | DeepMTLR | 0.1879 (0.0102) | 0.1882 (0.0064) | **0.1781** (0.0057) | 0.1856 (0.0094) | 0.1892 (0.0056) |
| | Average | 0.1871 (0.0075) | **0.1791** (0.0083) | 0.1818 (0.0131) | 0.1794 (0.0098) | 0.1852 (0.0140) |
| ECE (↓) | DeepCox | 0.0785 (0.0200) | 0.0886 (0.0235) | **0.0630** (0.0121) | 0.0845 (0.0237) | 0.0766 (0.0169) |
| | DeepHit | 0.1171 (0.0126) | 0.1116 (0.0189) | 0.1156 (0.0067) | **0.0926** (0.0093) | 0.1253 (0.0327) |
| | DeepIBS | 0.0848 (0.0221) | **0.0826** (0.0103) | 0.0859 (0.0100) | 0.1092 (0.0155) | 0.1319 (0.0250) |
| | DeepMTLR | 0.0685 (0.0096) | 0.0806 (0.0094) | **0.0635** (0.0123) | 0.0804 (0.0121) | 0.0938 (0.0061) |
| | Average | 0.0872 (0.0210) | 0.0909 (0.0142) | **0.0820** (0.0248) | 0.0917 (0.0127) | 0.1069 (0.0262) |

Table 7: Comparison of ERM and mixup strategies ($\alpha = 0.4$) for survival analysis on OrganM-NIST3D dataset. The best performance in each row is highlighted in **bold**.

| Dataset | | | OrganMNIST3D | | | |
|---|---|---|---|---|---|---|
| Mixup | | ERM | Naive Int. | S-Mixup | CH-Mixup | H-Mixup |
| $C^{td}$ (↑) | DeepCox | 0.8741 (0.0055) | 0.8660 (0.0123) | 0.8646 (0.0162) | 0.8767 (0.0040) | **0.8819** (0.0022) |
| | DeepHit | 0.8425 (0.0080) | 0.8451 (0.0037) | 0.8391 (0.0093) | 0.8527 (0.0183) | **0.8810** (0.0124) |
| | DeepIBS | 0.8799 (0.0088) | 0.8702 (0.0114) | 0.8764 (0.0115) | 0.8890 (0.0102) | **0.8999** (0.0116) |
| | DeepMTLR | 0.8773 (0.0040) | 0.8890 (0.0064) | 0.8810 (0.0053) | 0.8910 (0.0092) | **0.9010** (0.0102) |
| | Average | 0.8685 (0.0175) | 0.8676 (0.0180) | 0.8653 (0.0188) | 0.8774 (0.0176) | **0.8910** (0.0110) |
| IBS (↓) | DeepCox | 0.1022 (0.0041) | 0.1001 (0.0094) | 0.0986 (0.0100) | 0.1010 (0.0048) | **0.0949** (0.0071) |
| | DeepHit | 0.0876 (0.0075) | 0.0843 (0.0062) | 0.0930 (0.0066) | 0.0831 (0.0120) | **0.0701** (0.0081) |
| | DeepIBS | 0.0907 (0.0076) | 0.0911 (0.0048) | 0.0828 (0.0061) | 0.0955 (0.0098) | **0.0772** (0.0061) |
| | DeepMTLR | 0.0936 (0.0088) | 0.0904 (0.0088) | 0.0864 (0.0053) | 0.1041 (0.0110) | **0.0824** (0.0101) |
| | Average | 0.0935 (0.0063) | 0.0915 (0.0065) | 0.0902 (0.0070) | 0.0959 (0.0093) | **0.0812** (0.0105) |
| ECE (↓) | DeepCox | 0.0619 (0.0035) | 0.0618 (0.0061) | **0.0427** (0.0054) | 0.0594 (0.0054) | 0.0544 (0.0052) |
| | DeepHit | 0.0643 (0.0111) | 0.0707 (0.0070) | 0.0870 (0.0069) | 0.0703 (0.0107) | **0.0536** (0.0090) |
| | DeepIBS | 0.0631 (0.0072) | 0.0575 (0.0035) | 0.0481 (0.0055) | 0.0635 (0.0069) | **0.0474** (0.0059) |
| | DeepMTLR | 0.0660 (0.0117) | 0.0689 (0.0086) | 0.0552 (0.0057) | 0.0826 (0.0110) | **0.0516** (0.0056) |
| | Average | 0.0638 (0.0018) | 0.0647 (0.0062) | 0.0583 (0.0198) | 0.0690 (0.0101) | **0.0518** (0.0031) |

most scenarios. This suggests that directly applying regression-based mixup techniques—which treat observed time as a ground truth scalar without accounting for the censoring mechanism—is insufficient for effective survival modeling.

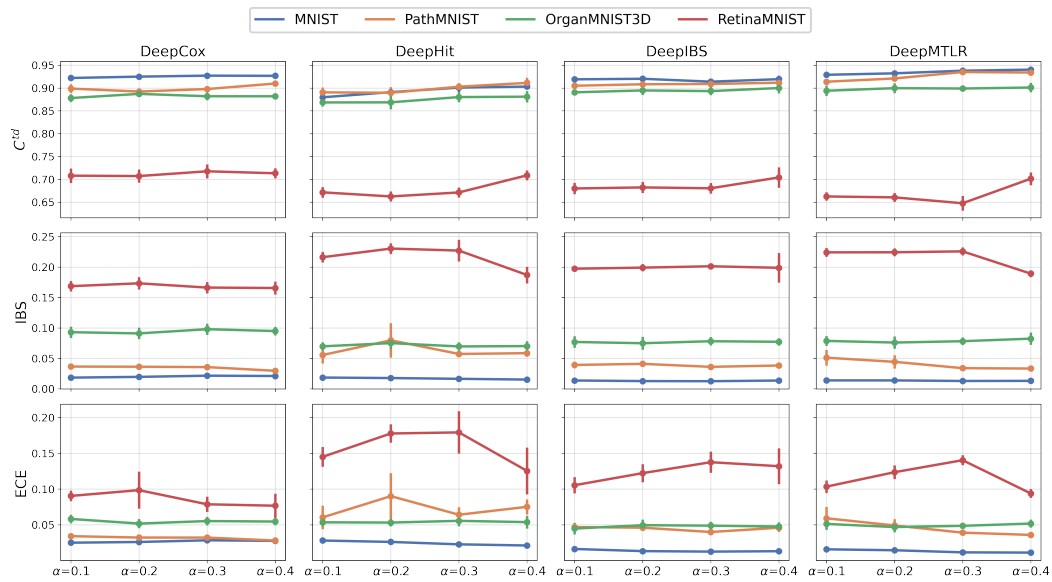

Figure 5: Sensitivity analysis on $\alpha \in \{0.1, 0.2, 0.3, 0.4\}$ for the different deep survival analysis models on the MNIST, PathMNIST, OrganMNIST3D, and RetinaMNIST datasets. Each row shows the $C^{td}$, IBS, and ECE performance, respectively, while each color represents the results of different datasets.

Table 8: Comparison of ERM, C-Mixup, and H-Mixup ($\alpha = 0.4$) on the semi-synthetic datasets. The best result among the three methods is highlighted in **bold**, and the value in parentheses denotes the standard deviation over the five repetitions.

| Metric | Dataset | DeepCox ERM | C-Mixup | H-Mixup | DeepHit ERM | C-Mixup | H-Mixup | DeepIBS ERM | C-Mixup | H-Mixup | DeepMTLR ERM | C-Mixup | H-Mixup |
|---|---|---|---|---|---|---|---|---|---|---|---|---|---|
| $C^{td}$ (↑) | MNIST | 0.9140 (0.0028) | **0.9315** (0.0032) | 0.9266 (0.0014) | 0.8479 (0.0077) | 0.8403 (0.0056) | **0.9031** (0.0067) | 0.8874 (0.0049) | 0.9016 (0.0030) | **0.9193** (0.0077) | 0.9087 (0.0065) | 0.9160 (0.0043) | **0.9402** (0.0026) |
| | OrganMNIST3D | 0.8741 (0.0055) | 0.8707 (0.0099) | **0.8819** (0.0022) | 0.8425 (0.0080) | 0.8501 (0.0065) | **0.8810** (0.0124) | 0.8799 (0.0088) | 0.8768 (0.0093) | **0.8999** (0.0116) | 0.8773 (0.0040) | 0.8826 (0.0100) | **0.9010** (0.0102) |
| | PathMNIST | 0.9063 (0.0099) | 0.9024 (0.0099) | **0.9068** (0.0040) | 0.8832 (0.0107) | 0.8688 (0.0047) | **0.9112** (0.0070) | 0.8952 (0.0082) | 0.8919 (0.0091) | **0.9116** (0.0070) | 0.9036 (0.0122) | 0.8994 (0.0087) | **0.9339** (0.0032) |
| | RetinaMNIST | 0.6990 (0.0190) | 0.7056 (0.0158) | **0.7132** (0.0108) | 0.6804 (0.0188) | 0.6776 (0.0212) | **0.7087** (0.0107) | 0.6928 (0.0101) | 0.6842 (0.0145) | **0.7041** (0.0223) | 0.6999 (0.0110) | 0.6711 (0.0098) | **0.7013** (0.0139) |
| IBS (↓) | MNIST | **0.0176** (0.0015) | 0.0201 (0.0014) | 0.0210 (0.0009) | 0.0197 (0.0011) | 0.0185 (0.0017) | **0.0151** (0.0011) | 0.0159 (0.0011) | 0.0163 (0.0067) | **0.0135** (0.0015) | 0.0163 (0.0015) | 0.0168 (0.0068) | **0.0130** (0.0012) |
| | OrganMNIST3D | 0.1022 (0.0041) | 0.0999 (0.0089) | **0.0949** (0.0071) | 0.0876 (0.0075) | 0.0839 (0.0055) | **0.0701** (0.0081) | 0.0907 (0.0076) | 0.0907 (0.0089) | **0.0772** (0.0061) | 0.0936 (0.0088) | 0.0944 (0.0075) | **0.0824** (0.0101) |
| | PathMNIST | 0.0355 (0.0054) | 0.0429 (0.0051) | **0.0317** (0.0026) | 0.0773 (0.0151) | 0.0679 (0.0051) | **0.0585** (0.0059) | 0.0495 (0.0019) | 0.0652 (0.0135) | **0.0382** (0.0049) | 0.0533 (0.0157) | 0.0535 (0.0048) | **0.0332** (0.0037) |
| | RetinaMNIST | 0.1777 (0.0124) | 0.1711 (0.0089) | **0.1656** (0.0106) | 0.1961 (0.0165) | 0.2113 (0.0050) | **0.1870** (0.0137) | **0.1865** (0.0070) | 0.2045 (0.0075) | 0.1988 (0.0242) | **0.1879** (0.0102) | 0.2350 (0.0122) | 0.1892 (0.0056) |
| ECE (↓) | MNIST | **0.0206** (0.0011) | 0.0320 (0.0020) | 0.0274 (0.0007) | 0.0288 (0.0017) | 0.0260 (0.0042) | **0.0209** (0.0015) | 0.0213 (0.0019) | 0.0214 (0.0023) | **0.0129** (0.0024) | 0.0203 (0.0014) | 0.0213 (0.0028) | **0.0110** (0.0009) |
| | OrganMNIST3D | 0.0619 (0.0035) | 0.0590 (0.0080) | **0.0544** (0.0052) | 0.1171 (0.0111) | **0.0642** (0.0069) | 0.1253 (0.0090) | 0.0631 (0.0072) | **0.0625** (0.0084) | 0.1319 (0.0059) | 0.0660 (0.0117) | 0.0672 (0.0061) | **0.0516** (0.0056) |
| | PathMNIST | 0.0302 (0.0046) | 0.0352 (0.0024) | **0.0288** (0.0017) | 0.0886 (0.0192) | **0.0715** (0.0053) | 0.0750 (0.0106) | 0.0546 (0.0067) | 0.0536 (0.0057) | **0.0458** (0.0042) | 0.0558 (0.0164) | 0.0461 (0.0028) | **0.0357** (0.0042) |
| | RetinaMNIST | 0.0785 (0.0200) | **0.0696** (0.0117) | 0.0766 (0.0169) | 0.0643 (0.0126) | 0.1269 (0.0079) | **0.0536** (0.0327) | 0.0848 (0.0221) | 0.1092 (0.0111) | **0.0474** (0.0250) | **0.0685** (0.0096) | 0.0905 (0.0060) | 0.0938 (0.0061) |

Table 9: Comparison of ERM, C-Mixup, and H-Mixup ($\alpha = 0.4$) on the tabular datasets. The best result among the three methods is highlighted in **bold**, and the value in parentheses denotes the standard deviation over the five repetitions.

| Metric | Dataset | DeepCox | | | DeepHit | | | DeepIBS | | | DeepMTLR | | |
|---|---|---|---|---|---|---|---|---|---|---|---|---|---|
| | | ERM | C-Mixup | H-Mixup | ERM | C-Mixup | H-Mixup | ERM | C-Mixup | H-Mixup | ERM | C-Mixup | H-Mixup |
| $C^{td}$ (↑) | Flchain | 0.7956 (0.0084) | 0.7945 (0.0060) | **0.7966** (0.0077) | **0.7954** (0.0082) | 0.6819 (0.0134) | 0.7922 (0.0072) | 0.6959 (0.0143) | 0.6469 (0.0196) | **0.7808** (0.0109) | 0.7008 (0.0149) | 0.6910 (0.0117) | **0.7811** (0.0090) |
| | Framingham | **0.7730** (0.0183) | 0.7370 (0.0174) | 0.7719 (0.0199) | **0.7403** (0.0092) | 0.7366 (0.0100) | 0.7296 (0.0179) | **0.7678** (0.0192) | 0.7405 (0.0191) | 0.7663 (0.0264) | 0.7696 (0.0186) | 0.7473 (0.0119) | **0.7709** (0.0175) |
| | Metabric | 0.6452 (0.0131) | 0.6302 (0.0126) | **0.6532** (0.0081) | 0.6340 (0.0125) | 0.6312 (0.0117) | **0.6527** (0.0146) | **0.6533** (0.0101) | 0.6319 (0.0167) | 0.6512 (0.0175) | 0.6492 (0.0034) | 0.6445 (0.0129) | **0.6638** (0.0140) |
| IBS (↓) | Flchain | 0.1074 (0.0032) | **0.1066** (0.0022) | 0.1072 (0.0033) | **0.1121** (0.0038) | 0.1789 (0.0093) | 0.1310 (0.0054) | **0.1101** (0.0020) | 0.2054 (0.0205) | 0.1102 (0.0022) | 0.1076 (0.0018) | 0.2147 (0.0112) | **0.1067** (0.0015) |
| | Framingham | **0.0630** (0.0060) | 0.0939 (0.0049) | **0.0630** (0.0062) | 0.3485 (0.0226) | **0.3073** (0.0075) | 0.3250 (0.0201) | **0.0638** (0.0057) | 0.0979 (0.0049) | 0.0640 (0.0054) | **0.0635** (0.0060) | 0.0955 (0.0033) | 0.0640 (0.0054) |
| | Metabric | 0.1902 (0.0050) | 0.2007 (0.0046) | **0.1895** (0.0049) | 0.2026 (0.0038) | 0.2126 (0.0089) | **0.1983** (0.0047) | **0.1920** (0.0067) | 0.2217 (0.0082) | 0.1928 (0.0060) | 0.1998 (0.0070) | 0.2144 (0.0120) | **0.1928** (0.0102) |
| ECE (↓) | Flchain | **0.0704** (0.0119) | 0.0730 (0.0091) | 0.0717 (0.0127) | 0.1244 (0.0226) | 0.2223 (0.0122) | **0.0911** (0.0082) | **0.0639** (0.0108) | 0.2642 (0.0144) | 0.1068 (0.0130) | **0.0409** (0.0073) | 0.2791 (0.0056) | 0.0520 (0.0094) |
| | Framingham | 0.0155 (0.0021) | 0.0294 (0.0091) | **0.0150** (0.0028) | 0.5223 (0.0266) | **0.4313** (0.0088) | 0.4977 (0.0161) | **0.0215** (0.0082) | 0.0481 (0.0070) | 0.0254 (0.0112) | **0.0170** (0.0055) | 0.0380 (0.0044) | 0.0233 (0.0077) |
| | Metabric | **0.0666** (0.0129) | 0.0828 (0.0064) | 0.0667 (0.0142) | 0.1162 (0.0107) | **0.0966** (0.0060) | 0.1181 (0.0059) | **0.0638** (0.0069) | 0.0952 (0.0080) | 0.0769 (0.0127) | 0.0524 (0.0115) | 0.0854 (0.0125) | **0.0508** (0.0105) |

