# OpenReview forum: "Mixup for Survival Analysis"
_ICLR.cc/2026/Conference — ICLR 2026 Conference Desk Rejected Submission_

### Official Review · Reviewer_TFSk · 2025-10-23

**Soundness:** 1
**Presentation:** 2
**Contribution:** 1
**Rating:** 2
**Confidence:** 4

**Summary:**

The paper aims at solving the overfitting and poor generalization of the deep learning survival analysis, using mixup -- a type of vicinal risk minimization. The paper proposes H-Mixup, for survival analysis that interpolates covariates and hazard function through data augmentation.  Theoretically, the paper claims the local linearity in the hazard function and tighter generalization bounds than empirical risk minimization (ERM). Empirically, the method demonstrates superior performance on semi-synthetic datasets, and similar performance with ERM on real datasets.

**Strengths:**

1. This paper is the first work that aims to mitigate the sharp decision boundary issues in survival analysis.

**Weaknesses:**

I find the paper fundamentally flawed in three aspects: (1) an unconvincing and even misleading motivating example, (2) incorrect theorem and proof, and (3) weak empirical evidence. I elaborate on these points below.

---
## Flawed Motivation and Misleading Example

The motivating example in Figure 1 is not convincing. While the figure visually shows H-MixUp producing a smooth decision boundary between high- and low-risk samples, this effect is largely an artifact of an unrealistically extreme setup. If one carefully reads Appendix A, the two groups’ hazard functions -- both constant -- differ by several orders of magnitude:
- Low-risk group: $h(t) = 0.25*\exp(3)\approx 5$
- High-risk group: $h(t) = 0.25*\exp(10)\approx 5507$

That is more than a **thousand-fold difference**. Meanwhile, the censoring hazard is a negligible 0.18, meaning virtually no censoring occurs. Under such a large signal gap, a sharp decision boundary is expected and not problematic -- indeed, this is what ERM yields.

Plotting the survival functions makes the issue clearer: for the high-risk group, the survival probability essentially collapses to zero immediately after $t>0$ (e.g., $S(t=0.001) = \exp(-5507* 0.001)\approx 0.004$). The ERM prediction correctly reflects this -- darkest regions align with high-risk samples. In contrast, the H-MixUp panel misplaces the darkest regions (and colors the high-risk region as not-so-dark), indicating potential **miscalibration**. In fact, the naive interpolation seems to outperform H-MixUp here.

This raises a deeper concern: if a model is already well-calibrated, why should we prefer a smooth hazard or survival boundary over a sharp one? Since the counterfactual outcomes in the interpolated regions are **unobservable**, there is no principled reason to favor smoothness. If the true boundary is sharp, enforcing smoothness is actively harmful. This undermines the entire motivational basis of the paper.

---
## Incorrect and Non-Substantive Theoretical Results

Theorem 1, the core theoretical result, is both **incorrect in conclusion** and **incorrectly derived**.

First, the stated conclusion (Eq. 8) is not what one would want:

$h(t \mid \tilde{x}) = \lambda h(\underline{\lambda} t \mid x) + (1 - \lambda) h(\underline{(1-\lambda)} t \mid x')$

i.e., a convex combination of hazards *at earlier times* (the two underlined terms). But the intended property should be

$h(t \mid \tilde{x}) = \lambda h(t \mid x) + (1 - \lambda) h(t \mid x')$

a convex combination **at the same time point**. The proven statement is thus irrelevant to the algorithm’s desired behavior.

Second, the proof itself is incorrect. In line 795 of the Appendix, the mixup time is represented as $\min(\frac{T}{\lambda}, \frac{T'}{1-\lambda})$ -- using event times. However, in the definition (Eq. 8), the mixup involves $\min(\frac{O}{\lambda}, \frac{O'}{1-\lambda})$  -- **observation times**. This is not a subtle difference: plugging in the correct definition breaks the derivation since $P(O > \lambda t) \not= S(\lambda t \mid x)$ and $P(O' > (1 - \lambda) t) \not= S((1-\lambda) t \mid x')$ in line 801. The subsequent steps simply do not hold.

Moreover, Theorem 2 is stated with minimal justification. The authors essentially port existing Mixup theory to the survival setting without showing why the log-likelihood or partial likelihood is L-Lipschitz, which is not trivial. I think it requires some extra assumption for this property to hold.

---
## Weak and Selective Empirical Evidence

While the method performs adequately on synthetic image-based datasets, its real-world performance tells a different story. Table 8 (buried at the end of the Appendix) shows that across 36 comparisons, standard ERM actually outperforms H-MixUp 18 times and ties once. These results contradict the strong claims in the main text about “consistent improvements” (line 410).

It seems that the proposed methods (and the other Mixup methods) mostly work on image datasets, yet the authors provide no evidence on real image-based survival datasets, or any synthetic datasets which is not derived from MNIST.

More importantly, the comparison with a naive interpolation baseline is missing. Without it, it is unclear whether the gains (if any) are due to the proposed mixup strategy or simply from data (feature) augmentation.

**Questions:**

Beyond the three major weaknesses discussed above, I also have several additional concerns:
1. The paper claims (line 193) that the mixed observation times “tend to skew a bit later, ..., but some will still be early.” However, since $0<\lambda<1$, both rescaled observation times $\frac{o}{\lambda}$ and $\frac{o'}{1-\lambda}$ must be strictly larger than the original observed times. It is therefore unclear how the resulting mixup time could ever be earlier than either of the original observation times.
2. It is difficult to grasp the key implications of Theorem 2 by reading Section 3 alone. If the authors choose to present the formal statement of the theorem in the main text, they should at minimum explain all the terms, assumptions, and their significance directly in that section rather than deferring entirely to the appendix. As written, it is unnecessarily opaque.
3. The ECE metrics for evaluating calibration: how does it handle censoring?
4. In Section 2, non-informative censoring is defined as $T \perp C \mid X$, while in lines 1304-1305, it is defined as censoring time is independent of both progrnostic factors and true failure times. Leads to contradictory definitions.
5. Section 5 introduces two alternative mixup strategies, but they are neither theoretically justified nor empirically evaluated. If these variants are intended to strengthen the paper’s contribution, the lack of corresponding experiments undermines that goal; if not, they only add unnecessary noise.

---

> ### Comment · Reviewer_TFSk · 2025-11-14
> **Incorrect references**
>
> It is interesting to me that reviewers `488W ` and `RLto ` have both raised concerns regarding the references in this submission (while both have high scores).
>
> After a quick scan of the reference list up to page 11, I also identified several problematic references:
>
> > Arslan Chaudhry, Christoph H. Lampert, and Muhammad Haris Khan. The role of local linearity in generalization. In NeurIPS, 2022.
>
> This paper does not exist.
>
> > Haiyong Cui, Fan Zhang, Hui Fan, and Na Xu. Mulitdeepsurv: survival analysis of gastric cancer based on deep learning and multimodal data. PMC, 2024.
>
> Wrong author, wrong venue.
>
> > Serge Harb Fotso. Deep neural networks for survival analysis based on a multi-task framework.arXiv preprint arXiv:1801.05512, 2018.
>
> Incorrect author (first and middle) name.
>
> > Mark Goldstein, Xintian Han, Aahlad Puli, Adler J. Perotte, and Rajesh Ranganath. X-cal: Explicit calibration for survival analysis. In NeurIPS, 2020. Semi-synthetic Survival-MNIST with Gamma times.
>
> Unclear post description attached after reference.
>
> > Hrayr Harutyunyan, Adrian R. L. Oliva, and Mihaela van der Schaar. Match-net: Dynamic prediction in survival analysis using convolutional neural networks. In NeurIPS ML4H, 2019.
>
> Wrong author, wrong venue.
>
> > Yiding Jiang, Behnam Neyshabur, Hossein Mobahi, Dilip Krishnan, and Samy Bengio. Fantastic generalization measures and where to find them. In Advances in Neural Information Processing Systems (NeurIPS), pp. 11780–11791, 2019.
>
> Wrong venue.
>
> After page 11, I quickly encountered at least three more incorrect references -- wrong authors, wrong venues, or both. At this point, rather than debugging the entire bibliography for the authors, I will leave that as their responsibility.
>
> ---
>
> These issues raise serious concerns regarding the scholarly integrity of the submission and the apparent use of LLM-generated or LLM-assisted references without proper verification. While I am keeping my score unchanged for now, I believe the authors must explain and address these problems in their rebuttal.

---

> ### Author Response · Authors · 2025-11-22
>
> ### Flawed Motivation and Misleading Example
> - We sincerely thank the reviewer for this thoughtful comment, which gives us an opportunity to clarify the motivation behind Figure 1. Our intention with Figure 1 is precisely to offer a clear visual illustration of VRM behavior. In the original mixup paper, a closely related toy example with synthetic, clearly separated classes is used for the same purpose to show how training on interpolated samples reshapes the distribution of predicted values. Our toy example mirrors that construction in a survival analysis setting, adapting the same explanatory idea rather than proposing a realistic clinical data-generating process.
>
> - Our Figure 1 is meant to show exactly this contrast: in our toy example, ERM, C-Mixup (newly added), and naive interpolation all spuriously assign relatively low risk to the center of the dense high-risk cluster; visually, the darkest regions are pushed toward the empirical boundary instead of covering the interior of the high-risk group. H-Mixup, in contrast, preserves high risk in the center of the high-risk cluster while producing a smoother survival surface in its vicinity. By construction, C-Mixup and naive interpolation simply average follow-up times between observed samples. They do not define a coherent vicinal distribution in covariate–time space and therefore fail to provide the desired smoothing on the vicinal distribution that H-Mixup achieves.
>
> - Under VRM, one does not assume that the true decision boundary or hazard surface is smooth in the input space. Instead, one replaces ERM on point masses with minimization of vicinal risk, where the empirical distribution is smoothed by a vicinity distribution around training examples. Mixup instantiates this by training on convex combinations of inputs and labels, which encourages approximately linear behavior between observed samples. In other words, the objective is to learn a function with controlled complexity on the vicinal distribution, not to exactly reproduce a possibly discontinuous true hazard boundary.
> From this perspective, the argument that "if the model is already well-calibrated and the true boundary is sharp, enforcing smoothness is harmful" is, to the best of our knowledge, conceptually orthogonal to what mixup is designed to do. If the model class were perfectly specified, data were infinite, and ERM produced a fully calibrated estimator, then any regularizer could in principle be harmful. But in the finite-sample, high-dimensional regimes where deep survival models are used, ERM may yield over-confident predictions with sharp transitions in dense regions of the feature space. Mixup-style VRM is introduced to mitigate this behavior by smoothing decision regions and reducing overconfidence, which has been repeatedly observed in the classification literature. Our Figure 1 is meant to show that ERM can assign nearly deterministic high-risk predictions in the dense high-risk cluster, whereas H-Mixup produces a smoother survival surface in those regions.
>
> - Finally, we note that the motivating example is deliberately stylized (constant hazards with a large gap) because it is inherited from the original mixup visualization and is intended as an intuitive toy example, not as a realistic clinical data-generating process. The key point is that H-Mixup implements the analogous VRM principle as the original mixup paper in the survival setting: training on synthetic interpolated samples to regularize the model toward smoother mappings on the vicinal distribution.

---

> > ### Author Response · Authors · 2025-11-22
> >
> > ### Incorrect and Non-Substantive Theoretical Results
> > - Regarding Theorem 1
> >     - Thank you for pointing this out. The reviewer is correct that there's a gap between H-Mixup and Theorem 1: Equation (8) defines H-Mixup in terms of observation times $\min(O/\lambda, O'/(1-\lambda))$, whereas Theorem 1 derivation is about event times $\min(T/\lambda, T'/(1-\lambda))$.
> >     The connection between the two is not trivial and must be addressed. In the revision, we have rewritten Theorem 1 to rigorously bridge this gap. The new proof proceeds in two steps. First, we work at the latent level and define the mixed event time $\tilde T = \min(T/\lambda, T'/(1-\lambda))$; using conditional independence, we show that its hazard satisfies the temporally scaled linearity:$$h_{\tilde T}(t\mid x,x') = \lambda h(\lambda t\mid x) + (1-\lambda)h((1-\lambda)t\mid x').$$Second, we return to the observable construction and show algebraically that the H-Mixup output $(\tilde o,\tilde\delta) = (\min(O/\lambda, O'/(1-\lambda)), \tilde\delta)$ is exactly a right-censored observation of $\tilde T$, and that under our covariate-dependent independent censoring assumption, the event-time hazard of $\tilde T$ is identifiable from $(\tilde o,\tilde\delta)$. We present only this sketch in the main text and provide the full, rigorous derivation in Appendix B.1.
> >
> > - Regarding Theorem 2 and Proposition 1
> >     - We thank the reviewer for critically examining the theoretical justification of Theorem 2. We acknowledge that in standard Mixup literature, theoretical analysis often relies on global Lipschitz continuity or smoothness to characterize Mixup as implicit gradient regularization. As the reviewer correctly points out, the Cox partial likelihood does not satisfy this global Lipschitz condition, which would make such a port of the theory invalid.
> >     However, our proof for Theorem 2 is not based on this implicit regularization approximation. Instead, it builds directly on Lemma 1, which is a general generalization bound established under the sub-Gaussian assumption. Lemma 1 proves that for sub-Gaussian data, the generalization gap can be controlled by the tail behavior of the distribution, without requiring the loss function to be globally $L$-Lipschitz. In Theorem 2, we apply this bound to the Mixup-augmented distribution. Since the convex combination of sub-Gaussian random variables remains sub-Gaussian (for covariates), the conditions of Lemma 1 are satisfied.
> >     - The standard mixup style analysis can be applied if the loss function is $L$-Lipschitz. We have mentioned this after Lemma 1. However, like the reviewer mentioned, $L$-Lipschitzness is not trivial for survival models. We added Lemma 2 which demonstrates the Lipschitz continuitiy of survival NLLs under certain conditions (especially when the outputs are bounded via e.g. clipping). Although this is pretty restrictive case, this further allows us to port H-Mixup into the standard mixup style analysis.
> >     - However, we acknowledge that our bound for the vicinal bias term, $B_H(\phi)$, explicitly assumes that the loss function is $L$-Lipschitz with respect to the hazard to control the approximation error. To ensure this proposition holds for survival objectives like the Cox partial likelihood—which are not globally Lipschitz—we have added Lemma 2 (Lipschitz Continuity of Survival NLL) in Appendix B.2. This lemma demonstrates that under practical constraints where model outputs are bounded (e.g., via clipping or weight decay), standard survival loss functions satisfy the Lipschitz continuity required. Consequently, Lemma 2 validates the assumptions necessary for Proposition 1, thereby completing the theoretical framework supporting Theorem 2.
> >
> > ### Weak and Selective Empirical Evidence
> > - We thank the reviewer for this careful empirical assessment. We agree that, in the original version, the evidence was concentrated on semi-synthetic image data.
> > - To address the concern about real-world performance, we added experiments on a real image-based survival dataset, COVID-19-NY-SBU , consisting of 1,384 patients with an 86.8% censoring rate (Table 1, Sec. 4.2). Across all four backbone models (DeepCox, DeepHit, DeepIBS, DeepMTLR), H-Mixup improves in the majority of cases and on average it yields better performance than ERM. These results show that H-Mixup can provide tangible gains beyond semi-synthetic benchmarks.
> > - Regarding the missing naive interpolation baseline, we now compare H-Mixup against both C-Mixup across four backbone models and four image-based semi-synthetic datasets, and we also include these baselines in the motivating toy example (Figures 1 and 4). Across these benchmarks, H-Mixup outperforms both ERM and C-Mixup. This pattern indicates that the gains of H-Mixup cannot be explained solely by generic feature mixing, but are attributable to the hazard-aware construction of the vicinal distribution.

---

> > > ### Author Response · Authors · 2025-11-22
> > >
> > > #### Questions
> > > - The paper claims (line 193) that the mixed observation times “tend to skew a bit later, ...,
> > >     - By 'some will still be early,' we meant that the mixed time $\tilde{o}$ can be smaller than one of the parent times (e.g., $\tilde{o} < o$). For instance, if $o = 10$, $o' = 2$, and $\lambda= 0.5$, then $\tilde{o} = \min\{20, 4\} = 4$. In this case, $\tilde{o} > o'$ but $\tilde{o} < o$. However, we acknowledge that this statement caused confusion without aiding the understanding of our approach, so we have removed it.
> > >     - Although $\tilde{o}$ can be numerically larger than the original times, this does not inflate the true follow-up duration. H-Mixup defines a new random variable $\tilde{O}$ under a vicinal distribution $P_v$. Therefore, $\tilde{o}$ is a reparametrization of the time for the mixed samples; $\tilde{o}$ and $o$ are defined in different scales. The apparent inflation is an artifact of this rescaled random variable rather than a change in the underlying event process. Provided the censored likelihood is evaluated under this new vicinal data distribution, the estimator remains consistent. Moreover, since survival models predict normalized quantities rather than raw outcomes, they avoid scale sensitivity.
> > >
> > > - It is difficult to grasp the key implications of Theorem 2 by reading Section 3 alone. If the authors choose to present the formal statement of the theorem in the main text, they should at minimum explain all the terms, assumptions, and their significance directly in that section rather than deferring entirely to the appendix. As written, it is unnecessarily opaque.
> > >     - We thank the reviewer for this valuable feedback. We fully agree that the current presentation of Section 3 is dense and difficult to follow without constant reference to the appendix. This compression was an unfortunate necessity due to the strict space limitations of the initial submission, which forced us to move key details out of the main text. We acknowledge that this resulted in the section being unnecessarily opaque. If allowed the additional space, we are committed to revising Section 3 to explicitly explain these terms and assumptions in the main text, ensuring Section 3 is self-contained and accessible.
> > >
> > > - The ECE metrics for evaluating calibration: how does it handle censoring?
> > >     - At each evaluation time $t$, we define a binary outcome $Y(t) = \mathbb{I}(T > t)$ and the predicted survival probability $\hat{S}(t)$. The ECE is then computed by binning subjects based on $\hat{S}(t)$ and calculating the weighted average of the absolute differences between the mean predicted survival and the empirical proportion of event-free subjects within each bin.
> > >     This handles censoring using a complete-case analysis, which is consistent with the setup for a time-dependent Brier score. The ECE at time $t$ is computed only using subjects for whom this binary status $Y(t)$ is definitively known. This includes all subjects who experienced an event (at any time) and those who were observed event-free at or after time $t$; any subjects censored before $t$ are excluded from the calculation.
> > >
> > > - In Section 2, non-informative censoring is defined as $T \perp C | X$, while in lines 1304-1305, it is defined as censoring time is independent of both progrnostic factors and true failure times. Leads to contradictory definitions.
> > >     - Thank you for highlighting the distinction between the model assumption and the simulation scenario. We have clarified that while the standard assumption is conditional independence ($T \perp C | X$), our simulation employs the stronger condition of completely random censoring, where $C$ is independent of both $X$ and $T$. This condition implies the required assumption of non-informative censoring ($T \perp C | X$), ensuring the model's requirements are met. We have revised the description of the scenario to ensure the terminology is consistent with the definition provided in Section 2.
> > >
> > > - Section 5 introduces two alternative mixup strategies, but they are neither theoretically justified nor empirically evaluated. If these variants are intended to strengthen the paper’s contribution, the lack of corresponding experiments undermines that goal; if not, they only add unnecessary noise.
> > >     - We agree that the alternative mixup strategies are somewhat redundant, as we do not provide the same depth of analysis and insight as we do for H-Mixup. Consequently, we have moved these strategies to the appendix to serve as potential directions for future study without claiming them as central contributions. This adjustment allowed us to prioritize the added details on temporally-scaled linearity (Figure 2) and the results on real-world data (Table 1 and Section 4.2).

---

> ### Author Response · Authors · 2025-11-22
>
> We sincerely apologize for the errors regarding the refereces, which arose from our naive use of LLMs to assist with the retrieval and discovery of related work. As noted in the submission, LLMs assisted us in assembling the reference list, but we did not conduct the thorough manual verification that should have accompanied this step. This was our responsibility, and we genuinely regret the oversight.
>
> In response, we have now reviewed the entire reference list item by item to ensure that each citation corresponds to an existing work, that the author list and venue are correct, and that the work is cited in an appropriate context. This careful check should have been completed before the original submission.
>
> The detailed corrections are provided in the revision text above.

---

### Official Review · Reviewer_RLto · 2025-10-26

**Soundness:** 4
**Presentation:** 3
**Contribution:** 4
**Rating:** 8
**Confidence:** 5

**Summary:**

This paper introduces H-Mixup, an adaptation of the Mixup data augmentation strategy for  censored survival outcomes. Standard Mixup interpolates inputs and labels to improve generalization, but this approach breaks down when applied to time-to-event data due to censoring constraints. H-Mixup resolves this by shifting the focus from observed censored outcomes to the latent event-time distribution by interpolates covariates and hazards through data augmentation. The method provides both theoretical guarantees (showing reductions in Rademacher complexity and tighter generalization bounds) and practical improvements across multiple benchmarks.

Overall, this is a strong paper that effectively extends Mixup to censored survival data, bridging a theoretical and practical gap in deep time-to-event modeling. H-Mixup demonstrates clear performance improvements, sound theory, and broad applicability. The paper is well-executed and accessible, with only minor limitations related to hyperparameter sensitivity and robustness analysis

**Strengths:**

1.  This is the first rigorous formulation of Mixup for censored survival data.

2.  The paper establishes generalization bounds specific to survival modeling.

3. Through comprehensive experiments, H-Mixup demonstrates consistent gains across synthetic and real-world datasets.

**Weaknesses:**

1.  The method’s dependence on the alignment between mixing assumptions and true hazard structure.

2.  It Requires tuning of the Beta distribution parameter α for stability.

3.  Limited analysis of domain shifts.

**Questions:**

1. Could the approach be extended to informative censoring scenarios, where censoring depends on covariates or risk, and how would it work?

2. Could adaptive or learned mixing coefficients improve alignment between interpolated and true hazards?

3. How sensitive is performance to the event/censor ratio correction scheme in highly imbalanced data?

4. Another potential competitor to H-Mixup is generative modeling. How would its performance compare with that of H-Mixup?

Note:  The reference for Zhong et al. (2021) is incorrect.

---

> ### Author Response · Authors · 2025-11-22
>
> - The method’s dependence on the alignment between mixing assumptions and true hazard structure.
>     - We thank the reviewer for highlighting this fundamental aspect of our method. We agree that H-Mixup imposes an inductive bias, specifically, local linearity in the hazard space, and that its effectiveness relies on the extent to which this assumption holds.
>     However, we respectfully note that this is not a limitation unique to our specific method, but a fundamental characteristic of the Mixup principle and VRM in general. The primary goal of VRM is not to perfectly recover the true data manifold, but to introduce a regularization effect that trades a controlled amount of bias for a significant reduction in variance. As formalized in Proposition 1, we explicitly bound the misalignment resulting from this assumption, showing that it is controlled by the smoothness of the underlying hazard function. In the context of deep survival models, which are notoriously prone to overfitting, this trade-off proves practically valuable. Even if the true hazard structure is not perfectly linear, the regularization prevents the model from memorizing noise, leading to the improved generalization observed in our experiments.
>
> - It Requires tuning of the Beta distribution parameter α for stability.
>     - We agree that the performance of H-Mixup depends on the choice of the Beta distribution parameter $\alpha$. As well-known, this sensitivity is a standard characteristic of the mixup framework, where $\alpha$ acts as a hyperparameter controlling regularization strength. This observation directly motivates the reviewer's suggestion regarding adaptive or learned mixing coefficients. Automating this selection would naturally address this stability issue, and we view it as a compelling direction for future work.
>
> - Limited analysis of domain shifts.
>     - We agree that analyzing robustness to domain shifts is an important and interesting direction for real-world deployment. Our primary focus in this work is to establish the foundational framework for H-Mixup in the standard survival analysis setting. We aimed to define the hazard-based interpolation mechanism and validate its core benefits. While we believe H-Mixup will likely serve as a strong component for domain adaptation, similar to mixup in classification, we leave the specific investigation of distribution shifts to future work to ensure the current study remains focused on the fundamental methodology.
>
> - Could the approach be extended to informative censoring scenarios, where censoring depends on covariates or risk, and how would it work?
>     - This is a critical and insightful point. Our current framework, like the majority of deep survival models, operates under the standard assumption of non-informative censoring (conditional on covariates; $T \perp C | X$). Extending H-Mixup to informative censoring scenarios, where censoring time $C$ and event time $T$ are dependent given $X$, is non-trivial and presents unique challenges.
>     The core difficulty lies in the fact that under informative censoring, the observed data distribution is a biased representation of the latent event process. Applying mixup directly in this setting without adjusting for the censoring mechanism (e.g., via IPCW or copulas) could inadvertently regularize the model toward a biased objective. Furthermore, it is unclear how the dependence structure between $T$ and $C$ should behave under linear interpolation of covariates.
>     Given that establishing robust deep learning baselines for informative censoring is itself an open challenge in the field, we believe developing a informative censoring-aware mixup strategy (perhaps by integrating H-Mixup with re-weighting schemes or competing risk frameworks) is a substantial independent topic. We view H-Mixup as a foundational step for the standard setting, which future work can extend to more complex censoring mechanisms.
>
> - Could adaptive or learned mixing coefficients improve alignment between interpolated and true hazards?
>     - That is an interesting direction. Conceptually, adaptive or learned mixing coefficients could function similarly to advanced mixup variants in classification, potentially allowing the model to fine-tune temporal alignment in complex regions. In our current experiments, we found that standard Beta-distributed sampling provides a stable and effective baseline. We view adaptive mixing as a natural sophistication of the H-Mixup framework. An interesting direction for future research is to determine whether existing adaptive techniques can be straightforwardly adopted, or if a strategy specific to the survival analysis context is necessary. We look forward to exploring this possibility.

---

> > ### Author Response · Authors · 2025-11-22
> >
> > - How sensitive is performance to the event/censor ratio correction scheme in highly imbalanced data?
> >     - This visual robustness translates to the quantitative results on the MNIST dataset summarized in the table below. Consistent with the toy example, H-Mixup demonstrates remarkable stability in discriminative performance, outperforming ERM in concordance index ($C^{td}$) across all tested censoring ratios and model architectures. This confirms that the regularization observed in Figure 4 (in the appendix) successfully helps the model rank risk profiles correctly, regardless of the imbalance level.
> >     Regarding the IBS, a proper scoring rule reflecting overall predictive accuracy, absolute performance generally degrades as the censoring rate increases, a natural consequence of the shrinking effective sample size. However, the relative benefit of H-Mixup varies by architecture. For fully non-parametric models (DeepHit, DeepIBS, DeepMTLR) that learn the survival distribution end-to-end, H-Mixup consistently mitigates this degradation, yielding lower IBS errors than ERM. A nuanced behavior is observed with the semi-parametric DeepCox model, where H-Mixup improves discrimination ($C^{td}$) but yields IBS scores comparable to or slightly higher than the baseline. We attribute this to the architectural difference: H-Mixup regularizes the learned partial hazard, thereby stabilizing the ranking, but the baseline hazard, which is reconstructed post-hoc and essential for computing the survival function for IBS, does not directly benefit from the mixup process. Collectively, these results indicate that while absolute accuracy is sensitive to data scarcity, H-Mixup provides a persistent relative advantage in learning robust risk representations.
> >
> > | Metric       | Censored Ratio | DeepCox             |                     | DeepHit         |                     | DeepIBS         |                     | DeepMTLR        |                     |
> > | ------------ | -------------- | ------------------- | ------------------- | --------------- | ------------------- | --------------- | ------------------- | --------------- | ------------------- |
> > |              |                | ERM                 | H-Mixup             | ERM             | H-Mixup             | ERM             | H-Mixup             | ERM             | H-Mixup             |
> > | **$C^{td}$** | 0.2            | 0.9254 (0.0016)     | **0.9363 (0.0008)** | 0.7961 (0.0031) | **0.8550 (0.0109)** | 0.8805 (0.0030) | **0.9156 (0.0055)** | 0.8955 (0.0091) | **0.9251 (0.0087)** |
> > |              | 0.4            | 0.9283 (0.0017)     | **0.9355 (0.0009)** | 0.7810 (0.0014) | **0.8111 (0.0070)** | 0.8776 (0.0021) | **0.9161 (0.0070)** | 0.9058 (0.0075) | **0.9278 (0.0026)** |
> > |              | 0.6            | 0.9305 (0.0025)     | **0.9361 (0.0015)** | 0.7825 (0.0022) | **0.8045 (0.0027)** | 0.8808 (0.0026) | **0.9136 (0.0065)** | 0.9144 (0.0077) | **0.9312 (0.0038)** |
> > |              | 0.8            | 0.9295 (0.0014)     | **0.9308 (0.0011)** | 0.7959 (0.0012) | **0.8063 (0.0028)** | 0.8659 (0.0069) | **0.9104 (0.0055)** | 0.9000 (0.0035) | **0.9299 (0.0042)** |
> > | **IBS**      | 0.2            | **0.0201 (0.0007)** | 0.0204 (0.0007)     | 0.0210 (0.0010) | **0.0147 (0.0006)** | 0.0133 (0.0006) | **0.0103 (0.0005)** | 0.0138 (0.0005) | **0.0106 (0.0005)** |
> > |              | 0.4            | **0.0222 (0.0011)** | 0.0241 (0.0015)     | 0.0367 (0.0023) | **0.0182 (0.0005)** | 0.0171 (0.0013) | **0.0123 (0.0005)** | 0.0174 (0.0009) | **0.0125 (0.0004)** |
> > |              | 0.6            | **0.0268 (0.0015)** | 0.0305 (0.0011)     | 0.0858 (0.0035) | **0.0457 (0.0026)** | 0.0244 (0.0019) | **0.0161 (0.0008)** | 0.0246 (0.0019) | **0.0175 (0.0018)** |
> > |              | 0.8            | **0.0411 (0.0014)** | 0.0478 (0.0008)     | 0.1927 (0.0085) | **0.1489 (0.0107)** | 0.0452 (0.0040) | **0.0350 (0.0036)** | 0.0456 (0.0010) | **0.0341 (0.0025)** |
> > | **ECE**      | 0.2            | **0.0364 (0.0015)** | 0.0401 (0.0020)     | 0.0418 (0.0025) | **0.0238 (0.0015)** | 0.0231 (0.0016) | **0.0137 (0.0024)** | 0.0232 (0.0019) | **0.0121 (0.0013)** |
> > |              | 0.4            | **0.0343 (0.0004)** | 0.0369 (0.0017)     | 0.0747 (0.0049) | **0.0309 (0.0027)** | 0.0220 (0.0010) | **0.0131 (0.0009)** | 0.0214 (0.0010) | **0.0113 (0.0006)** |
> > |              | 0.6            | 0.0345 (0.0009)     | **0.0342 (0.0012)** | 0.1394 (0.0039) | **0.0749 (0.0071)** | 0.0260 (0.0017) | **0.0138 (0.0011)** | 0.0239 (0.0015) | **0.0158 (0.0017)** |
> > |              | 0.8            | **0.0382 (0.0013)** | 0.0400 (0.0012)     | 0.2269 (0.0061) | **0.1841 (0.0096)** | 0.0334 (0.0027) | **0.0249 (0.0018)** | 0.0314 (0.0008) | **0.0261 (0.0011)** |

---

> ### Author Response · Authors · 2025-11-22
>
> - We have also done a qualitative analysis using a 2D toy dataset in Figure 4 to visualize how censoring impacts the learned hazard function. As the censoring rate increases from 9.5% to 47.9%, the estimates produced by ERM and Naive Interpolation become progressively irregular, failing to capture the underlying ground truth. In contrast, H-Mixup consistently preserves a smooth and coherent hazard estimate that closely resembles the low-censoring baseline. This visual evidence suggests that H-Mixup effectively encourages vicinal smoothness, preventing the deterioration of the learned hazard even when observed events become scarce.
>
> - Another potential competitor to H-Mixup is generative modeling. How would its performance compare with that of H-Mixup?
>     - Thanks for insightful suggestion. Fundamentally, the two approaches operate through distinct mechanisms. H-Mixup is grounded in VRM, where the vicinal distribution is defined via simple data augmentation as a (temporally-scaled) interpolation. In contrast, generative models aim to explicitly learn the complex underlying data distribution to synthesize new samples, which is a significantly different objective.
>     While generative models offer powerful capabilities for synthetic data generation, H-Mixup focuses on efficient regularization for discriminative tasks without the architectural complexity of training a full generative network. We hypothesize that H-Mixup may offer training stability and data efficiency for standard risk prediction. A direct comparison to delineate the specific trade-offs between vicinal augmentation and generative synthesis would be valuable, and we would like to leave this extensive benchmarking as a subject for future study.

---

### Official Review · Reviewer_1sqc · 2025-10-27

**Soundness:** 2
**Presentation:** 2
**Contribution:** 3
**Rating:** 4
**Confidence:** 4

**Summary:**

This paper introduces a novel, model-agnostic regularization approach for survival analysis by extending the mix-up technique to handle right-censored data—a challenge unique to survival settings. The authors further provide a theoretical analysis of the proposed strategy, deriving its generalization bound to justify its regularization effect. Experimental evaluations are conducted on semi-synthetic image-based survival datasets and real-world tabular datasets, where the proposed mix-up method is applied across various deep survival models with different architectures. The paper also compares the proposed approach against several existing mix-up variants adapted for survival analysis.

**Strengths:**

- Provides theoretical justification for the effect of applying mix-up on generalization performance.
- Includes other variants, such as S-Mixup, for a more comprehensive comparison of the proposed mix-up strategy in survival analysis.
- Applies the proposed mix-up strategy to a range of deep survival models with different architectures and output types, including those that predict conditional hazard functions or probability mass functions.

**Weaknesses:**

Major comments:

- The linearity derived from applying mix-up via Eq. (7) does not imply linearity in the time-to-event outcomes or the corresponding survival risks. Demonstrating the linearity of the conditional hazard function across different time steps is not a valid justification. For instance, proof of Vicinal Bias of H-Mixup is done assuming the linearity of hazard functions at the same time points, which misaligns with the proof given in Theorem 1. Moreover, Eq. (7) inflates the time-to-event outcomes, which would inevitably bias the estimator toward underestimating the risk for given samples. Please refer to Question 1 for further details.
- While provided in Thoerem 2 and Proposition 1, experiment results on how the proposed methods behave with different censoring rates is not fully observed.
- While the analysis on Vicinal Bias of H-Mixup is thoroughly done, the analysis focuses on the hazard function not the cumulative effect of survival outcomes (e.g., time-to-event outcomes or survival functions), which limits the understanding of the vicinal bias on the time-to-event outrcomes.
- The current experiments primarily compare against the authors' own variants of Mixup. While this internal comparison is thorough, this comparison must be extended to include relevant state-of-the-art Mixup techniques from the regression domain, such as C-Mixup [A]. (A straightforward implementation treating the observed time $o$ as a standard regression target would be sufficient for this comparison.)

References:
[A] H. Yao et al., "C-Mixup: Improving Generalization in Regression," NeurIPS 2022.



Minor comments:
- The application of the traditional mix-up to the survival target (Lines 152-162) is not appropriate. The event indicator $\delta$ is a binary variable (event vs. censoring) and cannot be meaningfully interpolated. Instead, applying a naive interpolation to the observed time $o$ and how it would likely introduce bias and an unwanted smoothing effect on the risk predictions will be more motivating and sound.
- In (7), how to draw \lambda should be specified.
- Showing the abbreviation for H-Mixup (presumably, hazards) could be helpful.

**Questions:**

- (Validity of Theorem 1) The theorem claims that the H-Mixup data generation process inherently guarantees hazard linearity. However, Eq. (8) does not demonstrate the linearity of $h(t|\tilde{x})$ with respect to the conditional hazards at the same time step, i.e., $h(t|x')$ and $h(t|x)$. It is unclear why the authors instead show linearity across different time steps, as this does not directly translate to linearity in the time-to-event outcomes. Furthermore, Eq. (7) appears to inflate the time-to-event outcome. For instance, when $\lambda = 0.5$ and $\delta = \delta' = 0$, the equation yields $\tilde{o} = 2o$ or $\tilde{o} = 2o'$. It is unclear how this formulation contributes to learning a correct time-to-event predictor, as it would likely cause the model to underestimate the risk.
- How does the analysis of the vicinal bias of H-Mixup extend to different types of survival models and outcome formulations? For instance, certain methods (e.g., DeepHit) do not explicitly estimate the hazard function, making the implications of the vicinal bias less straightforward.

---

> ### Author Response · Authors · 2025-11-22
>
> - The linearity derived from applying mix-up via Eq. (7) does not imply linearity in the time-to-event outcomes or the corresponding survival risks. Demonstrating the linearity of the conditional hazard function across different time steps is not a valid justification. For instance, proof of Vicinal Bias of H-Mixup is done assuming the linearity of hazard functions at the same time points, which misaligns with the proof given in Theorem 1. Moreover, Eq. (7) inflates the time-to-event outcomes, which would inevitably bias the estimator toward underestimating the risk for given samples. Please refer to Question 1 for further details.
>     - We appreciate the reviewer's thoughtful feedback regarding the H-Mixup formulation. We have revised the manuscript to clarify the distinction between direct linearity and the proposed temporally-scaled linearity.
>     - **(Local Linearity)** We apologize if the distinction was not sufficiently emphasized in the initial text. The reviewer is correct that H-Mixup encourages a temporally-scaled linearity rather than a direct linearity at fixed time points. Specifically, H-Mixup is constructed to enforce:$$h(t | \tilde{x}) \approx \lambda h(\lambda t | x) + (1-\lambda)h((1-\lambda)t | x')$$ rather than the naive unscaled version $h(t | \tilde{x}) \approx \lambda h(t | x) + (1-\lambda)h(t | x')$.
>     Applying unscaled linearity treats the observed time $O$ as a static class label rather than a duration. Merely averaging risk intensities at the same time step $t$ implies that the synthetic patient experiences risk spikes at both original event times $O$ and $O'$ simultaneously. As we illustrate in Figure 2 (revised manuscript), this results in a multi-modal distribution with artificial risk spikes, creating an incoherent hazard trajectory. In contrast, H-Mixup effectively stretches the timeline. It treats the mixing process as blending the rate of risk accumulation. This ensures the resulting hazard function is unimodal and coherent, shifting the risk peak to a single intermediate time $\tilde{O}$ rather than creating double peaks. We clarify that the proof of Vicinal Bias (Proposition 1) is indeed aligned with this temporally-scaled definition. The proof explicitly decomposes the bias into two components:
>     Spatial Term: The error arising from feature interpolation.
>     Temporal Term: The error arising from the time rescaling (warping). The proof does not assume linearity at fixed time points; rather, it bounds the discrepancy caused by the time-warping mechanism (the "Temporal Term") using the $\beta$-Hölder regularity of the hazard function. We have expanded Section 3 and Appendix A to make this decomposition explicit and demonstrate consistency with Theorem 1.
>     - **(Inflation)** We appreciate the reviewer flagging the concern that $\tilde{o}$ (which can be numerically larger than $o$) might bias the estimator toward underestimating risk. However, this numerical increase does not constitute a bias in the estimator because $\tilde{o}$ and $o$ exist in different probability spaces. H-Mixup maps the empirical distribution $P$ (with variables $X, O$) to a vicinal distribution $P_v$ (with variables $\tilde{X}, \tilde{O}$). The relationship $\tilde{O} = \min\lbrace O/\lambda, O'/(1-\lambda)\rbrace$ should be understood as a reparametrization of the time scale for the mixed samples, not a doubling of the survival duration for the same subject. The "inflation" is an artifact of this rescaled random variable. Crucially, survival models (e.g., Cox, DeepHit) are trained to predict the relationship between covariates and the hazard rate. Since the likelihood for the mixed sample is evaluated using the reparametrized time $\tilde{O}$ alongside the mixed input $\tilde{X}$, the model learns the correct association between the mixed features and the scaled risk profile. Provided the censored likelihood is evaluated under this consistent vicinal distribution, the estimator remains consistent and avoids the scale sensitivity inherent in standard regression. We have revised Section 3.2 to explicitly discuss this reparametrization and clarify why it preserves the underlying survival structure without introducing negative bias.

---

> ### Author Response · Authors · 2025-11-22
>
> - While provided in Thoerem 2 and Proposition 1, experiment results on how the proposed methods behave with different censoring rates is not fully observed.
>   - We agree that our experiments do not fully characterize how performance evolves over a wide range of censoring rates. We conjecture that, under standard Rademacher-type analysis with censoring handled via IPCW, the generalization error for a survival model $\phi$ satisfies a bound of the form $$R_{\mathbb{P}}(\phi) \leq \hat R_{n}(\phi) + 2 A(c)\hat{\mathfrak{R}}_{n}(\ell \circ \mathcal{H}) + B(c) \sqrt{\frac{\log(1/\eta)}{n}},$$
>
>     where $\hat R_{n}(\phi)$ is the empirical (possibly IPCW-weighted) risk, $\hat{\mathfrak{R}}_{n}(\ell \circ \mathcal{H})$ is the empirical Rademacher complexity of the loss class, and $A(c)$ and $B(c)$ are positive functions of the censoring rate $c$. Under standard independent-censoring assumptions, these functions typically increase with $c$ (like $1/(1-c)$), so the bound inflates for both ERM and H-Mixup as censoring rate increases. In this sense, it is expected that the performance of any survival model degrades as $c$ increases.
>
>   - Our conjecture is that the shape of these censoring-dependent terms differs between ERM and H-Mixup. Theorem 2 and Proposition 1 suggest that H-Mixup effectively reduces the complexity term by enforcing vicinal smoothness and by making more systematic use of censored samples through the vicinal distribution. In particular, if we write the bounds as
>
>     $$R_{\mathbb{P}}(\phi) \le \hat R_n(\phi) + 2 A_{ERM}(c)\hat{\mathfrak{R}}_n(\ell \circ \mathcal{H}) + B\_{ERM} (c) \sqrt{\frac{\log(1/\eta)}{n}},$$
>
>     $$R_{\mathbb{P}}(\phi) \leq \hat R_{n, \mathbb{P}_{\mathrm{v}}}(\phi) + 2 A\_{H}(c)  \hat{\mathfrak{R}}\_{n}(\ell \circ \mathcal{H}) + B\_{H}(c) \sqrt{\frac{\log(1/\eta)}{n}} + Bias\_{vic}(\phi),$$
>
>     then our conjecture is that $A_{\mathrm{H}}(c) < A_{\mathrm{ERM}}(c)$ and $B_{\mathrm{H}}(c) < B_{\mathrm{ERM}}(c)$, and that $A_{\mathrm{H}}(c)$ and $B_{\mathrm{H}}(c)$ grow more slowly in $c$ than their ERM counterparts, provided that the vicinal bias $\mathrm{Bias}_{\mathrm{vic}}(\phi)$ remains sufficiently small.
>
>   - While the theoretical derivation remains unproven, our extended toy example with varying censoring rates is consistent with this picture: as $c$ increases, ERM performance deteriorates more rapidly, whereas H-Mixup remains comparatively stable. We have clarified in the revision that this robustness to censoring is a conjectural consequence of the bound's structure, supported by this toy example and contingent on small vicinal bias, rather than a fully established theorem for all censoring mechanisms.
>
> - While the analysis on Vicinal Bias of H-Mixup is thoroughly done, the analysis focuses on the hazard function not the cumulative effect of survival outcomes, which limits the understanding of the vicinal bias on the time-to-event outrcomes.
>     - Thank you for this valuable comment. We agree that while the analysis of the hazard function in the original manuscript was decent, extending this to the survival probability provides a deeper insight into the vicinal bias of time-to-event outcomes. This suggestion helps connect our theoretical framework more robustly to the general class of survival models, where the survival function is the primary object of interest for prediction and evaluation.
>     Proposition 1 in our manuscript bounds the hazard-level bias. Following your feedback, we have now explicitly analyzed the propagation from this hazard bias to the survival function bias. Since $S(t|x) = \exp(-\int_{0}^{t} h(s|x)ds)$, the survival function bias can be bounded as follows:$$|S_H(t|\tilde{x}) - S_*(t|\tilde{x})| \le S_*(t|\tilde{x}) \cdot \left|\exp\left(\int_{0}^{t} [h_H(s|\tilde{x}) - h_*(s|\tilde{x})]ds\right) - 1\right|$$Using the hazard bias bound $B_h = \mathcal{O}(\alpha/(2\alpha+1))$ established in Proposition 1, and assuming $|h_H - h_*| \le B_h$ uniformly, we derive the approximation:$$|S_H(t|\tilde{x}) - S_*(t|\tilde{x})| \le S_*(t|\tilde{x}) \cdot (\exp(B_h \cdot t) - 1) \approx S_*(t|\tilde{x}) \cdot B_h \cdot t$$for small $B_h \cdot t$.
>     This is added to the appendix as Proposition 2.
>     - This result is important as it demonstrates that the survival bias grows linearly with the time horizon $t$, weighted by the true survival probability. This clarifies the impact on our NLL loss, which depends on both the instantaneous hazard and the survival probability. The total bias naturally inherits both the instantaneous hazard bias and this newly quantified accumulated survival bias over $[0, o_i]$. Since observed times are bounded ($o_i \leq \tau$) in practice, the cumulative bias remains controlled.
> - We updated the manuscript to include a remark immediately following Proposition 1 that details this propagation from hazard to survival-level quantities, ensuring the reader understands how the vicinal bias impacts the final time-to-event predictions.

---

> ### Author Response · Authors · 2025-11-22
>
> - The current experiments primarily compare against the authors' own variants of Mixup. While this internal comparison is thorough, this comparison must be extended to include relevant state-of-the-art Mixup techniques from the regression domain, such as C-Mixup [A]. (A straightforward implementation treating the observed time
>  as a standard regression target would be sufficient for this comparison.)
>     - We thank the reviewer for this insightful suggestion. We agree that benchmarking against state-of-the-art Mixup techniques from the regression domain is crucial to validate and strengthen the necessity of our survival-specific approach.
>     - We first conducted a qualitative analysis using the toy example presented in Figure 1 and Figure 4 (in the appendix) of the revised manuscript. These visualizations compare the decision boundaries learned by ERM, C-Mixup, and H-Mixup. The analysis highlights the distinction in how these methods enforce smoothness over the vicinity distribution. As described in the figure, while H-Mixup preserves the high central risk, C-Mixup and naive interpolation spuriously assign low risk to the center, particularly in settings with high censoring. This suggests that the vicinal bias introduced by C-Mixup does not transfer into smoothness in covariate-survival probability space.
>     In contrast, H-Mixup encourages smoothness within the hazard space, ensuring that the vicinal distribution aligns with the properties of time-to-event models.
>     - Building on this visual intuition, we extended our quantitative evaluation to include C-Mixup across our semi-synthetic benchmarks. The results, summarized in the tables above (and Table 8 in the revised manuscript, in the appendix), support our qualitative analysis. We observed that C-Mixup did not clearly outperform the standard ERM baseline across the evaluated configurations. In contrast, H-Mixup showed better performance in most cases, confirming the benefit of defining an appropriate vicinity for survival analysis.
>     - We have updated the manuscript to include both the qualitative analysis of the decision boundaries (Figure 1 and 4 captions) and the benchmark results (Table 8).
>
> - The application of the traditional mix-up to the survival target (Lines 152-162) is not appropriate. The event indicator $\delta$
>  is a binary variable (event vs. censoring) and cannot be meaningfully interpolated. Instead, applying a naive interpolation to the observed time $o$ and how it would likely introduce bias and an unwanted smoothing effect on the risk predictions will be more motivating and sound.
>     - We agree that the event indicator cannot be meaningfully interpolated. In the original version, we tried to mimic standard Mixup by defining the mixed label via a probabilistic choice: with probability $\lambda$ we used $\delta$ and with probability $1-\lambda$ we used $\delta'$. This was essentially a heuristic to preserve the mixup form and, as the reviewer notes, lacks a clear interpretation. In the revision, following the reviewer's suggestion, we additionally introduce C-Mixup, which mixes only the observed time $o$. We now include both Naive interpolation and C-Mixup in our motivation and experiments, where C-Mixup in particular serves as a comparison that highlights the bias that can arise when one ignores the event indicator in the interpolation.
>
> - We specified how $\lambda$ is drawn
> - We added the abbreviation for H-Mixup (Hazard-Mixup)

---

### Official Review · Reviewer_488W · 2025-10-31

**Soundness:** 3
**Presentation:** 4
**Contribution:** 3
**Rating:** 8
**Confidence:** 4

**Summary:**

The authors propose an extension of the mixup strategy for data augmentation to the setting of survival analysis, where some of the outcomes are censored (with the assumption of non-informative censoring). They provide theoretical evidence of the linear behavior of the proposed approach, as well as a very thorough experimental evaluation on semi-synthetic image datasets, and on real-world tabular data.

**Strengths:**

This work addresses a clear gap in the literature, that seems to prove really useful in practical applications, both for images and tabular data. The article is very clear and well-written. The experiments are very thorough, with several metrics covering all aspects of the evaluation.

**Weaknesses:**

The experiments on images are solely for semi-synthetic data. There exists datasets with survival outcomes and images, like the TCGA. It would be interesting with respect to the censoring distribution.

**Questions:**

1. How relevant is the expected calibration error for survival outcomes? There are several metrics of calibration for survival analysis, for example those implemented in the SurvivalEVAL package (Qi, Shi-ang, Weijie Sun, and Russell Greiner. "SurvivalEVAL: A comprehensive open-source python package for evaluating individual survival distributions." Proceedings of the AAAI Symposium Series. Vol. 2. No. 1. 2023.). Would you add them to the results?
2. For figures 2 and 3, are the points average over several experiments? it could be insightful to add error bars to assess the significativity of variations.
3. Why is MNIST retina not in figures 2 and 3?
4. reference "Haiyong Cui, Fan Zhang, Hui Fan, and Na Xu. Mulitdeepsurv: survival analysis of gastric cancer
based on deep learning and multimodal data. PMC, 2024." seems to be with the wrong authors and year and journal (or wrong title). Can you correct that and check carefully the other references?

---

> ### Author Response · Authors · 2025-11-22
>
> - The experiments on images are solely for semi-synthetic data. There exists datasets with survival outcomes and images, like the TCGA. It would be interesting with respect to the censoring distribution.
>     - We agree that adding experiments on a real-world dataset is beneficial to validate the method beyond semi-synthetic data. Following your recommendation, we conducted additional experiments using the COVID-19-NY-SBU dataset (Saltz et al., 2021).
>     This dataset consists of 1,384 patients and includes both image data (chest X-rays) and survival outcomes. Notably, it features a high censoring rate of 86.8%, which allows us to evaluate the robustness of H-Mixup under challenging real-world censoring distributions.
>     As shown in the table above (in the overall comment), H-Mixup mostly outperforms the baselines across all tested backbones (DeepCox, DeepHit, DeepIBS, and DeepMTLR). These results confirm that the performance gains observed in semi-synthetic data translate effectively to real-world medical imaging tasks.
>     We have integrated these results into Table 1 and added the corresponding details to Section 4.2 of the revised manuscript.
>
> - How relevant is the expected calibration error for survival outcomes? There are several metrics of calibration for survival analysis, for example those implemented in the SurvivalEVAL package (Qi, Shi-ang, Weijie Sun, and Russell Greiner. "SurvivalEVAL: A comprehensive open-source python package for evaluating individual survival distributions." Proceedings of the AAAI Symposium Series. Vol. 2. No. 1. 2023.). Would you add them to the results?
>     - Our hypothesis is that the regularization induced by H-Mixup prevents the model from becoming overconfident, thereby improving calibration performance. Empirically, we confirmed that ECE improves over ERM in most cases. This demonstrates that the H-Mixup enhances not just discrimination, but also the calibration of survival predictions. Given the relevance of accurate calibration for reliable risk prediction, we will further validate these findings by adding the additional metrics listed in the SurvivalEVAL package, such as D-Calibration, to our final results.
>
> - For figures 2 and 3, are the points average over several experiments? it could be insightful to add error bars to assess the significativity of variations.
>     - We added vertical error bars to the figures.
>
> - Why is MNIST retina not in figures 2 and 3?
>     - We added RetinaMNIST to Figure 3 and Figure 5 (which were Figure 2 and Figure 3 in the original submission).
>
> - reference "Haiyong Cui, Fan Zhang, Hui Fan, and Na Xu. Mulitdeepsurv: survival analysis of gastric cancer based on deep learning and multimodal data. PMC, 2024." seems to be with the wrong authors and year and journal (or wrong title). Can you correct that and check carefully the other references?
>     - Thank you for pointing this out. We have carefully and thoroughly reviewed all the reference items. Please see the overall response above for the details.

---

### Author Response · Authors · 2025-11-22
**Overall Comments**

We sincerely thank the reviewers for their careful and constructive feedback. In response, we have revised the manuscript. Below we highlight the main updates, and in the following we provide tailored, point-by-point responses to each reviewer's comments and questions. We have done our best to clarify and address all concerns and questions, and we would be very grateful for any further comments if any part remains unclear.

### Regarding Theorem 1 (Reviewers 1sqc and TFSk)
**Gap between H-Mixup and Theorem 1**

As pointed out by reviewer TFSk, there's a gap between H-Mixup and Theorem 1: in the manuscript we define H-Mixup in terms of observation times $\min(O/\lambda, O'/(1-\lambda))$, whereas Theorem 1 derivation is about event times $\min(T/\lambda, T'/(1-\lambda))$.
The connection between the two is not trivial and must be addressed. In the revision, we have rewritten Theorem 1 to rigorously bridge this gap. The new proof proceeds in two steps. First, we work at the latent level and define the mixed event time $\tilde T = \min(T/\lambda, T'/(1-\lambda))$; using conditional independence, we show that its hazard satisfies the temporally scaled linearity:$$h_{\tilde T}(t\mid x,x') = \lambda h(\lambda t\mid x) + (1-\lambda)h((1-\lambda)t\mid x').$$Second, we return to the observable construction and show algebraically that the H-Mixup output $(\tilde o,\tilde\delta) = (\min(O/\lambda, O'/(1-\lambda)), \tilde\delta)$ is exactly a right-censored observation of $\tilde T$, and that under our covariate-dependent independent censoring assumption, the event-time hazard of $\tilde T$ is identifiable from $(\tilde o,\tilde\delta)$. We present only this sketch in the main text and provide the full, rigorous derivation in Appendix B.1.

**Local Linearity**

We apologize for the unclarity. We agree that the local linearity encouraged by H-Mixup is not direct ($h(t | \tilde{x}) = \lambda h(t | x) + (1-\lambda)h(t | x')$) but temporally re-scaled $(h(t | \tilde{x}) = \lambda h(\lambda t | x) + (1-\lambda)h((1-\lambda)t | x'))$. The reviewer's understanding is correct, and we recognize our description may not have emphasized this alignment sufficiently.

First of all, the proposed H-mix is intentionally constructed to encourage the relationship

$$h(t | \tilde{x}) = \lambda h(\lambda t | x) + (1-\lambda)h((1-\lambda)t | x'),$$

and this is not equivalent to

$$h(t | \tilde{x}) = \lambda h(t | x) + (1-\lambda)h(t | x').$$

While applying local linearity in unscaled time is the straightforward application of mixup to survival analysis, it treats the observed time $O$ as a static label rather than a duration. By merely averaging the risk intensities $h(t|\tilde{x}) = \lambda h(t|x) + (1-\lambda)h(t|x')$, this creates a synthetic patient is implied to experience the event at both observed times $O$ and $O'$ simultaneously. This results in a multi-modal distribution (e.g., risk spikes around $t=O$ and $t=O'$) that creates less clear hazard trajectories.

We therefore adopt the temporally scaled formulation, treating observed time $O$ as a variable that depends on how fast the risk increases. Rather than stacking two separate risks, this approach effectively stretches the timeline, shifting the risk peak to a single, intermediate time $\tilde{O}$. Intuitively, this means that mixing patients acts like mixing the level of risk (where higher risk means the event happens sooner), ensuring the model learns smooth, valid curves rather than memorizing artificial double peaks. Fundamentally, H-Mixup acts not as combining two separate event outcomes, but as blending how fast the risk grows. This aligns with vicinal risk minimization by regularizing toward a coherent hazard surface, avoiding artifacts akin to independent risk spikes.

---

> ### Author Response · Authors · 2025-11-22
>
> #### Observed Time Inflation (Reviewers 1sqc and TFSk)
>
> Thank you for flagging the inflation of observed times issue. As the reviewer pointed out, numerically, $\tilde o$ can be larger than $o$ or $o'$, and this may seem to inflate the observed follow-up times.
>
> H-Mixup defines a new random variable $\tilde{O} = \min\lbrace O/\lambda, O'/(1-\lambda)\rbrace$ along with the augmented random variables $(\tilde{X}, \tilde{O}, \tilde{\Delta})$ rather than the original random variables $(X, O, \Delta)$. In this context, the follow-up time $O$ under the empirical data distribution $P$ and the follow-up time $\tilde{O}$ under the vicinal distribution $P_v$ are distinct variables. Therefore, $\tilde{o} = 2o$ should be understood as a reparametrization of the time for the mixed samples, not a doubling of the original follow-up duration, i.e., $\tilde{o}$ and $o$ are defined in different scales. This suggests that the inflation of time is an artifact of this rescaled random variable rather than a change in the underlying event process.
>
> Again, the H-mixup maps the empirical distribution $P$ to a vicinal distribution $P_v$. While this generally shifts observed times upward, it does not appear to alter the underlying survival structure. Provided the censored likelihood is evaluated under the new probability distribution, $\log S_\theta(\tilde{o} \mid \tilde{x})$, the estimator remains consistent. Moreover, since survival models predict normalized quantities (e.g., hazards or survival probabilities) rather than raw outcomes, they avoid the scale sensitivity inherent in standard regression.
>
> We made this clearer in our revised manuscript.
>
> #### Including C-Mixup \& Comparing with Other Mixup Strategies (Reviewer 1sqc and TFSk)
> As pointed out by reviewers, comparison against other mixup strategies is critical to ensure whether if the performance improvement is mainly from mixing the inputs. We compare H-Mixup with C-Mixup using four backbone models (DeepCox, DeepHit, DeepIBS, and DeepMTLR) across the semi-synthetic benchmarks. We also added C-Mixup to our motivating example (Figure 1 and Figure 4).
>
> Across all datasets and metrics, H-Mixup consistently matches or outperforms C-Mixup, while C-Mixup performs often competitive but unstable and sometimes even worse than ERM. This pattern suggests that the gains of H-Mixup cannot be explained solely by input mixing, but rather by the specific hazard-aware construction of the vicinal distribution.
>
>
> - DeepCox
>
> | Metric       | Dataset      | ERM                 | C-Mixup             | H-Mixup             |
> | ------------ | ------------ | ------------------- | ------------------- | ------------------- |
> | $C^{td}$ (↑) | MNIST        | 0.9140 (0.0028)     | **0.9315 (0.0032)** | 0.9266 (0.0014)     |
> |              | OrganMNIST3D | 0.8741 (0.0055)     | 0.8707 (0.0099)     | **0.8819 (0.0022)** |
> |              | PathMNIST    | 0.9063 (0.0099)     | 0.9024 (0.0099)     | **0.9068 (0.0040)** |
> |              | RetinaMNIST  | 0.6990 (0.0190)     | 0.7056 (0.0158)     | **0.7132 (0.0108)** |
> | IBS (↓)      | MNIST        | **0.0176 (0.0015)** | 0.0201 (0.0014)     | 0.0210 (0.0009)     |
> |              | OrganMNIST3D | 0.1022 (0.0041)     | 0.0999 (0.0089)     | **0.0949 (0.0071)** |
> |              | PathMNIST    | 0.0355 (0.0054)     | 0.0429 (0.0051)     | **0.0317 (0.0026)** |
> |              | RetinaMNIST  | 0.1777 (0.0124)     | 0.1711 (0.0089)     | **0.1656 (0.0106)** |
> | ECE (↓)      | MNIST        | **0.0206 (0.0011)** | 0.0320 (0.0020)     | 0.0274 (0.0007)     |
> |              | OrganMNIST3D | 0.0619 (0.0035)     | 0.0590 (0.0080)     | **0.0544 (0.0052)** |
> |              | PathMNIST    | 0.0302 (0.0046)     | 0.0352 (0.0024)     | **0.0288 (0.0017)** |
> |              | RetinaMNIST  | 0.0785 (0.0200)     | **0.0696 (0.0117)** | 0.0766 (0.0169)     |

---

> ### Author Response · Authors · 2025-11-22
>
> - DeepHit
>
> | Metric       | Dataset      | ERM             | C-Mixup             | H-Mixup             |
> | ------------ | ------------ | --------------- | ------------------- | ------------------- |
> | $C^{td}$ (↑) | MNIST        | 0.8479 (0.0077) | 0.8403 (0.0056)     | **0.9031 (0.0067)** |
> |              | OrganMNIST3D | 0.8425 (0.0080) | 0.8501 (0.0065)     | **0.8810 (0.0124)** |
> |              | PathMNIST    | 0.8832 (0.0107) | 0.8688 (0.0047)     | **0.9112 (0.0112)** |
> |              | RetinaMNIST  | 0.6804 (0.0188) | 0.6776 (0.0212)     | **0.7087 (0.0107)** |
> | IBS (↓)      | MNIST        | 0.0197 (0.0011) | 0.0185 (0.0017)     | **0.0151 (0.0011)** |
> |              | OrganMNIST3D | 0.0876 (0.0075) | 0.0839 (0.0055)     | **0.0701 (0.0081)** |
> |              | PathMNIST    | 0.0773 (0.0151) | 0.0679 (0.0051)     | **0.0585 (0.0059)** |
> |              | RetinaMNIST  | 0.1961 (0.0165) | 0.2113 (0.0050)     | **0.1870 (0.0137)** |
> | ECE (↓)      | MNIST        | 0.0288 (0.0017) | 0.0260 (0.0042)     | **0.0209 (0.0015)** |
> |              | OrganMNIST3D | 0.0643 (0.0111) | 0.0642 (0.0069)     | **0.0536 (0.0090)** |
> |              | PathMNIST    | 0.0886 (0.0192) | **0.0715 (0.0053)** | 0.0750 (0.0106)     |
> |              | RetinaMNIST  | 0.0643 (0.0126) | 0.1269 (0.0079)     | **0.0536 (0.0327)** |
>
>
> - DeepIBS
>
> | Metric       | Dataset      | ERM                 | C-Mixup             | H-Mixup             |
> | ------------ | ------------ | ------------------- | ------------------- | ------------------- |
> | $C^{td}$ (↑) | MNIST        | 0.8874 (0.0049)     | 0.9016 (0.0030)     | **0.9193 (0.0077)** |
> |              | OrganMNIST3D | 0.8799 (0.0088)     | 0.8768 (0.0093)     | **0.8999 (0.0116)** |
> |              | PathMNIST    | 0.8952 (0.0082)     | 0.8919 (0.0091)     | **0.9116 (0.0070)** |
> |              | RetinaMNIST  | 0.6928 (0.0101)     | 0.6842 (0.0145)     | **0.7041 (0.0223)** |
> | IBS (↓)      | MNIST        | 0.0159 (0.0011)     | 0.0163 (0.0067)     | **0.0135 (0.0015)** |
> |              | OrganMNIST3D | 0.0907 (0.0076)     | 0.0907 (0.0089)     | **0.0772 (0.0061)** |
> |              | PathMNIST    | 0.0495 (0.0019)     | 0.0652 (0.0135)     | **0.0382 (0.0049)** |
> |              | RetinaMNIST  | **0.1865 (0.0070)** | 0.2045 (0.0075)     | 0.1988 (0.0242)     |
> | ECE (↓)      | MNIST        | 0.0213 (0.0019)     | 0.0214 (0.0023)     | **0.0129 (0.0024)** |
> |              | OrganMNIST3D | 0.0631 (0.0072)     | **0.0625 (0.0084)** | 0.1319 (0.0059)     |
> |              | PathMNIST    | 0.0546 (0.0062)     | 0.0536 (0.0067)     | **0.0458 (0.0057)** |
> |              | RetinaMNIST  | 0.0848 (0.0221)     | 0.1092 (0.0111)     | **0.0474 (0.0250)** |
>
>
> - DeepMTLR
>
> | Metric       | Dataset      | ERM                 | C-Mixup         | H-Mixup             |
> | ------------ | ------------ | ------------------- | --------------- | ------------------- |
> | $C^{td}$ (↑) | MNIST        | 0.9087 (0.0065)     | 0.9160 (0.0043) | **0.9402 (0.0026)** |
> |              | OrganMNIST3D | 0.8773 (0.0040)     | 0.8826 (0.0100) | **0.9010 (0.0102)** |
> |              | PathMNIST    | 0.9036 (0.0122)     | 0.8994 (0.0087) | **0.9339 (0.0032)** |
> |              | RetinaMNIST  | 0.6999 (0.0110)     | 0.6711 (0.0098) | **0.7013 (0.0139)** |
> | IBS (↓)      | MNIST        | 0.0163 (0.0015)     | 0.0168 (0.0068) | **0.0130 (0.0012)** |
> |              | OrganMNIST3D | 0.0936 (0.0088)     | 0.0944 (0.0075) | **0.0824 (0.0101)** |
> |              | PathMNIST    | 0.0533 (0.0157)     | 0.0535 (0.0048) | **0.0332 (0.0037)** |
> |              | RetinaMNIST  | **0.1879 (0.0102)** | 0.2350 (0.0122) | 0.1892 (0.0056)     |
> | ECE (↓)      | MNIST        | 0.0203 (0.0014)     | 0.0213 (0.0028) | **0.0110 (0.0009)** |
> |              | OrganMNIST3D | 0.0660 (0.0117)     | 0.0672 (0.0061) | **0.0516 (0.0056)** |
> |              | PathMNIST    | 0.0558 (0.0164)     | 0.0461 (0.0028) | **0.0357 (0.0042)** |
> |              | RetinaMNIST  | **0.0685 (0.0096)** | 0.0905 (0.0060) | 0.0938 (0.0061)     |

---

> > ### Author Response · Authors · 2025-11-22
> >
> > #### Real-world Data (Reviewrs 488W and TFSk)
> >
> > As pointed out by reviewers, our major experimental results were from semi-synthetic data.
> >
> > We conducted additional experiments using a real-world dataset, COVID-19-NY-SBU (Saltz et al., 2021), consisting of 1,384 patients. This dataset includes both image data and survival outcomes, featuring a high censoring rate of 86.8%.
> >
> > As shown in the table below, H-Mixup outperforms the ERM and C-Mixup in the majority of cases, confirming the performance improvement of our approach in a real-world dataset. We added this result in Table 1 of the manuscript and added details in Section 4.2.
> >
> > <div align="center"><b>Results for COVID-NY-SBU dataset</b></div>
> >
> > | Model        | $C^{td}$        |                 |                     | IBS                 |                 |                     | ECE             |                 |                     |
> > | ------------ | --------------- | --------------- | ------------------- | ------------------- | --------------- | ------------------- | --------------- | --------------- | ------------------- |
> > |              | ERM             | C-Mixup         | H-Mixup             | ERM                 | C-Mixup         | H-Mixup             | ERM             | C-Mixup         | H-Mixup             |
> > | **DeepCox**  | 0.6177 (0.0558) | 0.6121 (0.0293) | **0.6433 (0.0132)** | 0.2065 (0.0228)     | 0.2183 (0.0302) | **0.1984 (0.0146)** | 0.2459 (0.0636) | 0.2562 (0.0483) | **0.2355 (0.0113)** |
> > | **DeepHit**  | 0.5798 (0.0433) | 0.5596 (0.0970) | **0.6145 (0.0498)** | 0.2500 (0.0310)     | 0.2384 (0.0468) | **0.2180 (0.0372)** | 0.3939 (0.0507) | 0.3783 (0.0673) | **0.3212 (0.0607)** |
> > | **DeepIBS**  | 0.5708 (0.0429) | 0.5845 (0.0469) | **0.5871 (0.0280)** | **0.2995 (0.0611)** | 0.3375 (0.0861) | 0.3118 (0.0453)     | 0.2776 (0.0391) | 0.2705 (0.0703) | **0.2518 (0.0700)** |
> > | **DeepMTLR** | 0.5532 (0.0612) | 0.5717 (0.0608) | **0.5994 (0.0286)** | 0.3546 (0.0908)     | 0.3454 (0.0509) | **0.3011 (0.0576)** | 0.2882 (0.0557) | 0.2675 (0.0403) | **0.2260 (0.0299)** |
> >
> >
> >
> > Saltz, J., Saltz, M., Prasanna, P., Moffitt, R., Hajagos, J., Bremer, E., ... & Kurc, T. (2021). Stony brook university covid-19 positive cases. (https://wiki.cancerimagingarchive.net/pages/viewpage.action?pageId=89096912)
> >
> > #### Incorrect References (Reviewers 488W, RLto, and TFSk)
> >
> > We sincerely apologize for the incorrect references that resulted from our naive use of LLMs. As we stated in the submission, we used LLMs for retrieval and discovery of related work; however, we did not sufficiently verify the resulting reference items. This was our responsibility, and we regret this oversight.
> >
> > We have carefully reviewed all reference items one by one to ensure that they are correct, existing and cited in appropriate contexts - this is a check that should have been completed prior to submission. In total, we revised 21 items.
> >
> > Please see the separate comment below for the revision of the reference items.
> >
> > #### Swapping the results of RetinaMNIST and OrganMNIST3D
> > We discovered that in Table 1 the rows for RetinaMNIST and OrganMNIST3D were mistakenly swapped. We have corrected their order—note that the numerical results themselves remain unchanged. We apologize for any confusion this may have caused.

---

> > > ### Author Response · Authors · 2025-11-22
> > >
> > > We sincerely apologize for the incorrect references that resulted from our naive use of LLMs. As we stated in the submission, we used LLMs for retrieval and discovery of related work; however, we did not sufficiently verify the resulting reference items. This was our responsibility, and we regret this oversight.
> > >
> > > We have now carefully reviewed all references one by one to ensure that they are correct, existing works and are cited in appropriate contexts - this is a check that should have been completed prior to submission. In total, we revised 21 items.
> > >
> > > ##### Out of these 21, we identified 9 items that do not exist. We have replaced those entries with valid, appropriate references that exist in the literature and better match the contexts in which they are cited.
> > >
> > > - Dong Shen, Rui Wu, and others. Deep Learning in Neuroimaging: Overcoming Challenges With Data Augmentation and Self-Supervised Learning. Frontiers in Neuroscience, 16, 2022.
> > >     - $\to$ Jason Smucny, Ge Shi, and Ian Davidson. Deep learning in neuroimaging: overcoming challenges with emerging approaches. Frontiers in Psychiatry, 13:912600, 2022.
> > >
> > > - Shi-ang Yi, Xuan He, Xiaolong Qiu, Mei Lin, Hu Qinghua, Benyuan Liu, and Fei Wang. Deep Contrastive Learning for Dynamic Survival Analysis with Competing Risks. Computers in Biology and Medicine, 146:105677, 2022.
> > >     - $\to$ Caogen Hong, Fan Yi, and Zhengxing Huang. Deep-CSA: Deep contrastive learning for dynamic survival analysis with competing risks. IEEE Journal of Biomedical and Health Informatics, 26(8):4248–4257, 2022.
> > >
> > > - Hrayr Harutyunyan, Adrian R. L. Oliva, and Mihaela van der Schaar. MATCH-Net: Dynamic Prediction in Survival Analysis using Convolutional Neural Networks. In NeurIPS ML4H, 2019.
> > >     - $\to$ Daniel Jarrett, Jinsung Yoon, and Mihaela van der Schaar. Match-net: Dynamic prediction in survival analysis using convolutional neural networks. arXiv preprint arXiv:1811.10746, 2018.
> > >
> > > - Haiyong Cui, Fan Zhang, Hui Fan, and Na Xu. MulitDeepsurv: survival analysis of gastric cancer based on deep learning and multimodal data. PMC, 2024.
> > >     - $\to$ Songren Mao and Jie Liu. MulitDeepsurv: survival analysis of gastric cancer based on deep learning multimodal fusion models. Biomedical Optics Express, 16(1):126–141, 2024.
> > >
> > > - Mengyang Yu, Zhucheng Tu, Guosheng Yin, Zhiming Cui, Dan Wu, and Jiebo Luo. TE-SSL: Time and Event-aware Self Supervised Learning for Survival Analysis. In Proceedings of MICCAI, 2024.
> > >     - $\to$ Jacob Thrasher, Alina Devkota, Ahmad P. Tafti, Binod Bhattarai, Prashnna Gyawali, and Alzheimer’s Disease Neuroimaging Initiative. TE-SSL: Time and Event-aware Self Supervised Learning for Alzheimer’s Disease Progression Analysis. In International Conference on Medical Image Computing and Computer-Assisted Intervention, pages 324–333, 2024.
> > >
> > > - Zhen Zhang, Song He, Zijing Zhu, Yifei Wang, Yunlong Jiao, Sheng Wang, and Pingzhong Tang. CD-Surv: a contrastive-based model for dynamic survival analysis. Bioinformatics, 38(11):3222–3229, 2022.
> > >     - $\to$ Caogen Hong, Jinbiao Chen, Fan Yi, Yuzhe Hao, Fanwen Meng, Zhanghuiya Dong, Hui Lin, and Zhengxing Huang. CD-Surv: a contrastive-based model for dynamic survival analysis. Health Information Science and Systems, 10(1):5, 2022.
> > >
> > > - Shengwei Yu, Huimin Gu, Lingxiang Yang, Yanlin Wang, Qian Liu, Biaoyan Du, and Chunlin Wang. PCLSurv: a prototypical contrastive learning-based multi-omics data integration framework for cancer survival prediction. Briefings in Bioinformatics, 26(2):bbaf124, 2025.
> > >     - $\to$ Zhimin Li, Wenlan Chen, Hai Zhong, and Cheng Liang. PCLSurv: a prototypical contrastive learning-based multi-omics data integration model for cancer survival prediction. Briefings in Bioinformatics, 26(2):bbaf124, 2025.
> > >
> > > - Cynthia S. Crowson, Elizabeth J. Atkinson, and Terry M. Therneau. Assessing calibration of prognostic time-to-event models in the presence of competing risks. Statistics in Medicine, 35(24):4521–4536, 2016.
> > >     - $\to$ Cynthia S. Crowson, Elizabeth J. Atkinson, and Terry M. Therneau. Assessing calibration of prognostic risk scores. Statistical Methods in Medical Research, 25(4):1692–1706, 2016.
> > >
> > > - Tianxiang Zou, Xiaojie Xu, Jinglong Chen, Qun Lin, Shiliang Zhang, Yuhong Liu, and others. Understanding and Improving Mixup from Directional Derivative Perspective. In International Conference on Machine Learning (ICML), 2023.
> > >     - $\to$ Yingtian Zou, Vikas Verma, Sarthak Mittal, Wai Hoh Tang, Hieu Pham, Juho Kannala, Yoshua Bengio, Arno Solin, and Kenji Kawaguchi. MixupE: Understanding and improving mixup from directional derivative perspective. In Uncertainty in Artificial Intelligence, pages 2597–2607, 2023.

---

> ### Author Response · Authors · 2025-11-22
>
> ##### Incorrect authors (8)
>
> - Yucheng Li, Wenjie Liu, and others. A Novel Data Augmentation Method for Time Series in Contrastive Representation Learning. In Advances in Neural Information Processing Systems (NeurIPS), 2023.
>     - $\to$ Berken Utku Demirel and Christian Holz. Finding order in chaos: A novel data augmentation method for time series in contrastive learning. Advances in Neural Information Processing Systems, 36:30750–30783, 2023.
>
> - Jongheon Kim, Jaehyung Choo, and Suha Song. Puzzle Mix: Exploiting Saliency and Local Statistics for Optimal Mixup. In ICML, 2020.
>     - $\to$ Jang-Hyun Kim, Wonho Choo, and Hyun Oh Song. Puzzle mix: Exploiting saliency and local statistics for optimal mixup. In International conference on machine learning, pages 5275–5285, 2020.
>
> - Anh Phuong Nguyen, Quoc-Viet Nguyen, Duc Anh Le, and others. Enhanced mixup for improved time series analysis. International Journal of Advanced Intelligence and Informatics, 2023.
>     - $\to$ Khoa Tho Anh Nguyen, Khoa Nguyen, Taehong Kim, Ngoc Hong Tran, and Vinh Quang Dinh. Enhanced mixup for improved time series analysis. International Journal of Advances in Intelligent Informatics, 11(2), 2025.
>
> - Angela Dispenzieri, Jerry A. Katzmann, Robert A. Kyle, Daniel R. Larson, Terry M. Therneau, Chris L. Colby, Ronald J. Clark, Glen P. Mead, Shaji K. Kumar, L. Joseph Melton, and S. Vincent Rajkumar. Use of nonclonal serum immunoglobulin free light chains to predict overall survival in the general population. Mayo Clinic Proceedings, 87(6):517–523, 2012.
>     - $\to$ Angela Dispenzieri, Jerry A. Katzmann, Robert A. Kyle, Dirk R. Larson, Terry M. Therneau, Colin L. Colby, Raynell J. Clark, Graham P. Mead, Shaji Kumar, Lee Joseph Melton III, and others. Use of nonclonal serum immunoglobulin free light chains to predict overall survival in the general population. In Mayo Clinic Proceedings, 87(6):517–523, 2012.
>
> - Christina Curtis, Sohrab P. Shah, Suet-Feung Chin, Gord Turashvili, Omar M. Rueda, Mark J. Dunning, Doug Speed, Andrew G. Lynch, Shamith Samarajiwa, Yinyin Yuan, and et al. The genomic and transcriptomic architecture of 2,000 breast tumours reveals novel subgroups. Nature, 486(7403):346–352, 2012.
>     - $\to$ Christina Curtis, Sohrab P. Shah, Suet-Feung Chin, Gulisa Turashvili, Oscar M. Rueda, Mark J. Dunning, Doug Speed, Andy G. Lynch, Shamith Samarajiwa, Yinyin Yuan, and others. The genomic and transcriptomic architecture of 2,000 breast tumours reveals novel subgroups. Nature, 486(7403):346–352, 2012.
>
> - Arslan Chaudhry, Christoph H. Lampert, and Muhammad Haris Khan. The Role of Local Linearity in Generalization. In NeurIPS, 2022.
>     - $\to$ Arslan Chaudhry, Aditya Krishna Menon, Andreas Veit, Sadeep Jayasumana, Srikumar Ramalingam, and Sanjiv Kumar. When does mixup promote local linearity in learned representations? arXiv preprint arXiv:2210.16413, 2022.
>
> - Qidi Wu and others. tdCoxSNN: Time-dependent Cox survival neural network for continuous-time dynamic prediction. Journal of the Royal Statistical Society: Series C, 74(1):187–208, 2025.
>     - $\to$ Lang Zeng, Jipeng Zhang, Wei Chen, and Ying Ding. tdCoxSNN: Time-dependent Cox survival neural network for continuous-time dynamic prediction. Journal of the Royal Statistical Society Series C: Applied Statistics, 74(1):187–203, 2025.
>
> - Serge Harb Fotso. Deep Neural Networks for Survival Analysis Based on a Multi-Task Framework. arXiv preprint arXiv:1801.05512, 2018.
>     - $\to$ Stephane Fotso. Deep neural networks for survival analysis based on a multi-task framework. arXiv preprint arXiv:1801.05512, 2018.
>
> ##### Incorrect journal/conferences (4)
>
> - Yiding Jiang, Behnam Neyshabur, Hossein Mobahi, Dilip Krishnan, and Samy Bengio. Fantastic Generalization Measures and Where to Find Them. In Advances in Neural Information Processing Systems (NeurIPS), pages 11780–11791, 2019.
>     - $\to$ Yiding Jiang, Behnam Neyshabur, Hossein Mobahi, Dilip Krishnan, and Samy Bengio. Fantastic generalization measures and where to find them. International Conference on Learning Representations, 2020.
>
> - Hongyu Guo, Yongyi Mao, and Richong Zhang. Augmenting Data with Mixup for Sentence Classification: An Empirical Study. In International Joint Conference on Artificial Intelligence (IJCAI), pages 4293–4299, 2019.
>     - $\to$ Hongyu Guo, Yongyi Mao, and Richong Zhang. Augmenting data with mixup for sentence classification: An empirical study. arXiv preprint arXiv:1905.08941, 2019.
>
> - Chirag Nagpal, Xinyu Li, and Artur Dubrawski. Deep Cox Mixtures for Survival Analysis. In Proceedings of the 24th International Conference on Artificial Intelligence and Statistics (AISTATS), Proceedings of Machine Learning Research, 130:480–490, 2021.
>     - $\to$ Chirag Nagpal, Steve Yadlowsky, Negar Rostamzadeh, and Katherine Heller. Deep cox mixtures for survival regression. In Machine Learning for Healthcare Conference, pages 674–708, 2021.

---

> > ### Author Response · Authors · 2025-11-22
> >
> > - Ørnulf Borgan, Bryan Langholz, Sven Ove Samuelsen, Larry Goldstein, and Janice Pogoda. Exposure Stratified Case-Cohort Designs. Lifetime Data Analysis, 9(4):355–373, 2003.
> >     - $\to$ Ornulf Borgan, Bryan Langholz, Sven Ove Samuelsen, Larry Goldstein, and Janice Pogoda. Exposure stratified case-cohort designs. Lifetime data analysis, 6(1):39–58, 2000.

---

### Note · Program_Chairs · 2025-11-26
**Submission Desk Rejected by Program Chairs**

Reviewers and AC have identified numerous hallucinated references. While submission indicates LLM usage for retrieving related work which may have caused this issue, submitting a paper with such a large number of incorrect references is grounds for desk rejection per this year's policy.